# Landsat- and Sentinel-derived glacial lake dataset in the China-Pakistan Economic Corridor from 1990 to 2020

Muchu Lesi[1], Yong Nie[1, *], Dan Hirsh Shugar[2], Jida Wang[3], Qian Deng[1, 4], Huayong Chen[1], Jianrong Fan[1]

[1]Institute of Mountain Hazards and Environment, Chinese Academy of Sciences, Chengdu 610299, China

[2]Water, Sediment, Hazards, and Earth-surface Dynamics (waterSHED) Lab, Department of Geoscience, University of Calgary, Alberta, T2N 1N4, Canada

[3]Department of Geography and Geospatial Sciences, Kansas State University, Manhattan, Kansas 66506, USA

[4]University of Chinese Academy of Sciences, Beijing 100190, China

[*]Corresponding author, nieyong@imde.ac.cn

**Abstract.** The China-Pakistan Economic Corridor (CPEC) is one of the flagship projects of
the One Belt One Road Initiative, which faces threats from water shortage and mountain
disasters in the high-elevation region, such as glacial lake outburst floods (GLOFs). An up-to-
date high-quality glacial lake dataset with parameters such as lake area, volume, and type,
which is fundamental to water resource and flood risk assessments, and predicting glacier-
lake evolutions, is still largely absent for the entire CPEC. This study describes a glacial lake
dataset for the CPEC using a threshold-based mapping method associated with rigorous
visual inspection workflows. This dataset includes (1) multi-temporal inventories for 1990,
2000, and 2020 produced from 30 m resolution Landsat images, and (2) a glacial lake
inventory for the year 2020 at 10 m resolution produced from Sentinel-2 images. The results
show that, in 2020, 2234 lakes were derived from the Landsat images, covering a total area of
$86.31\pm14.98$ km$^2$ with a minimum mapping unit of 5 pixels (4500 m$^2$), whereas 7560 glacial
lakes were derived from the Sentinel-2 images with a total area of $103.70\pm8.45$ km$^2$ with a
minimum mapping unit of 5 pixels (500 m$^2$). The discrepancy shows that Sentinel-2 can
detect a significant quantity of smaller lakes than Landsat due to its finer spatial resolution.
Glacial lake data in 2020 was validated by Google Earth-derived lake boundaries with a
median ($\pm$standard deviation) difference of $7.66\pm4.96$ % for Landsat-derived product and
$4.46\pm4.62$ % for Sentinel-derived product. The total number and area of glacial lakes from
consistent 30 m resolution Landsat images remain relatively stable despite a slight increase
from 1990 to 2020. A range of critical attributes has been generated in the dataset, including
lake types and mapping uncertainty estimated by an improved Hanshaw's equation. This
comprehensive glacial lake dataset has the potential to be widely applied in studies on water
resource assessment, glacial lake-related hazards, and glacier-lake interactions, and is freely
available at https://doi.org/10.12380/Glaci.msdc.000001 (Lesi et al., 2022).

## 1 Introduction

Glaciers in High-mountain Asia (HMA) play a crucial role in regulating climate, supporting
ecosystems, modulating the release of freshwater into rivers, and sustaining municipal water
supplies (Wang et al., 2019; Viviroli et al., 2020), agricultural irrigation, and hydropower
generation (Pritchard, 2019; Nie et al., 2021). Most HMA glaciers are losing mass in the
context of climate change (Brun et al., 2017; Maurer et al., 2019; Shean et al., 2020;
Bhattacharya et al., 2021), therefore, unsustainable glacier melt and the passing of peak water
are reducing the hydrological role of glaciers (Huss and Hock, 2018) and impacting
downstream ecosystem services, agriculture, hydropower and other socioeconomic values
(Carrivick and Tweed, 2016; Nie et al., 2021). The present and future glacier changes not
only impact the water supply for the downstream area but also alter the frequency and
intensity of glacier-related hazards, such as glacier lake outburst floods (GLOFs) (Nie et al.,
2018; Rounce et al., 2020; Zheng et al., 2021), and rock and ice avalanches (Shugar et al.,
2021). Global glacial lake number and total area both increased between 1990 and 2018 in
response to glacier retreat and climate change (Shugar et al., 2020), affecting the allocation of
freshwater resources. The Indus is globally the most important and vulnerable water tower
unit where glaciers, lakes, and reservoir storage contribute about two-thirds of the water
supply (Immerzeel et al., 2020). Ice-marginal lakes store ~1% of total ice discharge in
Greenland and accelerate lake-terminating ice velocity by ~25% (Carrivick et al., 2022). An
increasing frequency and risk of GLOFs (Nie et al., 2021; Zheng et al., 2021) is threatening
the Asian population and infrastructures in the mountain ranges, such as the China-Pakistan
Economic Corridor (CPEC), as a flagship component of One Belt One Road Initiative
(Battamo et al., 2021; Li et al., 2021). The northern section of the CPEC passes through
Pamir, Karakoram, Hindu Kush, and Himalaya mountains where droughts and glacier-related
hazards are frequent and severe (Hewitt, 2014; Bhambri et al., 2019; Pritchard, 2019),
threatening local people, the existing, under-construction and planned infrastructures, such as
highways, hydropower plants, and railways. Understanding the risk posed by water shortage
and glacier-related hazards is a critical step toward sustainable development for the CPEC.
Glacial lake inventories with a range of attributes benefit water resource assessment and
disaster risk assessment related to glacial lakes (Wang et al., 2020; Carrivick et al., 2022),
and contribute to predicting glacier-lake evolution and cryosphere-hydrosphere interactions
under climate change (Nie et al., 2017; Brun et al., 2019; Maurer et al., 2019; Carrivick et al.,
2020; Liu et al., 2020). Remote sensing is the most viable way to map glacial lakes and detect
their spatio-temporal changes in the high-elevation zones where in situ accessibility is
extremely low (Huggel et al., 2002; Quincey et al., 2007). Studies in glacial lake inventories
using satellite observations have been heavily conducted at regional scales recently, such as
in the Tibetan Plateau (Zhang et al., 2015), the Himalaya (Gardelle et al., 2011; Nie et al.,
2017), the HMA (Wang et al., 2020; Chen et al., 2021), the Tien Shan (Wang et al., 2013),
the Alaska (Rick et al., 2022), the Greenland (How et al., 2021) and the northern Pakistan
(Ashraf et al., 2017). However, the latest glacial lake mapping in 2020 is still absent along the
CPEC. Among existing studies, Landsat archival images are the most widely used due to their
multi-decadal record of earth surface observations, reasonably high spatial resolution (30 m),
and publicly available distribution (Roy et al., 2014). Freely available Sentinel-2 satellite
images show a better potential than Landsat in glacial lake mapping and inventories due to
their higher spatial resolution (10 m) and global coverage, but have only been available since
late 2015 (Williamson et al., 2018; Paul et al., 2020). Glacial lake inventories using Sentinel-
2 images are relatively scarce at regional scales, and studies of the latest glacial lake mapping
as well as comparisons of glacial lake datasets derived from Sentinel-2 and Landsat
observations are still lacking.
Discrepancies between various glacial lake inventories (Zhang et al., 2015; Shugar et al.,
2020; Wang et al., 2020; Chen et al., 2021; How et al., 2021) result from differences in
mapping methods, minimum mapping units, the definition of glacial lakes, periods, data
sources and other factors. For example, the manual vectorization method was widely adopted
at the earlier stage for its high accuracy. However, it is time-consuming associated with high
labor intensity, and is only practical at regional scales (Zhang et al., 2015; Wang et al., 2020).
Automated and semi-automated lake mapping methods, such as multi-spectral index
classification (Gardelle et al., 2011; Nie et al., 2017; Zhang et al., 2018; How et al., 2021),
have been developed to improve the efficiency of glacial lake inventories using optical
images, although manual modification is often unavoidable to assure the quality of lake data
impacted by cloud cover, mountain shadows, seasonal snow cover and frozen lake surfaces
(Sheng et al., 2016; Wang et al., 2017, 2018). Backscatter images from Synthetic Aperture
Radar (SAR) (Wangchuk and Bolch, 2020; How et al., 2021) were used to remove the impact
of cloud cover for lake mapping. Besides, other approaches such as hydrological sink
detection using DEM (How et al., 2021) and land surface temperature-based detection
method (Zhao et al., 2020) were also used for lake inventories. Different classification
methods impact the results of lake mapping and monitoring. So far, we are lacking a unified
standard for the classification system of glacial lakes (Yao et al., 2018). Existing
classification systems are generally used for their research purposes, mainly based on the
relative positions of glacial lakes and glaciers, the supply conditions of glaciers, and the
attributes of dams. In addition to different classification standards, the same type of glacial
lakes may also have different names given by different scholars. For example, ice-marginal
(Carrivick and Quincey, 2014; Carrivick et al., 2020), ice-contact (Carrivick and Tweed,
2013), and proglacial (Nie et al., 2017) lakes all represent glacial lakes sharing the boundary
with glaciers. Glacier lakes in currently available datasets have been traditionally categorized
by their spatial relationship with upstream glaciers (Gardelle et al., 2011; Wang et al., 2020;
Chen et al., 2021), and classification attributes considering the formation mechanism and the
properties of dams are rare or incomplete in the CPEC (Yao et al., 2018; Li et al., 2020).
Dam-type classification of glacial lakes provides a crucial attribute for glacier-lake
interactions and risk assessment (Emmer and Cuřín, 2021). Therefore, an up-to-date glacial
lake dataset with critical, quality-assured parameters (e.g. lake area, volume, and type) is
necessary.
This study aims to (1) present an up-to-date glacial lake dataset in the CPEC in 2020 using
both Landsat 8 and Sentinel-2 images to accurately document its detailed lake distribution;
(2) present two historical glacial lake datasets for the CPEC to show the extent in 1990 and
2000 using consistent 30-m Landsat images to reveal glacial lake changes at three time
periods (1990, 2000 and 2020); and (3) generate a range of critical attributes for glacial lake
inventories to benefit studies on water resource evaluation, risk assessment of GLOFs, glacier
–lake evolution modeling in the HMA.

## 2 Study area

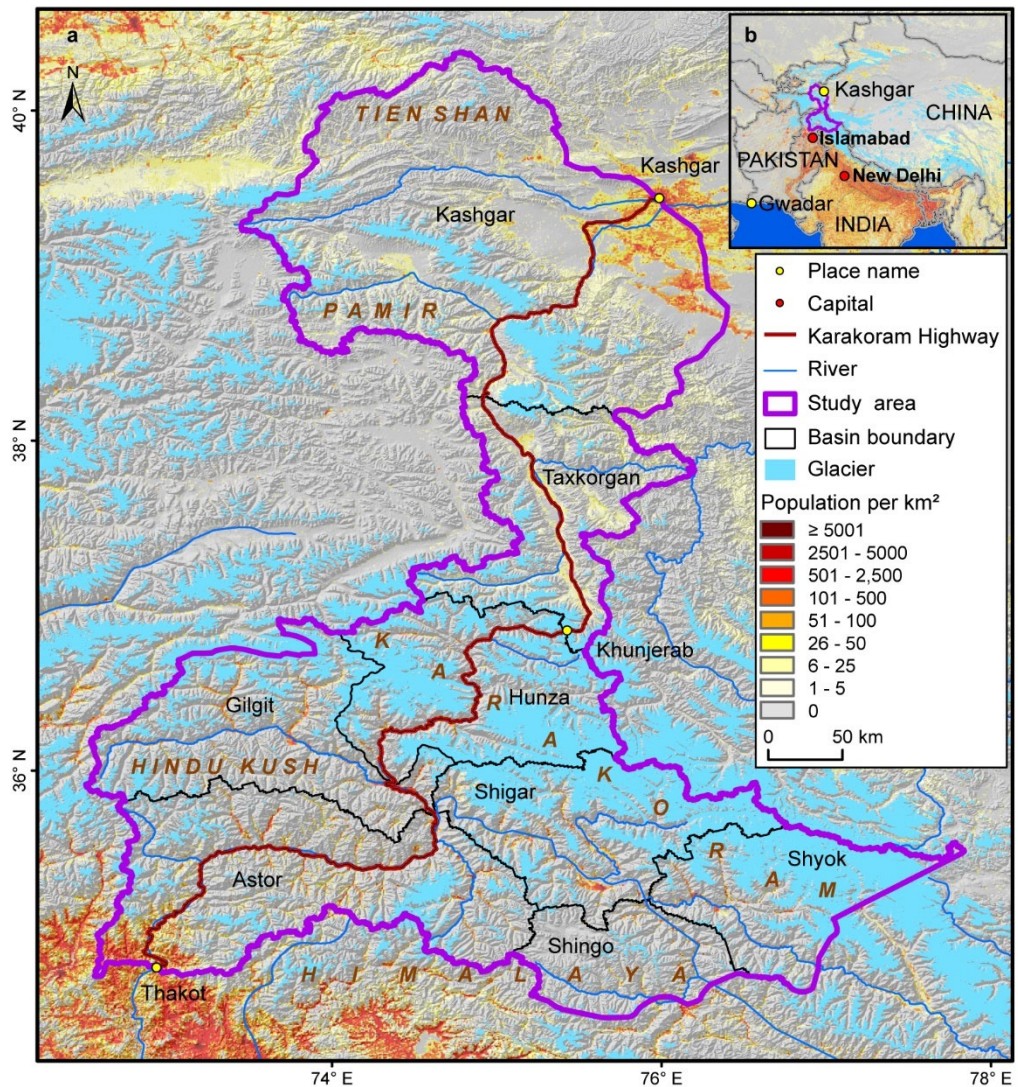

**Figure 1.** Location of the study area associated with the distribution of glaciers (RGI Consortium, 2017), mountains, basins, and population (Rose et al., 2021) (a), and its location within the CPCE (b).

The northern part of the CPEC is selected as the study area (Figure 1). The CPCE, originating from Kashgar of the Xinjiang Uygur Autonomous region, China and extending to Gwadar Port, Pakistan (Ullah et al., 2019; Yao et al., 2020), is connecting China and Pakistan via the only Karakoram Highway. The study area covers all the drainage basins along Karakoram Highway starting from Kashgar and ending at Thakot, with a total area of ~125,000 km². The upper Indus basins beyond the Pakistani-administrated border are excluded from this study due to the spatial coverage of the CPCE. The entire study area is divided into eight sub-basins, covering most of the Karakoram with the highest elevation up to 8611 m, western Himalaya and Tien Shan, eastern Hindu Kush, and the Pamir Mountains. The 9710 glaciers in the study area cover a total area of 17,447 km² and nearly 60% of glaciers are distributed in the Karakoram (5818 glaciers with a total area of 14,067.52 km²) (RGI Consortium, 2017). Most glaciers in the western Himalaya and eastern Hindu Kush are losing mass in the context of climate change (Kääb et

al., 2012; Yao et al., 2012; Brun et al., 2017; Shean et al., 2020; Hugonnet et al., 2021), whereas
the glaciers in the eastern Karakoram and Pamir have shown unusually little changes, including
unchanged, retreated, advanced and surged glaciers (Hewitt, 2005; Kääb et al., 2012; Bolch et
al., 2017; Brun et al., 2017; Shean et al., 2020; Nie et al., 2021). The spatially heterogeneous
distribution and changes of glaciers are primarily explained as a result of differences in the
dominant precipitation-bearing atmospheric circulation patterns that include the winter
westerlies the Indian summer monsoon, their changing trends, and their interactions with local
extreme topography (Yao et al., 2012; Azam et al., 2021; Nie et al., 2021).

## 3 Data sources

Both Landsat and Sentinel-2 images have been employed to map glacial lakes between 1990
and 2020 in the CPEC (Figure 2). A total number of 71 Landsat Thematic Mapper (TM),
Thematic Mapper Plus (ETM+), and Landsat 8 Operational Land Imager (OLI) images with a
consistent spatial resolution of 30 m were downloaded from the United States Geological
Survey Global Visualization Viewer (GloVis, https://glovis.usgs.gov/) to be used to create
glacial lake inventories in 1990, 2000 and 2020. High-quality Landsat-5 images around 2010
are insufficient to cover the entire study area, so we were unable to map lakes in 2010 due to
Landsat-7's scan-line corrector errors and significant cloud covers. In addition, 39 Sentinel-2
images (23 scenes in 2020) were downloaded from Copernicus Open Access Hub
(https://scihub.copernicus.eu/) to produce the 10-m resolution glacial lake inventory in 2020.
All images used in this study have been orthorectified before download, but we still find that
one Sentinel-2 image was not well matched with Landsat images, leading to the discrepancy
between the two glacial lake datasets. We manually georeferenced the shifted image to
minimize the difference between Sentinel- and Landsat-derived glacial lakes.
Cloud and snow covers heavily affect the usability of optical satellite images (Wulder et
al., 2019) and their availability in the entire study area, so we took advantage of the images
acquired before and after each of the baseline years 1990, 2000 and 2020 to construct the
glacial lake inventories. Only 4 images in 1990 (the largest covering the study area), 16
images in 2000, and 23 images in 2020 were used for matching baseline year. Spatially, high-
quality images in given baseline years were preferentially chosen, or we selected one or more
alternative images acquired in adjacent years to delineate glacial lakes by removing the effect
of cloud and snow covers. To minimize the impact of intra-annual changes on glacial lakes,
most of the used images (82% for Sentinel-2 and 75% for Landsat) were acquired from
August to October in the given baseline year with cloud coverage of <20% for each image.
For some specific scenes where cloud cover exceeded the threshold of 20%, we selected
more than one image to remedy the effect of cloud contamination (Nie et al., 2010, 2017;
Jiang et al., 2018).
Other datasets used include the Randolph Glacier Inventory version 6.0 (Pfeffer et al.,
2014; RGI Consortium, 2017) and the Glacier Area Mapping for Discharge from the Asian
Mountains (GAMDAM) glacier inventory (Sakai, 2019). These two glacier datasets were
used to determine glacial lake types, such as ice-contact, ice-dammed, and unconnected-
glacier-fed lakes. The Shuttle Radar Topography Mission Digital Elevation Model (SRTM
DEM) at a 1-arc second (30 m) resolution (Jarvis et al., 2008) was employed to extract the
altitudinal characteristics of the glacial lakes. The absolute vertical accuracy of the SRTM
DEM is 16 m (90%) (Rabus et al., 2003; Farr et al., 2007). We also applied other published
glacial lake datasets for comparative analysis. They include the glacial lake inventories of
HMA in 1990 and 2018 downloaded from http://doi.org/10.12072/casnw.064.2019.db (Wang
et al., 2020), the Third Pole region in 1990, 2000, and 2010 publicly shared at
http://en.tpedatabase.cn/ (Zhang et al., 2015), the Tibet Plateau from 2008 to 2017 accessed at
https://doi.org/10.5281/zenodo.3700282 (Chen et al., 2021), and the entire world in 1990,
2000 and 2015 provided at https://nsidc.org/data /HMA_ GLI/versions/1 (Shugar et al.,
2020). In addition, field survey data collected between 2017 and 2018 were also used to assist
in lake mapping and glacial lake type classification.

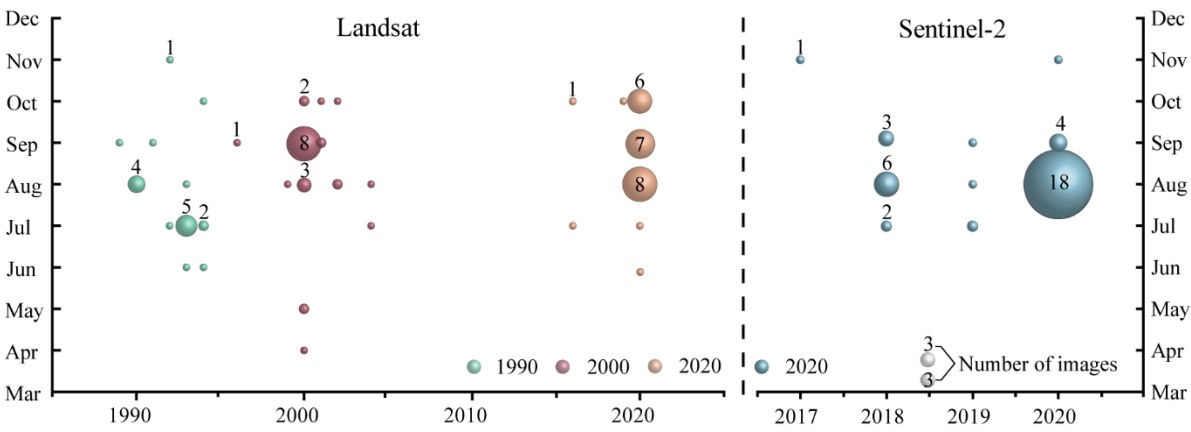

**Figure 2.** Acquisition of years and months of Landsat and Sentinel-2 images selected for glacial lake
inventories. The bubble size indicates the available high-quality image number.

## 4 Glacial lake inventory methods

### 4.1 Definition of glacial lakes

We consider a glacial lake as one that formed as a result of modern or ancient glaciation.
Contemporary glacial lakes are easily recognized using a combination of glacier inventories
and remote sensing images. Ancient glacial lakes can be identified from periglacial
geomorphological characteristics, including moraine remnants and U-shaped valleys that are
discernible from satellite observations (Post and Mayo, 1971; Westoby et al., 2014; Nie et al.,
2018; Martín et al., 2021). A 10-km buffering distance of RGI 6.0 glacier boundaries that has
been widely used in previous studies (Zhang et al., 2015; Wang et al., 2020), was created to
help map glacial lakes. A few glacial lakes in the study area (a total of 84 lakes for the
Sentinel-2 dataset and 55 lakes for the Landsat dataset in 2020) beyond the buffering zone,
located near buffering boundaries, were intentionally included due to clear evidence of
glaciation (Figure 3). Landslide-dammed lakes (Chen et al., 2017) in the buffering zone were
excluded from our inventories because of their irrelevance to glaciation. All glacial lakes in
the study area were mapped according to our definition. We were able to implement this
definition by carefully leveraging the spectral properties of glacial lakes and the periglacial
geomorphological features that are often evident in remote sensing images (see more in
sections 4.3 and 4.4).

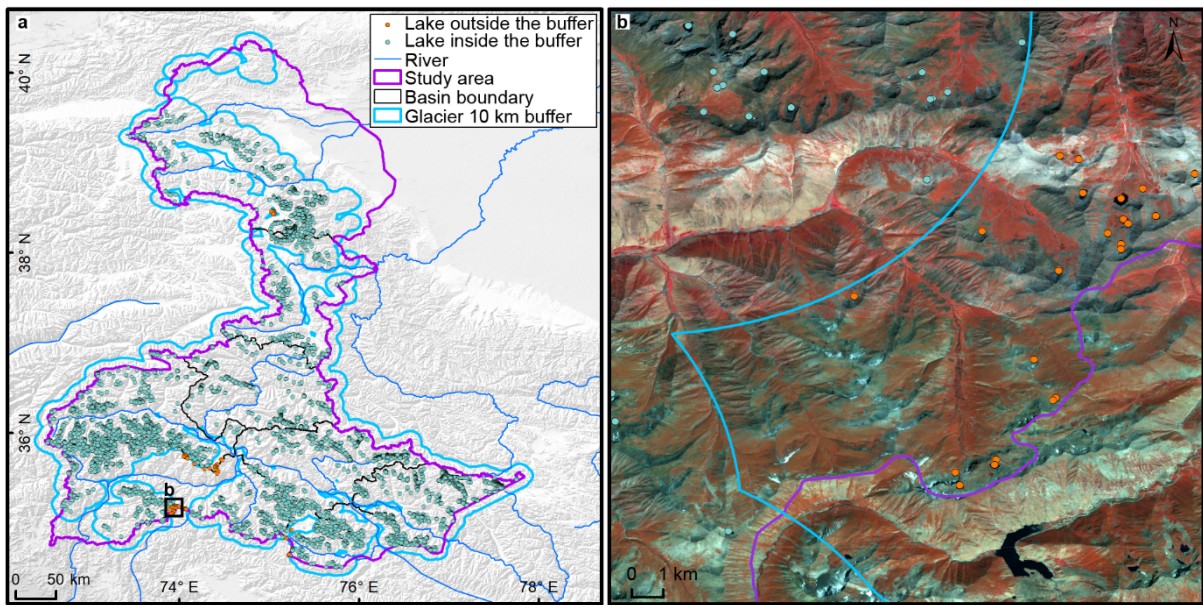

**Figure 3.** The 10-km buffer zone of RGI 6.0 glacier boundaries (a) and Sentinel-derived glacial lakes located near buffering boundary within the study area (b).

## 4.2 Interactive lake mapping

A human-interactive and semi-automated lake mapping method (Wang et al., 2014; Nie et al., 2017, 2020) was adopted to accurately extract glacial lake extents using Landsat and Sentinel-2 images, based on the Normalized Difference Water Index (NDWI) (Mcfeeters, 1996). The NDWI uses the green and near-infrared bands and is calculated by the following equation:

$$NDWI = \frac{Band_{Green} - Band_{NIR}}{Band_{Green} + Band_{NIR}} \tag{1}$$

where the green band and near-infrared band were provided by both Landsat and Sentinel multispectral images.

Specifically, the method calculated the NDWI histogram based on the pixels with each user-defined and manually-drawn region of interest. The NDWI threshold that separates the lake surface from the land was interactively determined by screening the NDWI histogram against the lake region in the imagery (Wang et al., 2014; Nie et al., 2020). This way, the determined NDWI threshold can be well-tuned to adapt to various spectral conditions of the studied glacier lakes. The raster lake extents segmented by the thresholds were then automatically converted to vector polygons. We first completed the glacial lake inventory in 2020 using this interactive mapping method, and the 2020 inventory was then used as a reference to facilitate the lake mapping for other periods.

The minimum mapping unit (MMU) was set to 5 pixels for both Landsat (0.0045 km$^2$) and Sentinel-2 images (0.0005 km$^2$) in this study. MMU determines the total number and area of glacial lakes in the dataset and varies in the previous studies, such as 3 pixels (Zhang et al., 2015), 6 pixels (Wang et al., 2020), or 9 pixels (Chen et al., 2021) for a regional scale, or 55 pixels (Shugar et al., 2020) for a global scale. While a smaller threshold leads to a large number of lakes mapped, it also generates larger mapping noises or uncertainties.

Considering this signal-noise balance and our focus on identifying prominent glacier lake
dynamics in the study area, we opted to use 5 pixels as the MMU for both Landsat and
Sentinel-2 images.
Several procedures were taken to assure the quality assurance and quality control for lake
mapping, including 1) visual inspection and modification using the threshold-based mapping
method for each lake according to Landsat and Sentinel-2 images, and Google Earth at a finer
scale overlaying preliminarily lake boundary extraction at the given period; 2) time series
check for Landsat-derived glacial lake datasets from 1990 and 2020, and cross-check
between Landsat and Sentinel-2-derived lake dataset in 2020 to reduce errors of omission and
commission; 3) topological validation of glacial lake mapping, such as repeated removal,
elimination of small sliver polygons; and 4) logical check for lake types between two
classification systems of glacial lakes. False lake extents resulting from cloud or snow cover,
lake ice, and topographic shadows (Nie et al., 2017, 2020) were modified using the previous
semi-automated mapping method based on alternative images acquired in adjacent years.
Those procedures were time-consuming but helped to minimize the effect of cloud and snow
covers, and lake mapping errors, and to maximize the quality of the produced lake product
and the derived glacial lake changes.
4.3 Classification of glacial lakes
Two glacial lake classification systems (GLCS) have been established based on the
relationship of interaction between glacial lakes and glaciers as well as lake formation
mechanism and dam material properties. In the first GLCS (GLCS1), glacial lakes were
classified into four types based on their spatial relationship to upstream glaciers: supraglacial,
ice-contact, unconnected-glacier-fed lakes, and non-glacier-fed lakes according to Gardelle et
al. (2011) and Carrivick et al. (2013). Alternatively, combining the formation mechanism of
glacial lakes and the properties of natural dam features, glacial lakes were classified into five
categories (herein named GLCS2) modified from Yao's classification system (2018):
supraglacial, end-moraine-dammed, lateral-moraine-dammed, glacial-erosion lakes and ice-
dammed lakes. Subglacial lakes were excluded due to the mapping challenge from spectral
satellite images alone. Characterization and examples for each type are provided in Table 1
and Table 2. Individual glacial lakes were categorized into the specific types for each GLCS
according to available glacier inventory data, and geomorphological and spectral
characteristics interpreted from Landsat and Sentinel images, and Google Earth. The synergy
of these two GLCSs is beneficial to predicting glacier-lake evolutions and providing
fundamental data for water resource and glacial lake disaster risk assessment.

**Table 1.** A classification system of glacial lake types (GLCS1) according to the relationship between
glacial lakes and glaciers (© Google Earth 2019). Glacier outlines are from RGI 6.0 (RGI Consortium,
2017), and the yellow marker represents the target lake.

| Lake types | Characteristics | Landsat | Sentinel-2 | Google Earth |
|---|---|---|---|---|
| Supraglacial | Lakes formed on the surface of glaciers, generally dammed by ice and thin debris.<br><br>Case location: 35°43'49.74" N 76°13'53.88" E |  |  |  |
| Ice-contact | Lakes are dammed by moraine, ice, or bedrock, supplied by glacial meltwater, and shared boundary with glaciers.<br><br>Case location: 39°09'32.40" N 73°43'12.00" E |  |  |  |
| Unconnected-glacier-fed | Lakes are currently supplied by upstream glacial meltwater but disconnected from glaciers.<br><br>Case location: 35°47'60.00" N 72°55'15.60" E |  |  |  |
| Non-glacier-fed | Lakes formed by glaciology, dammed by moraine or bed rock, and currently not supplied by glacial meltwater.<br><br>Case location: 34°50'39.99" N 74°48'29.31" E |  |  |  |


**Table 2.** A classification system of glacial lake types (GLCS2) according to the formation mechanism of
glacial lakes and dam material properties (© Google Earth 2019). The glacier outlines from RGI 6.0 (RGI
Consortium, 2017), and the yellow marker represents the target lake.

| Lake types | Characteristics | Landsat | Sentinel-2 | Google Earth |
|---|---|---|---|---|
| Supraglacial | Lakes formed on the surface of glaciers, generally dammed by ice and thin debris.<br><br>Case location:<br>36°46'7.39" N<br>74°20'7.59" E | | | |
| End-moraine-dammed | Lakes formed behind moraines as a result of glacier retreat and downwasting.<br><br>Case location:<br>35°42'50.40" N<br>73°09'57.60" E | | | |
| Lateral-moraine-dammed | Lakes formed behind lateral glacial moraine ridges and are dammed by debris, different from an ice-dammed glacial lake.<br><br>Case location:<br>38°28'45.62" N<br>75°20'52.30" E | | | |
| Glacial-erosion | Lakes formed in depressions created by glacial over-deepening. Bedrock dam dominates, partially superimposed by top moraine in rugged terrain. Dams are unclear in the satellite images.<br>Case location:<br>35°55'55.56" N<br>73°38'20.13" E | | | |
| Ice-dammed | Lakes formed behind glaciers, dammed by glacier ice (partially covered by debris on the top).<br><br>Case location:<br>35°28'31.32" N<br>77°30'46.81" E | | | |


## 4.4 Attributes of glacial lake data

A total of 18 attribute fields were input into our glacial lake datasets (Table 3). They include
lake location (longitude and latitude), lake elevation (centroid elevation), orbital number of the
image source, image acquisition date, lake area, lake perimeter, lake types of the two GLCSs,
mapping uncertainty, lake water volume and the country, sub-basin, and mountain range
associated with the lake. Amongst the attributes, lake location was calculated based on the
centroid of each glacial lake polygon associated with the DEM, N represents northing and E
represents easting. The orbital number of the image source was filled with the corresponding
satellite image, with the codes expressed as "PxxxRxxx" or "Txxxxx", where P and R indicate
the path and row for Landsat image and T represents the tile of Sentinel-2 image associated
with 5 digit code of military grid reference system. SceneID indicated identifying information
of image source for Landsat or Sentinel-2, consisting of the orbital number, sensor ID, and
acquisition date (YYYYMMDD) for Landsat image, or the orbital number and acquisition date
(YYYYMMDD) for Sentinel-2 image. Area and perimeter were automatically calculated based
on glacial lake extents. Lake water volume was estimated by an area-volume empirical
equation (Cook and Quincey, 2015). Lake types were attributed using the characterization and
interpretation marks described in Section 4.3. Mapping uncertainty was estimated using our
modified equation which will be introduced in section 4.5 and the appendix tutorial. Located
country, sub-basin, and the mountain range of each glacial lake were identified by overlapping
the geographic boundaries of countries, basins, and mountain ranges.
**Table 3.** Attributes of glacial lake dataset.

| Field Name | Type | Description | Note |
|---|---|---|---|
| FID or OBJECTID | Object ID | Unique code of glacial lake | Number |
| Shape | Geometry | Feature type of glacial lake | Polygon |
| Latitude | String | Latitude of the centroid of glacial lake polygon | Degree minute second |
| Longitude | String | Longitude of the centroid of glacial lake polygon | Degree minute second |
| Elevation | Double | Elevation of the centroid of glacial lake polygon | Unit: meter above sea level |
| SceneID | String | Scene ID of image source for Landsat or Sentinel-2 | PxxxRxxx_xxxDYYYYMMDD or Txxxxx_YYYYMMDD |
| ACQDATE | String | The acquisition date of the source image | YYYYMMDD |
| GLCS1 | String | The first classification system of glacial lakes based on the relationship of interaction between glacial lakes and glaciers | Supraglacial, Ice-contact, Unconnected-glacier-fed, and None-glacier-fed |
| GLCS2 | String | The second classification system of glacial lakes is based on lake formation mechanism and dam material properties | Supraglacial, End-moraine-dammed, Lateral-moraine-dammed, Glacial-erosion and Ice-dammed |
| Basin | String | Basin name where the glacial lake locates in | |

| Field Name | Type | Description | Note |
|---|---|---|---|
| Mountain | String | Mountain name where the glacial lake locates in | |
| Country | String | Country name where the glacial lake locates in | |
| Perimeter | Double | The perimeter of the glacial lake boundary | Unit: meter |
| Area | Double | Area of glacial lake coverage | Unit: square meter |
| AreaUncer | Double | Area uncertainty of glacial lake mapping estimated based on modified Hanshaw's equation (2014) | Unit: square meter |
| Operator | String | The operator of the glacial lake dataset | Muchu, Lesi |
| Examiner | String | Examiner of glacial lake dataset | Yong, Nie |
| Volume | Double | The water volume of a glacial lake estimated by an area-volume empirical equation | Unit: cubic meter |


4.5 Error and uncertainty assessment
4.5.1 Improved uncertainty estimating method
We modified Hanshaw's (2014) equation that had been used to calculate lake-area mapping
uncertainty. Lake perimeter and displacement error are widely used to estimate the
uncertainty of glacier and lake mapping from satellite observation (Carrivick and Quincey,
2014; Hanshaw and Bookhagen, 2014; Wang et al., 2020). Hanshaw and Bookhagen (2014)
proposed an equation to calculate the error of area measurement by the number of edge pixels
of the lake boundary multiplied by half of a single pixel area. The number of edge pixels is
simply calculated by the perimeter divided by the grid size. The equation is expressed below:
$$Error(1\sigma) = \frac{P}{G} \times 0.6872 \times \frac{G^2}{2} \tag{2}$$
$$D = \frac{Error(1\sigma)}{A} \times 100\% \tag{3}$$
Where $G$ is the cell size of the remote sensing imagery (10 m for Sentinel-2 image and 30 m
for Landsat image). $P$ is the perimeter of individual glacial lake (m), and the coefficient of
0.6872 (1σ), which means nearly 69% of the edge pixels are subject to errors (Hanshaw and
Bookhagen, 2014), was chosen assuming that area measurement errors follow a Gaussian
distribution. Relative error ($D$) was calculated by equation 3, in which A is the area of an
individual glacial lake.
In the original equation 2, the number of edge pixels varies by the shape of the lake and is
indicated by $\frac{P}{G}$. However, the pixels in the corner are double-counted (Figure 4). The total
number of repeatedly calculated edge pixels equals the number of inner nodes. Therefore, we
adjusted the calculation of the actual number of edge pixels as the maximum of edge pixels
$(\frac{P}{G})$ subtracting the number of inner nodes. Accordingly, the equation of uncertainty
estimation for lake mapping is modified as below:

$$Error(1\sigma) = (\frac{P}{G} - N_{Inner}) \times 0.6872 \times \frac{G^2}{2} \tag{4}$$


Where $N_{Inner}$ is the number of inner nodes (inflection points) of each lake. The modified
equation is also suitable for lakes with islands (as illustrated in Figure 4b).
For polygons without islands (Figure 4a), use the following equation:

$$N_{Inner} = (\frac{N_{Total} - 4 - 1}{2}) \tag{5}$$


$N_{Total}$ is the total number of nodes, including both the outer and inner. $N_{Total}$ is calculated by
the "Field Calculator" in ArcGIS, in some cases, it is necessary to remove the redundant
nodes before calculating the total number of nodes (See the Appendix for more details). An
inner node is a polygon vertex where the interior angle surrounding it is greater than 180
degrees. An outer node is the opposite of the inner node, where the interior angle is less than
180 degrees. We found that the outer nodes are usually four more than the inner nodes in our
glacial lake dataset. The total nodes in ArcGIS contain one overlapping node to close the
polygon, meaning the endpoint is also the start point. This extra count was deleted from the
calculation (equation 5).
For polygons with island (Figure 4b) use the following equation:

$$N_{Inner} = (\frac{N_{Total} - (N_{Island} + 1) \times 5}{2}) \tag{6}$$


$N_{Island}$ is the number of islands within each polygon. A calculation method of $N_{Island}$ is
given in the Appendix.

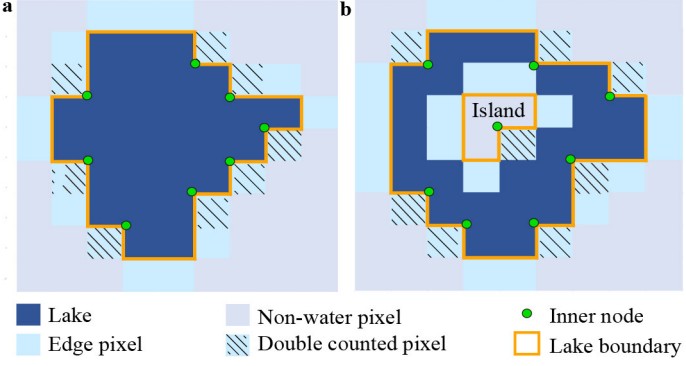


**Figure 4.** Sketch of estimating the actual edge pixels for uncertainty calculation of individual glacial lakes
(with (a) and without islands (b)).

4.5.2 Validation of glacial lake mapping
A total of 89 glacial lakes were selected by stratified random sampling and manually digitized
based on the Google Earth images in circa 2020 with a spatial resolution of ∼2 m acquired
from WorldView, GeoEye, Pleiades, etc. satellites (© 2022 Maxar technologies and © 2022
CNES/Airbus) to further validate the absolute error of our glacial lake products in 2020 due
to lacking field measurements for glacial lakes in the study area. During the sampling, we set
a minimum lake area to be 4500 $m^2$ and a relative difference between Landsat- and Sentinel-
derived lake areas of less than 18% (nearly equaling the average relative error of ±17.36% for
Landsat lake mapping) to minimize the effect of lake changes from multi-temporal satellite
observations in circa 2020. The 89 sample lakes range from 0.005 $km^2$ to 0.802 $km^2$ with a
median (standard deviation) size of 0.047±0.134 $km^2$ and a total area of 8.033 $km^2$ for
Landsat-derived dataset, and range from 0.005 $km^2$ to 0.849 $km^2$ with a median (standard
deviation) size of 0.045±0.144 $km^2$ and a total area of 8.447 $km^2$ for Sentinel-derived dataset.

## 5 Results

5.1 Glacier lake distribution and changes observed from Landsat
We mapped 2,234 glacial lakes for 2020 across the studied CPEC from Landsat-8 images,
with a total area of 86.31±14.98 $km^2$ (Figure 5a and b). Unconnected-glacier-fed lakes are
dominant in the first classification system, followed by non-glacier-fed lakes (Figure 6)
whereas glacial-erosion lakes dominate at both number (1478) and area (57.02 $km^2$) in the
second classification system (Figure 7), followed by end-moraine-dammed lakes and
supraglacial lakes. Among the classified lakes, 137 are ice-contact lakes and cover an area of
5.56 $km^2$, implying a higher mean size of ice-contact lakes than supraglacial lakes.

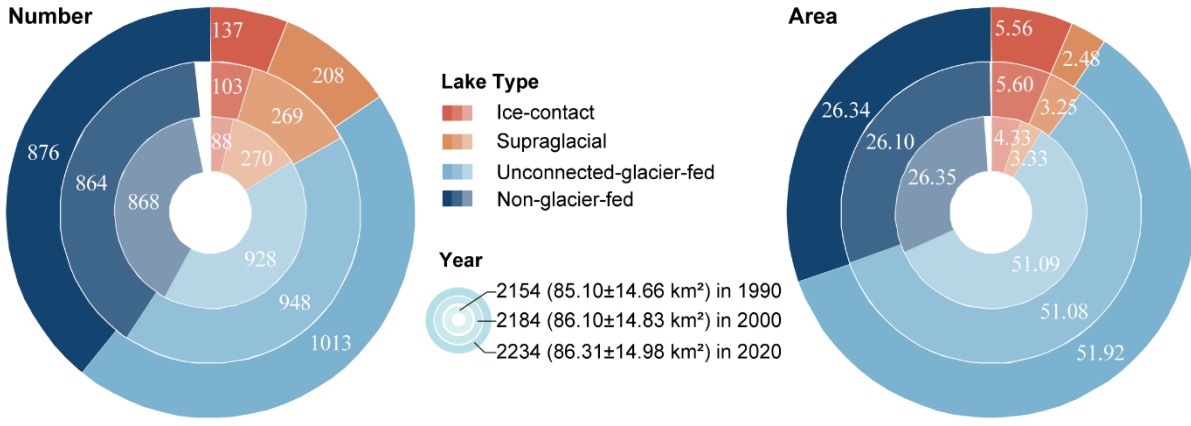


**Figure 5.** Distribution of glacial lakes in 2020 extracted from Landsat (a, b) and Sentinel-2 (c, d) images.
Panels a and c are classified by GLCS1 and GLCS2 for sub-graph b and d.


**Figure 6.** The number and area of different types of glacial lakes are classified based on the condition of
glacier supply in the study area (GLCS 1). The outermost ring represents glacial lake data in 2020, the
middle ring for 2000, and the innermost ring for 1990. Lake number and area in 2020 were selected as
references, meaning a concept of "100 %" for a complete ring. Labeled values are scaled in degrees rather
than the radius of rings.

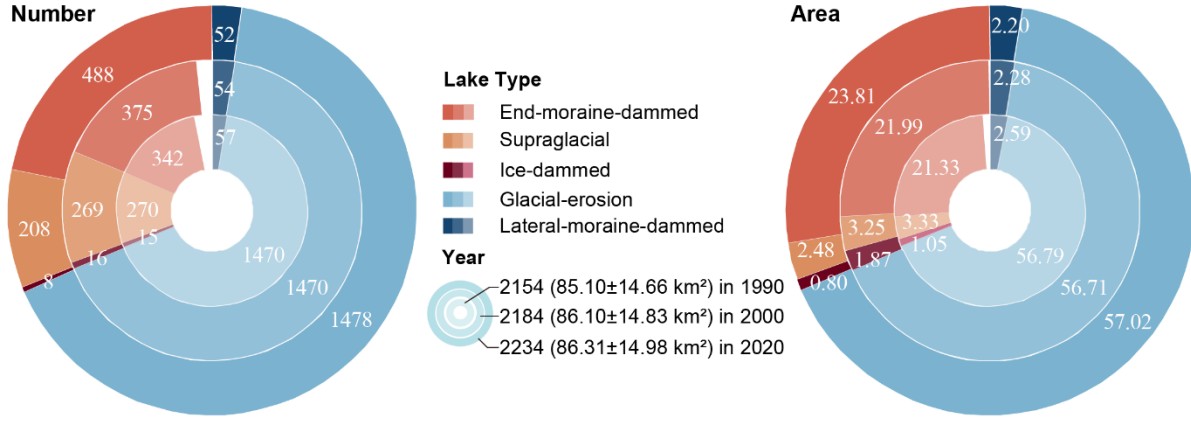

**Figure 7.** The number and area of different types of glacial lakes are classified based on glaciation and the
nature of the dam in the study area (GLCS 2). The outermost ring represents glacial lake data in 2020, the
middle ring for 2000, and the innermost ring for 1990. Lake number and area in 2020 were selected as
references, meaning a concept of "100 %" for a complete ring. Labeled values are scaled in degrees rather
than the radius of rings.

The total number and area of glacial lakes in the study remain relatively stable with a
slight increase between 1990 and 2020, and the changes in count and area among various
types of glacial lakes vary substantially (Figure 6 and Figure 7). From 1990 to 2020, the total
number of glacial lakes increased by 80 or 3.70%, while the area grew by 1.21 km$^2$ (or
1.42%). In GLCS1, unconnected-glacier-fed lakes have the largest increase in number,
followed by ice-contact and non-glacier-fed lakes, whereas supraglacial lakes decreased by
62 in count. Ice-contact lakes expanded by 1.24 km$^2$ (equaling an increase of 26% in ice-
contact lakes), contributing one-third of the total area increase. Supraglacial lakes decreased
by 0.85 km$^2$ in area whereas the areas of unconnected-glacier-fed and non-glacier-fed lakes
remained stable as a result of disconnections from glaciers (Figure 6). In GLCS2, end-
moraine-dammed lakes increased by 2.48 km$^2$ and contributed most of the glacier lake area
expansion, whereas supraglacial, ice-dammed, and lateral-moraine-dammed lakes decreased
slightly in both number and area. Glacial-erosion lakes accounted for the maximum
percentage (about 66% for both count and area) in each period and remained stable (Figure
7).

5.2 Glacier lake distribution observed from Sentinel-2
Sentinel-derived results show that there are 7,560 glacial lakes (103.70±8.45 km$^2$) in 2020
across the entire CPEC under an MMU of 5 pixels (500 m$^2$). Compared with Landsat-derived
product, glacial lakes from Sentinel-2 have similar spatial distribution characteristics (Figure
5); meanwhile, a larger quantity of glacier lakes, with more accurate boundaries and a greater
total lake area, were generated from Sentinel-2 images (Table 4). The smallest size class
(0.0005-0.0045 km$^2$) contains the maximum lake number (4,969) but the least lake area
(7.73±2.62 km$^2$), which is not available in the Landsat-derived lake data due to a coarser
spatial resolution. In each size class, the overlap ratios are greater than 85% in count and
area, and there are also a higher number and larger area of glacial lakes from Sentinel than
that from Landsat images. Sentinel-2 images (10 m) with a finer spatial resolution produce
more glacial lakes than those from Landsat images (30 m). The discrepancy is mainly
attributed to the inconsistency of spatial resolutions and image acquisition dates, as discussed
in section 6.2.
**Table 4.** Count and area of glacial lakes mapped from Sentinel-2 and Landsat images in 2020 in various
size classes.

| Lake size km$^2$ | Glacial lakes from Sentinel-2 count (km$^2$) | Glacial lakes from Landsat count (km$^2$) | Overlap % (%) |
|---|---|---|---|
| 0.0045-0.05 | 2182  (35.52±3.72) | 1870  (31.47±9.57) | 85.70 (88.60) |
| 0.05-0.1 | 237  (16.37±0.89) | 204  (14.07±2.18) | 86.08 (85.95) |
| 0.1-0.2 | 122  (16.88±0.68) | 115  (15.91±1.83) | 94.26 (94.25) |
| ≥0.2 | 50  (27.20±0.54) | 45  (24.86±1.40) | 90.00 (91.40) |
| Total | 2591  (95.97±5.83) | 2234  (86.31±14.98) | 86.22 (89.93) |

Note: Second column excludes 4969 (7.73±2.62 km$^2$) lakes in the 0.0005 to 0.0045 km$^2$ range. Overlap % (%) represents the
ratios between our Landsat-derived dataset and Sentinel-derived product in count and area, respectively.

# 6 Discussions

## 6.1 Uncertainty and error of lake mapping

The uncertainty estimated from our improved equation shows that the relative error of
individual glacial lakes decreases when lake size increases or the cell size of remote sensing
images reduces (Lyons et al., 2013; Carrivick and Quincey, 2014) (Figure 8). Total area
errors of glacial lakes in the study area are approximate ±14.98 km$^2$ and ±8.45 km$^2$ in 2020
for Landsat and Sentinel-2 datasets, respectively, and the average relative errors are ±17.36%
and ±8.15%. Generally, small lakes have greater relative errors. For example, the mean
relative error is 35.38% for Landsat-derived glacial lakes between 0.0045 and 0.1 km$^2$ and
10.63% for glacial lakes greater than 0.1 km$^2$. The mean area error of Sentinel-derived glacial
lakes is almost one-third of that extracted from Landsat images for glacial lakes of all or
specific size groups. Because the relative error was estimated as a function of satellite image
spatial resolution and lake perimeter, the calculated error for a large lake is proportionally
smaller than that of a small lake (Salerno et al., 2012) and the error for Landsat-derived lake
is naturally greater than that of Sentinel-derived lake at the same size group.

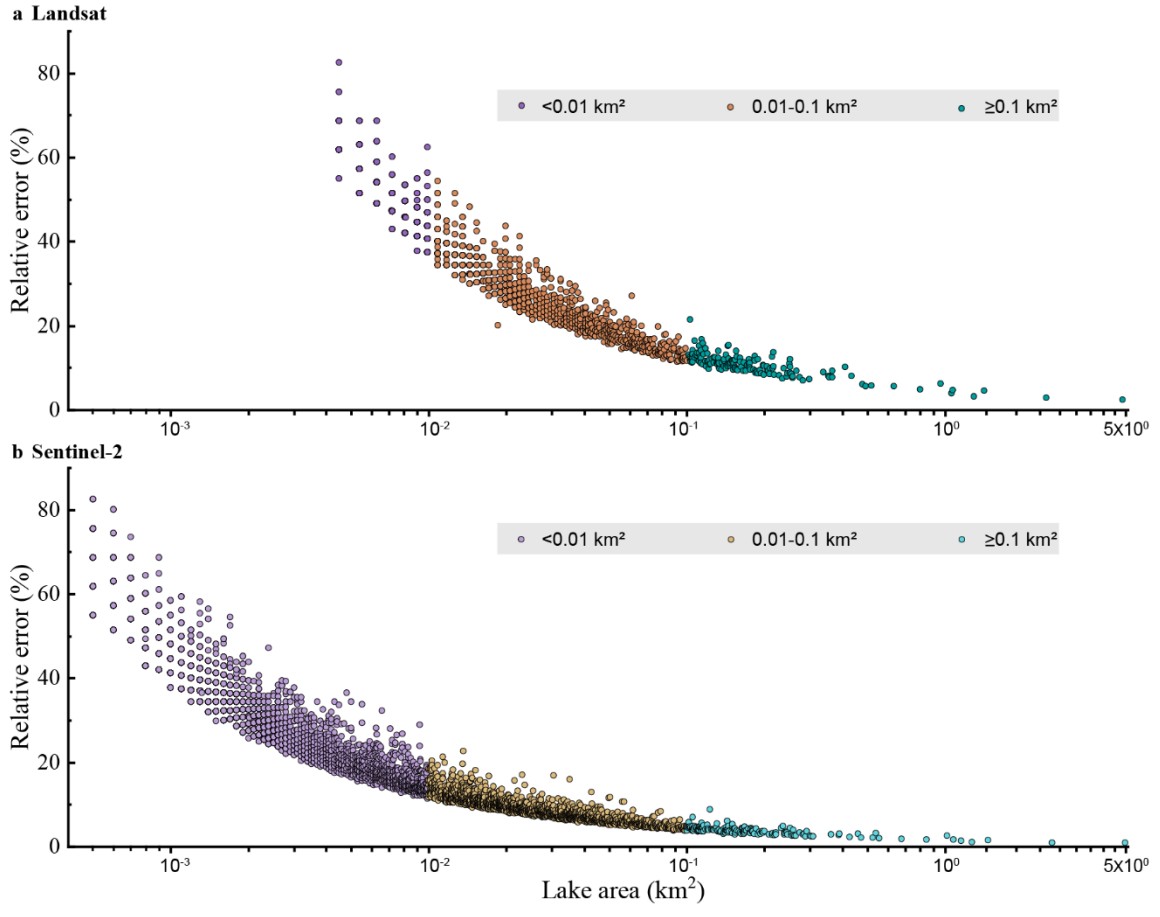

**Figure 8.** The estimated relative error for glacial lakes of all or specific size ranges in the study area. Error
estimation is based on the modified equation and lake data extracted from Landsat (a) and Sentinel-2
images (b).

Our Landsat- and Sentinel-derived glacial lake dataset match well lake boundaries in Google
Earth higher resolution images (Figure 9). The mean difference in area is 0.005 km$^2$ between
Landsat- and Google Earth-derived lakes and 0.001 km$^2$ between Sentinel- and Google Earth-
derived lakes, and major validation samples (84/89) are within the confidence interval of
95%, indicating high accuracy in lake mapping (Figure 9c and d). The error of 89 sample
lakes is 5.48% in the total area between Landsat- and Google Earth-derived data, and 0.61%
for Sentinel- and Google Earth-derived data. The median (±standard deviation) in a
discrepancy of the individual lake area is 7.66±4.96 % for Landsat- and Google Earth-derived
data, and 4.46±4.62 % for Sentinel- and Google Earth-derived data. Our glacial lake dataset
shows satisfactory mapping accuracy, although Sentinel-derived lake data performs more
accurately than those from Landsat images. We also validated the sampling of Landsat-
derived 89 lakes by the existing Landsat-extracted lake data produced by Wang et al. (2020).
A total of 83 lakes are available in Wang's data with a mean difference of 0.005 km$^2$ in the
lake area (Figure A8). This also shows an improvement in our lake product in contrast to the
existing dataset.

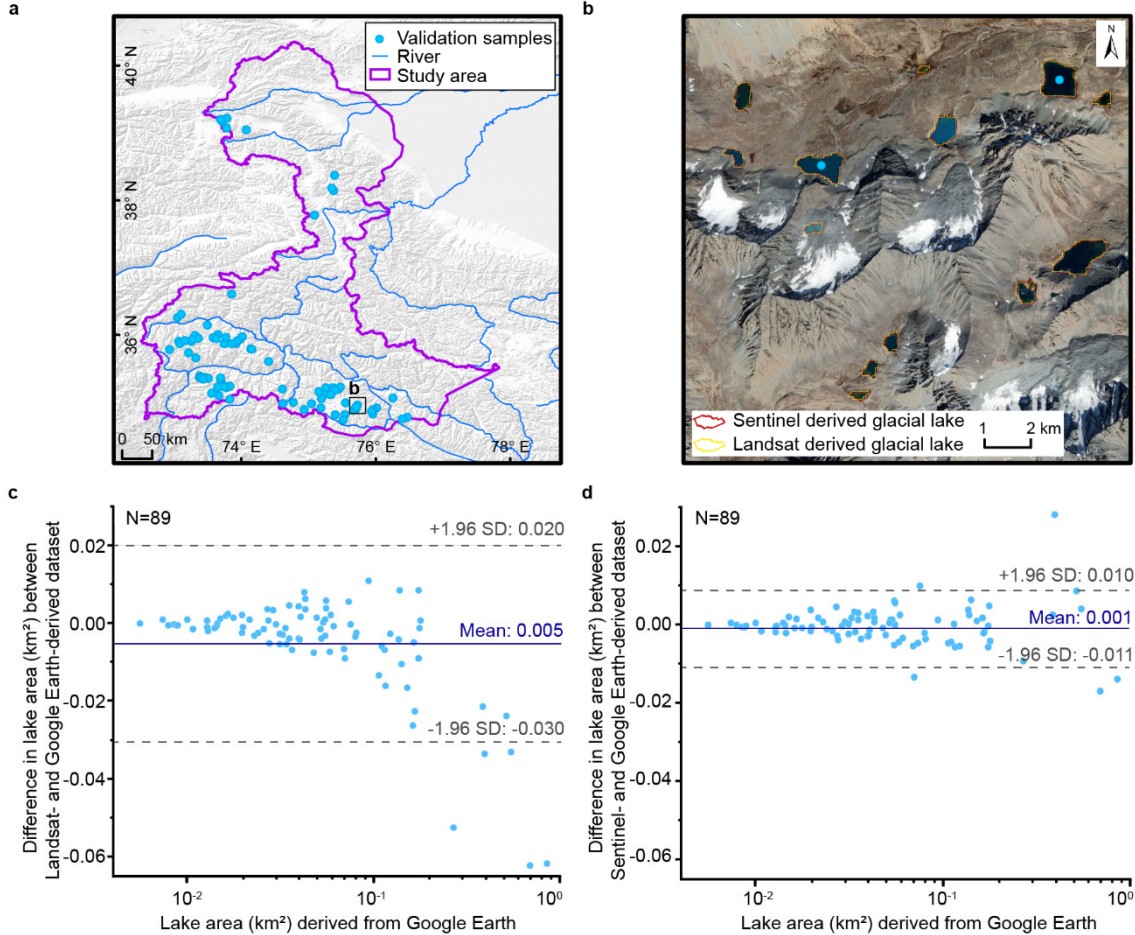

**Figure 9.** Distribution of the validation sample (a), visual comparison of glacial lakes derived from Landsat and Sentinel-2 images overlaying Google Earth imagery (© Google Earth 2019) in a zoomed site (b), and differences between our glacial lake product (mapped from Landsat and Sentinel-2 images) and the validation reference (digitized from Google Earth at a finer scale) (c and d).

6.2 Comparison of Sentinel- and Landsat-derived products

Glacial lakes from Landsat and Sentinel-2 images have high consistency in number and area with overlap rates from approximately 86% to 94% for all lakes greater than 0.0045 km$^2$ (Table 4), indicating a good potential for coordinated utility with Landsat archived observation (Figure 10). Lake extents extracted from Landsat and Sentinel images match well for various types and sizes (Figure 10 and Figure 11, Table 4). The best consistency rate reaches 94% for the glacial lakes between 0.1 km$^2$ and 0.2 km$^2$. The difference in the area of glacial lakes extracted from Landsat and Sentinel-2 images generally lies within the uncertainty ranges.

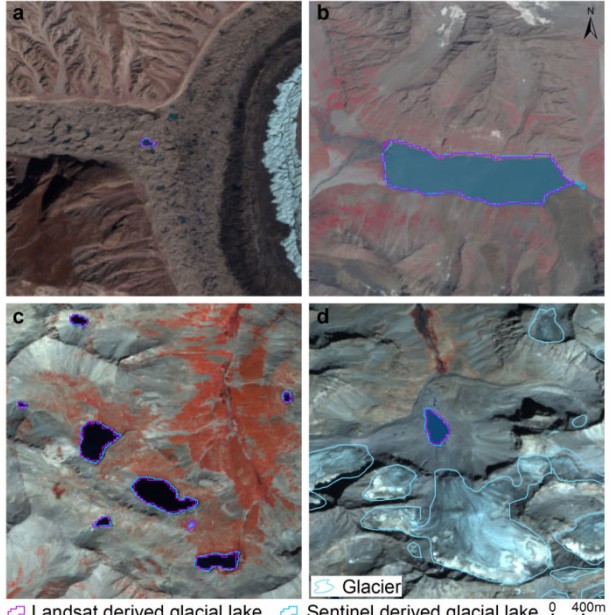

Landsat derived glacial lake · Sentinel derived glacial lake · 0 400m

**Figure 10.** High consistency of lake extents extracted from Landsat and Sentinel-2 images. Lake types shown include supraglacial (a), glacier-fed moraine-dammed (b), unconnected glacial-erosion lake without glacier melt supply (c), and glacier-fed moraine-dammed lakes (d).

The spatial resolution of satellite images plays a primary role in the discrepancies in count and area of glacial lakes extracted from Landsat (30 m) and Sentinel-2 (10 m) observations. Due to a finer spatial resolution, Sentinel-2 images can extract more glacial lakes and more accurate extents than those from Landsat images. We set the same 5 pixels as the MMU for both Landsat and Sentinel-2 images, which corresponds to a minimum area of 0.0045 km$^2$ and 0.0005 km$^2$, respectively. The minimum mapping area results in generating nearly 5000 more lakes from Sentinel-2 images than from Landsat images, causing the greatest discrepancy in number, such as Figure 11. Small lakes such as supraglacial lakes play an important role in analyzing glacier evolution and supraglacial drainage systems (Liu and Mayer, 2015; Miles et al., 2018), implying a potential of our dataset to be applied in studies of glacier-lake evolutions. Meanwhile, Sentinel-2 images can depict boundaries of glacial lakes with lower uncertainty, as some small islands and narrow channels (Figure 11b and c) were mapped from Sentinel-2 imagery that was unable to be detected in Landsat imagery.

In addition to the difference in image resolution, different acquisition dates between Sentinel-2 and Landsat images can also contribute to the discrepancy between those two glacial lake datasets. The total number of supraglacial lakes and ice-dammed glacial lakes are less than 300, but those lakes are controlled by glacier movement and temperature changes (Liu and Mayer, 2015; Miles et al., 2018), which vary faster with time than relatively stable glacial-erosion and moraine-dammed lakes. Acquiring same-day images from the two sensors was not always possible due to the impacts of cloud contaminations, topographic shadows, snow cover, and revisit periods (Williamson et al., 2018; Paul et al., 2020). Despite our efforts of leveraging all available high-quality images, the overlap of acquisition dates between Landsat and Sentinel-2 images for the same location is relatively low (only 7 scenes of Sentinel-2 images or 112 glacial lakes in 2020) in this study area, and the consequential

temporal gaps led to a difference in the number and area of the derived glacial lakes. As
exemplified in Figure 11d, the mapped supraglacial lakes in the same location exhibit a
considerable discrepancy, which is likely a joint consequence of both sensor difference and
glacier lake evolution.

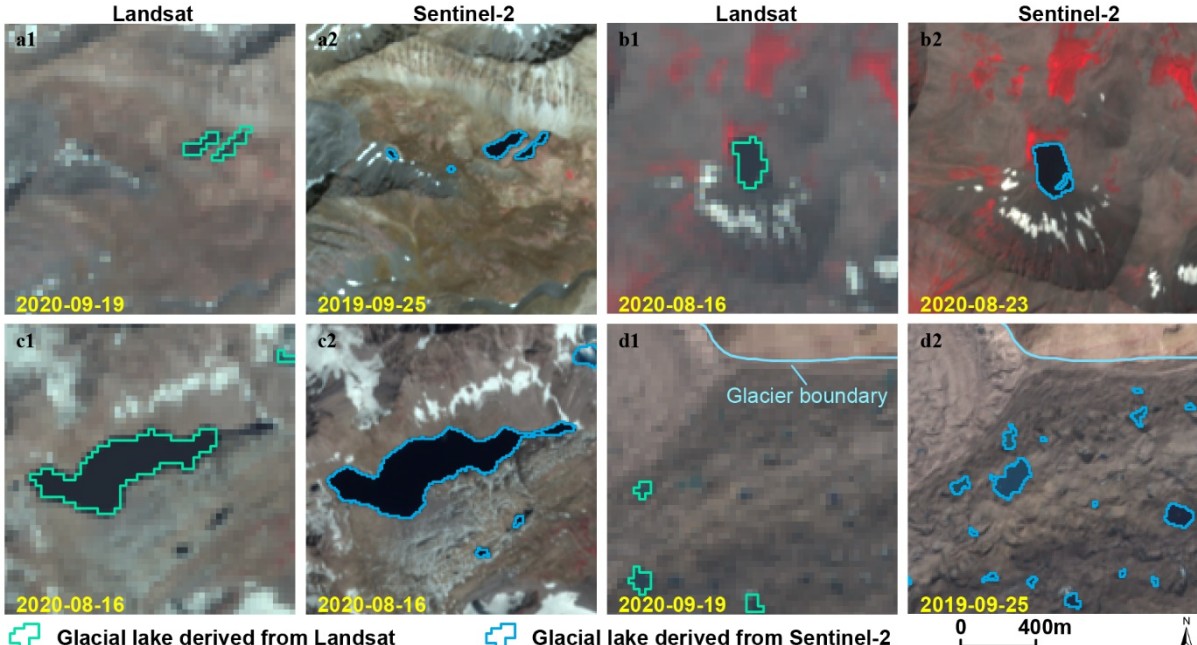

**Figure 11.** The discrepancy of lake extents extracted from Landsat and Sentinel-2 images.

6.3 Comparison with the previous similar dataset
An increasing number of glacier lake datasets have been released over the past years, and
most of them were produced from long-term Landsat archives. Regional glacial lake datasets
using Sentinel images are scarce. The lack of Sentinel-derived glacial lake data in the study
area makes it impossible to compare. Here we selected four available glacial lake datasets to
compare with our Landsat-derived dataset at the same MMU and study area.
We provide the latest glacial lake dataset (in 2020) and the most long-term 30-m Landsat
observation (1990 to 2020) for this study, with a range of critical attributes including two
types of classification systems. Within the same study area, our 2020 glacial lakes appear to
be closest to the 2018 dataset produced by Wang et al. (2020), with the highest overlap of
greater than 91% in count at the minimum mapping unit of 5400 $m^2$ or 6 pixels (Table 5).
Wang's dataset (2020) contains many large landslide-dammed lakes that are excluded in our
glacial lake mapping, so their total glacier lake area is greater than ours. The overlapping
rates between Wang's glacial lakes (2020) in 1990 and ours are more than 83% in count.
However, their results show a distinct increase of glacial lakes in number and area between
1990 and 2018 (Wang et al., 2020) whereas our data show a more stable change between
1990 and 2020. One possible reason is that manually delineating glacial lakes twice by
different operators during Wang's lake mapping (2020) exacerbates the errors of mapping.
Another reason is that their data contains landslide-dammed lakes that fluctuate greatly with
time and expanded recently. One example is Attabad Lake (Located at 36°18'22.33"N,
74°49'34.36"E).

**Table 5.** Comparison between our Landsat-based mapping and other third-party Landsat-based glacial lake
datasets in the study area.

| Baseline year (them/us) | Method (them/us) | MMU m² (pixels) | Count (them) | Count (us) | Ratio (%) | Reference |
|---|---|---|---|---|---|---|
| 1990/1990 | Manual/Semi-automated | 5400 (6) | 1720 | 2069 | 83.13 | Wang et al., 2020 |
| 1990/1990 | Automated/Semi-automated | 50000 (55) | 145 | 363 | 39.94 | Shugar et al., 2020 |
| 1990/1990 | Manual/Semi-automated | 4500 (5)* | 622 | 2154 | 28.88 | Zhang et al., 2015 |
| 2000/2000 | Manual/Semi-automated | 4500 (5)* | 724 | 2184 | 33.15 | Zhang et al., 2015 |
| 2000/2000 | Automated/Semi-automated | 50000 (55) | 155 | 361 | 42.94 | Shugar et al., 2020 |
| 2008/2000 | Automated & Manual/Semi-automated | 8100 (9) | 1067 | 1800 | 59.28 | Chen et al., 2021 |
| 2015/2020 | Automated/Semi-automated | 50000 (55) | 148 | 364 | 40.66 | Shugar et al., 2020 |
| 2017/2020 | Automated & Manual/Semi-automated | 8100 (9) | 1063 | 1813 | 58.63 | Chen et al., 2021 |
| 2018/2020 | Manual/Semi-automated | 5400 (6) | 1956 | 2149 | 91.02 | Wang et al., 2020 |

Note: MMU represents the minimum mapping unit that is possible to enable a valid comparison between our product and each
of the third-party datasets. * The MMU in the dataset of Zhang et al. (2015) is 3 pixels, finer than 5 pixels in our product, so
an MMU threshold of 5 pixels was used for this comparison.

The second highest overlapping rate is approximate 59% for 2008 and 58% for 2017 in
count comparing with Chen's data (Chen et al., 2021). Similarly, the overlapping rate between
Shugar's dataset (2020) and ours is lower than 43% in count at the minimum mapping unit of
50000 m². The dataset from Zhang et al. (2015) shows fewer glacial lakes in 1990 and 2000 at
the same MMU of 5 pixels. Our product has more lakes than each of the other 4 products at 9
time periods. By inspecting their dataset, we attributed this anomalous discrepancy to a range
of glacial lakes that were missing due to a lack of thorough cross-check quality assurance
during their lake mapping over a larger study area. And those more glacial lakes show an
improvement of our product in contrast to the previous similar datasets. Our Landsat-derived
glacial lake dataset has been visually cross-checked over three time periods after the step of
threshold-based semi-automated lake mapping and has also been visually validated by
Sentinel-derived glacial lakes. Through this series of quality assurance, we aim at delivering
one of the most reliable multi-decadal glacial lake products for this study area.
Other factors, such as image quality and acquisition dates, mapping methods, and quality
assurance workflow, might also lead to discrepancies between the glacial lake datasets. Despite
such discrepancies, an increasing number of publically-shared datasets benefit potential users
to select the most suitable one for their objectives. Herein, we provide an up-to-date glacial
lake dataset derived from both Landsat and Sentinel-2 observations, which further increased
the availability of glacial lake dataset for water resource and GLOFs risk assessment, predicting
glacier-lake evolutions (Carrivick et al., 2020) in the context of climate change.

6.4 Limitation and updating plan
We would like to acknowledge several limitations of our glacier lake dataset, largely due to
the availability of high-quality satellite images in the study area and inadequate field survey
data (Wang et al., 2020; Chen et al., 2021). First, it is unlikely to collect enough good-quality
images within one calendar year for the entire study area due to the high possibility of cloud

or snow cover. Even though the capacity of repeat observations for Landsat-8 OLI and Sentinel-2 increased (Roy et al., 2014; Williamson et al., 2018; Wulder et al., 2019; Paul et al., 2020), the 2020 glacial lake dataset has to employ images acquired in adjacent years besides 2020. Most images used from Landsat and Sentinel-2 platforms were imaged in autumn, and some images taken between April and July and in November also were employed. Distribution and changes in glacial lakes primarily represent the characteristics between August and October. Glacial lakes evolve with time and space (Nie et al., 2017), and subtle inter- and intra-annual changes (Liu et al., 2020) for each period were ignored. Second, field investigation data are limited due to the low accessibility of the high mountain environment in the study area, which restrained the accuracy in classifying the glacial lake types. Although very high-resolution Google Earth images were utilized to assist in lake-type interpretation, occasional misclassification was unavoidable. We implemented two types of classification systems based on a careful utilization of glacier data, DEM, geomorphological features, and expert knowledge. However, the lack of in situ surveys prohibited a thorough validation of the glacial lake types. Third, the rigorous quality assurance and cross-check after semi-automated lake mapping assures the quality of our lake dataset but are still time and cost-prohibitive. State-of-the-art mapping methods, such as deep learning method (Wu et al., 2020), Google Earth Engine cloud-computing (Chen et al., 2021), and synergy of SAR and optical images (Wangchuk and Bolch, 2020; How et al., 2021), would be used in the future to balance product accuracy and time cost.

The glacial lake dataset will be updated using newly collected Landsat and Sentinel images at a five-year interval or modified according to user feedback. The updated glacial lake dataset will continue to be released freely and publicly on the Mountain Science Data Center sharing platform.

## 7 Data availability

Our glacial lake dataset extracted from Sentinel-2 images in 2020 and Landsat observation between 1990 and 2020 are available online via the Mountain Science Data Center, the Institute of Mountain Hazards and Environment, the Chinese Academy of Sciences at https://doi.org/10.12380/Glaci.msdc.000001 (Lesi et al., 2022). The glacial lake dataset is provided in both ESRI shapefile format (total size of 22.6 MB) and the Geopackage format (version 1.2.1) with a total size of 9.2MB, which can be opened and further processed by open-source geographic information system software such as QGIS.

## 8 Conclusions

Glacial lake inventories of the entire China-Pakistan Economic Corridor in 2020 were provided based on Landsat and Sentinel-2 images using a threshold-based semi-automated mapping method. Both Landsat and Sentinel-2 derived glacial lake dataset show similar characteristics in spatial distribution and the statistics of count and area. By contrast, the glacial lake dataset derived from Sentinel-2 images with a spatial resolution of 10 m has a lower mapping error and more accurate lake boundary than those from 30 m spatial resolution Landsat images whereas Landsat imagery is more suitable to analyze spatial-temporal changes at a longer time scale due to its long-term archived observations at a

consistent 30 m spatial resolution starting from the late 1980s.
Glacial lakes in the study area remain relatively stable with a slight increase in number and
area between 1990 and 2020 according to Landsat observations. Our dataset reveals that 2154
glacial lakes in 1990 covering $85.1 \pm 14.66$ km$^2$ increased to 2234 lakes with a total area of
$86.31 \pm 14.98$ km$^2$. The same mapping method and rigorous workflow of quality assurance
and quality control used in this study reduced the error in multi-temporal changes of glacial
lakes.
Hanshaw's error estimation method for pixel-based lake mapping was improved by
removing repeatedly calculated edge pixels that vary with lake shape. Therefore, the newly
proposed method reduces the estimated value of uncertainty from satellite observations. The
average relative error is ±17.36% for the Landsat-derived dataset and ±8.15% for the product
from Sentinel-2.
Our glacial lake dataset contains a range of critical parameters that maximize their
potential utility for water resource and GLOFs risk evaluation, cryosphere-hydrological, and
glacier-lake evolution projection. The dual classification systems of glacial lake types were
developed and are very likely to attract broader researchers and scientists to use our datasets.
In comparison with other existing glacial lake datasets, our products were created through a
thorough consideration of lake types, cross-checks, and rigorous quality assurance, and will
be updated and released continuously in the Mountain Science Data Center. As such, we
expect that our glacial lake dataset will have significant value to cryospheric-hydrology
research, the assessment of water resources, and glacier-related hazards in the CPEC.
**Appendix.** The appendix related to this article is available online.
**Author contributions.** ML and YN conceived the study, ML, YN and XD performed data
processing and analysis of the glacial lake inventory data, JW contributed to tool
development and mapping methods, and ML and YN wrote the manuscript. All authors
reviewed and edited the manuscript before submission.
**Competing interests.** The authors declare no conflict of interest.

**Acknowledgments.**
We are grateful to the chief editor (ice) Kenneth Mankoff and three anonymous referees for
their constructive comments that greatly help us to improve this manuscript. This study was
supported by the second Tibetan Plateau Scientific Expedition and Research Program (grant
2019QZKK0603), the National Natural Science Foundation of China (Grant Nos. 42171086,
41971153), the International Science & Technology Cooperation Program of China (No.
2018YFE0100100), the Chinese Academy of Sciences "Light of West China" and Natural
Sciences and Engineering Research Council of Canada (Grant No. DG-2020-04207).

## Appendix

## Tutorial for Improved Uncertainty Estimating Method

Hanshaw's equation was originally proposed for pixelated polygons (such as a polygon directly extracted from a remote sensing image), and performed more robustly than manually digitized polygons (where vertices do not necessarily follow the pixel edges). Our improved method also performs better for pixelated polygons. This tutorial is dedicated to helping implement our improved uncertainty estimation method.

**The procedure of uncertainty estimating method (using ArcGIS (© ESRI) for example)**
1. Removing redundant nodes (optional)
We found that a small proportion (~1%) of the pixelated lake polygons (directly extracted from satellite images) have redundant nodes, which affects the value of inner nodes. If no redundant nodes exist, this step can be skipped. Or, we recommend using the "Simplify Polygon" tool in ArcGIS to remove those nodes (Figure A1).
    In the Simplify Polygon panel
- Input your dataset.
- Set the output path and output file name.
- Choose the simplification algorithm. We recommended "POINT_REMOVE".
- Set the tolerance of the simplification algorithm. In this step, we need to ensure that the polygon boundaries remain unchanged after deleting redundant nodes. Generally, a tolerance of 1 meter will suffice, or you can adjust the threshold until your satisfaction.

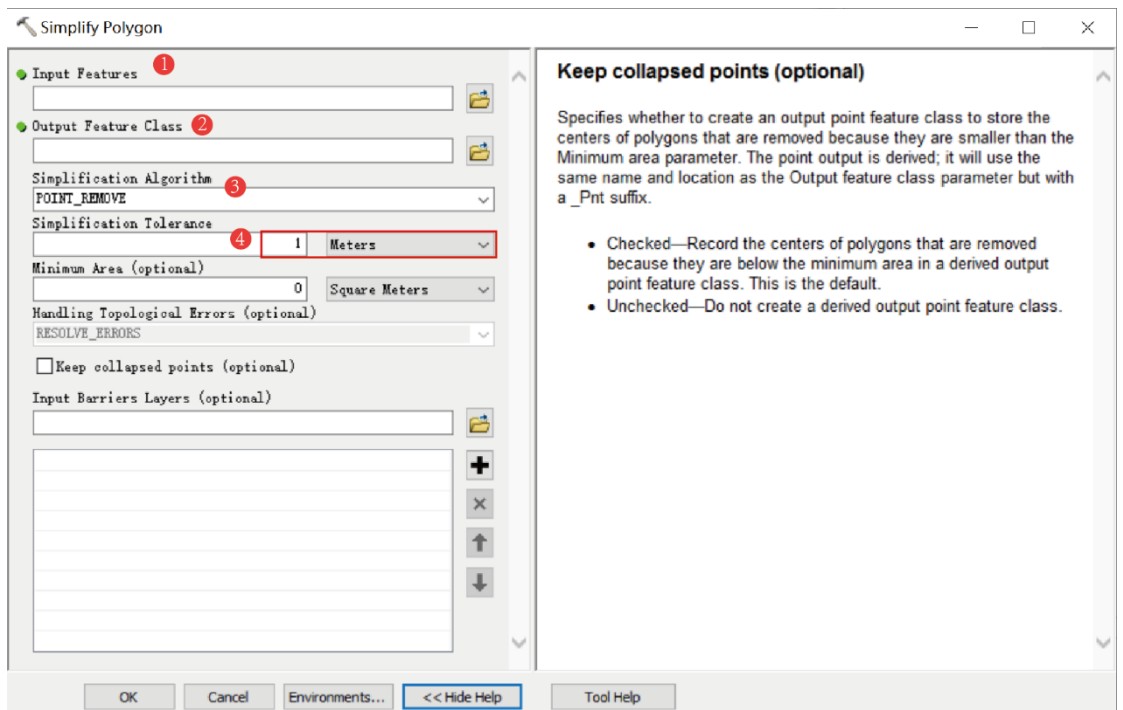

**Figure A1.** Input and option for Simplify Polygon in ArcGIS.

2. Calculating the total number of nodes using ArcGIS (Figure A2):
● Add a new field in the attribute table of the dataset.
● Open Field Calculator.
● Switch the parser to python-mode, and enter the following code "!shape.pointcount!" in
the blue box to calculate the total number of nodes for each glacial lake boundary.

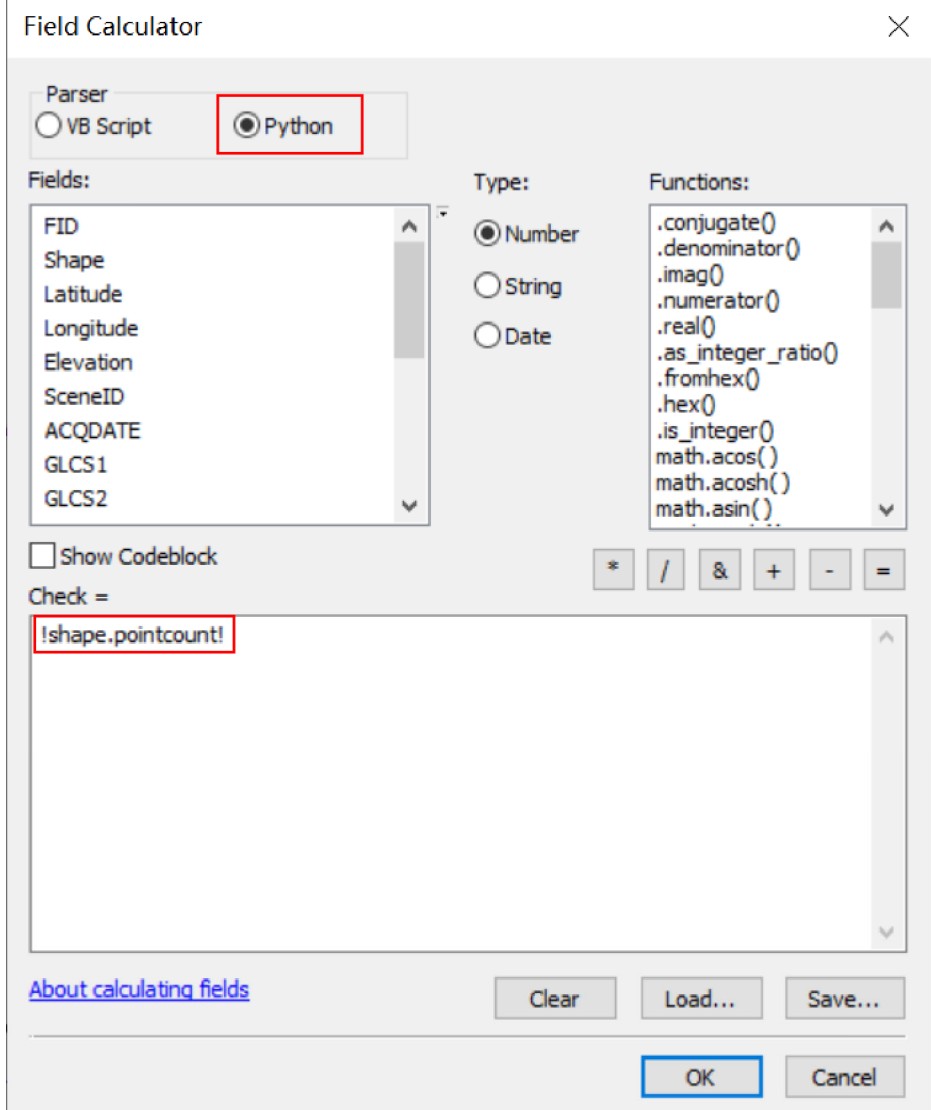

**Figure A2.** Total node calculation in ArcGIS.
3. Calculating the number of inner nodes:
For polygons without islands (Figure A3), use equation 5. An inner node is a polygon vertex
where the interior angle surrounding it is greater than 180 degrees. An outer node is the
opposite of the inner node, where the interior angle is less than 180 degrees. We found that
the outer nodes are usually four more than the inner nodes in our glacial lake dataset. The
total nodes in ArcGIS contain one overlapping node to close the polygon, meaning the
endpoint is also the start point. This extra count was deleted from the calculation (equation

5).

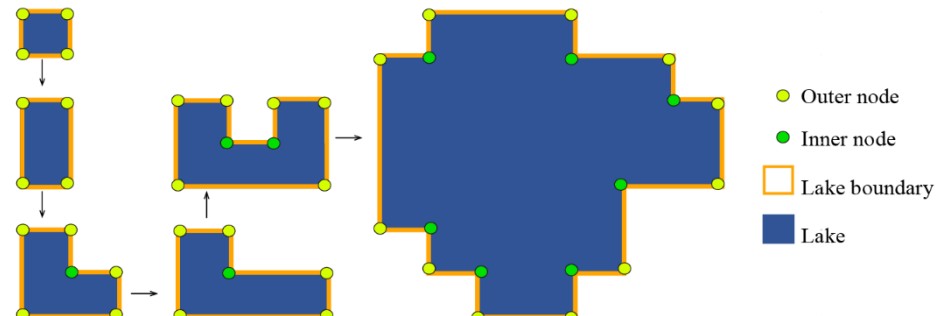

**Figure A3.** Sketch of outer and inner nodes of various glacial lakes without island.

For polygons with islands (Figure A4) use equation 6.

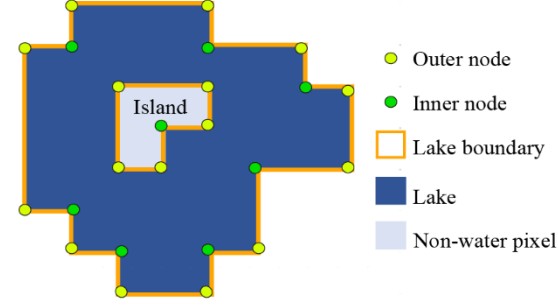

**Figure A4.** Sketch of outer and inner nodes for a glacial lake with an island.

We further specify the steps below to help implement equation 6.

Sept 1: detect the number of islands within each polygon.
● Convert the initial lake polygon to a polyline using the "Feature To Line" tool (Figure
A5).

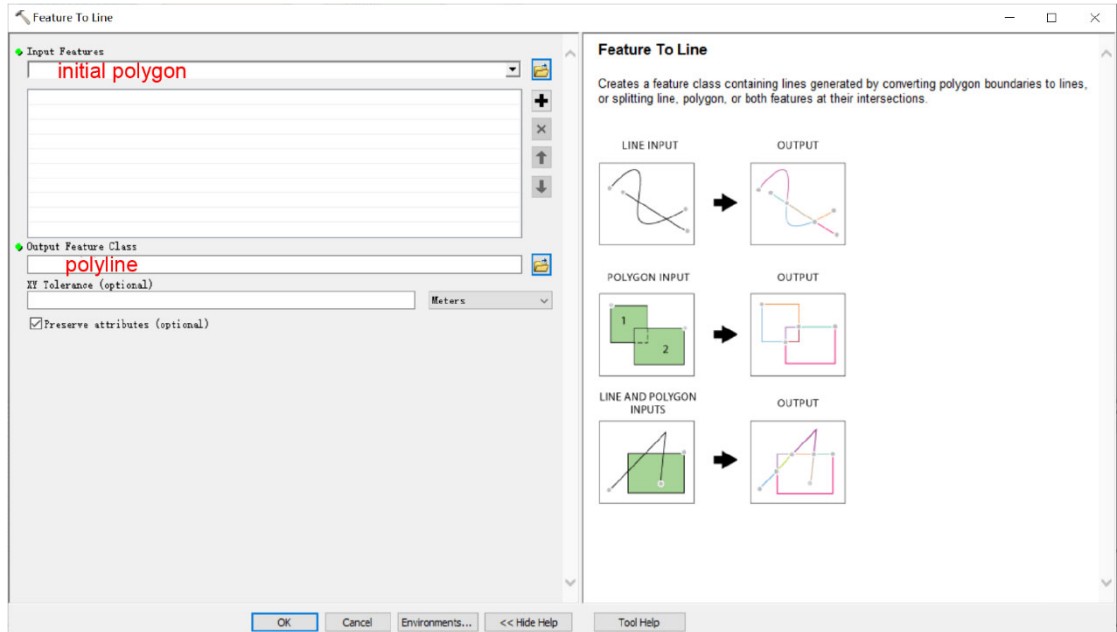


**Figure A5.** Feature To Line tool in ArcGIS



• Convert the polyline to generate a new polygon (Figure A6).

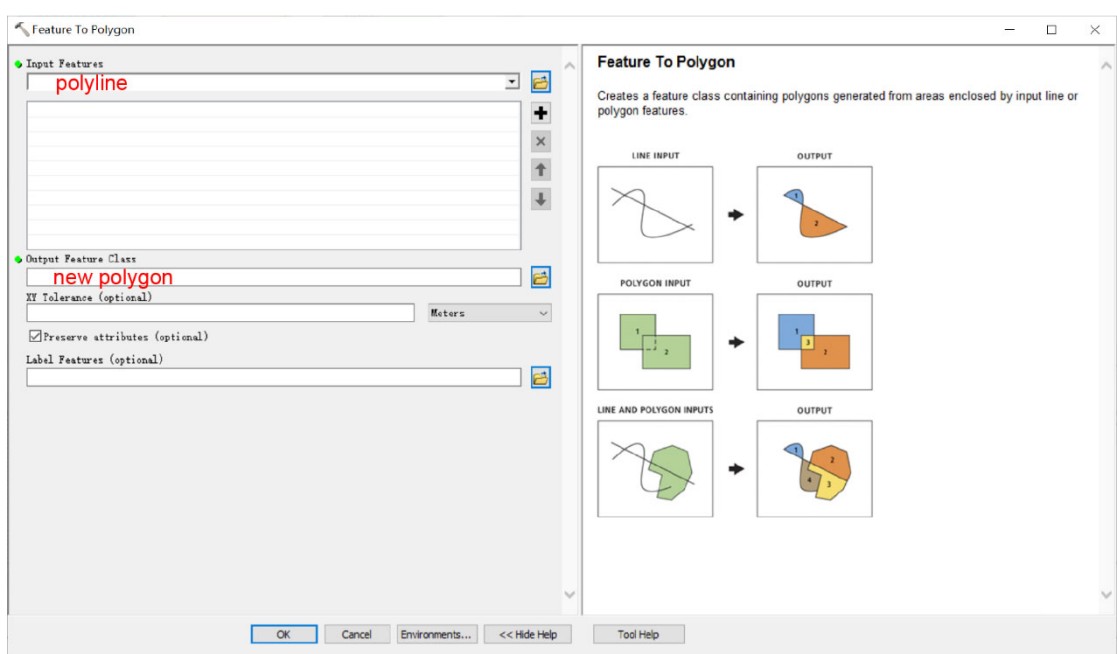


**Figure A6.** Feature To Polygon tool in ArcGIS



• Erase the new polygon by the initial polygon, which outputs the islands. Then we can
count how many islands there are in each lake (Figure A7).

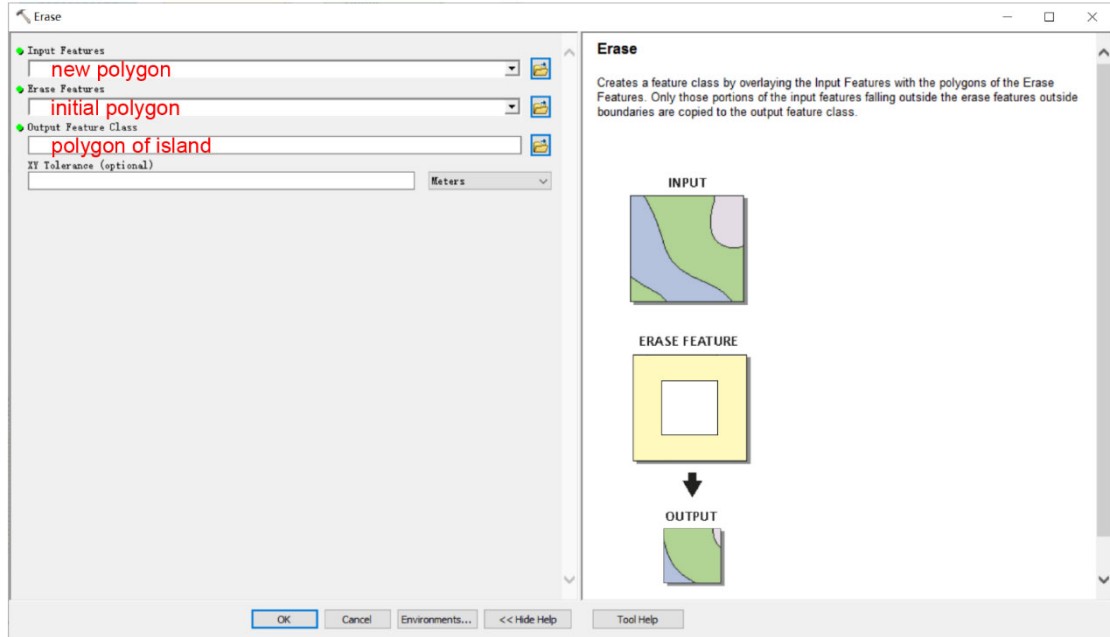

**Figure A7.** Erase tool in ArcGIS.
Step 2: calculate the number of inner nodes for each polygon with an island or islands using
equation 6.
4. Calculating the uncertainty of lake mapping using equation 4.

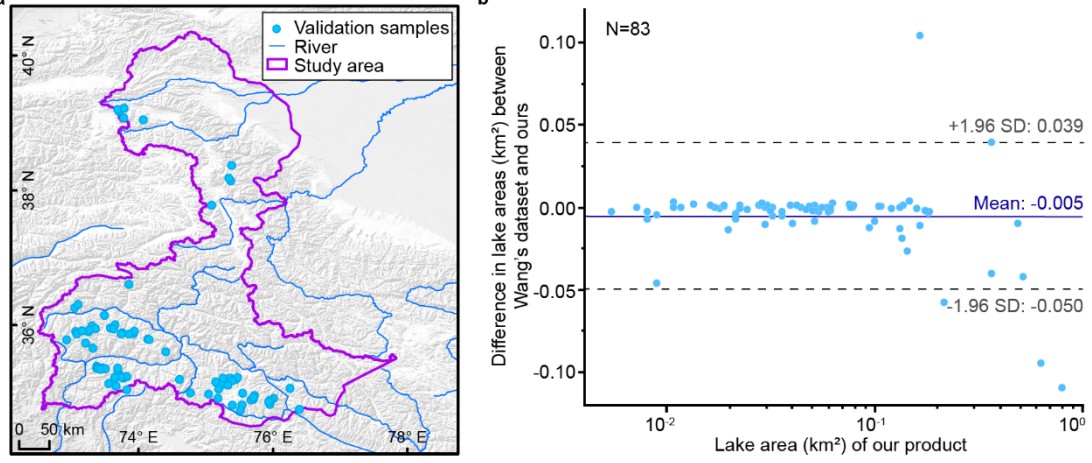

**Figure A8.** Distribution of validation samples (a) and comparison of glacial lakes (b) derived from our
Landsat product in 2020 and Wang's lake data in 2018.

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
