# Peer review of "Landsat- and Sentinel-derived glacial lake dataset in the China-Pakistan Economic Corridor from 1990 to 2020"

_Earth System Science Data, 2021_

## Author Response (AR1)

Earth Syst. Sci. Data Discuss., referee comment RC1 https://doi.org/10.5194/essd-2021-468-RC1, 2022

[Figure]

**Comment on essd-2021-468**

Anonymous Referee #1
* * *
Referee comment on "Landsat and Sentinel-derived glacial lake dataset in the China- Pakistan Economic Corridor from 1990 to 2020" by Muchu Lesi et al., Earth Syst. Sci. Data Discuss., https://doi.org/10.5194/essd-2021-468-RC1, 2022
* * *
**Response to reviewer's comments:**

Note: in the text that follows, reviewer comments appear in black, whilst author responses appear in blue.

Overall comment:

This manuscript is in good shape. But there are two issues that MUST have attention given to them. (i) Terminology..the classes of lakes are not named at all well. I have suggested what to do. (ii) Argument of why we need to detect small lakes.; I am totally unconvinced by the GLOF angle...rather I suggest thinking about the effects of lakes on glaciers and the fact that many of these newly-formed small lakes will become larger with ongoing glacier mass loss. I have offered a coitation to start these thoughts but really a whole paragraph needs adding. Else the abstract needs a complete overhaul too.

[Response] Thank you for your encouragement and constructive comments, which have helped us to improve the quality of this article. We respond to the comments point by point as follows.

Regarding the terminology of lake classification systems, we have revised it as suggested. Now, in the first glacial lake classification system, glacial lakes were classified into four types based on their spatial relationship to upstream glaciers: supraglacial, ice-contact, unconnected-glacier-fed lakes, and non-glacier-fed lakes. In the second glacial lake classification system, glacial lakes were classified into five categories (herein named GLCS2) modified based on Yao's classification system (2018): supraglacial, end-moraine-dammed, lateral-moraine-dammed, glacial-erosion lakes and ice-dammed lakes.

Regarding small lakes, we agree that small lakes have little or no hazardous impact due to their limited water release. The focus of this study is to generate a new glacial lake dataset for the CPEC, using 5 pixels as the mapping threshold for both Landsat and Sentinel images. We had to

map all the glacial lakes, including small ones. If users want to conduct GLOF hazard risk studies, they have the flexibility to set a minimum threshold, such as greater than 0.05 km$^2$, to eliminate smaller lakes from our dataset. In addition, we argue that small lakes, such as supraglacial lakes, play an important role in glacier and lake evolutions, and affect cryosphere-hydrological changes. Thus our dataset has potential to be widely applied in studies on glacial lake-related hazards, glacier-lake interactions and cryospheric hydrology. We have made some changes in the Abstract and main text, and responded to the query of Line32.

We have also cited the recommended references and revised the abstract.

Specific comments:

Line 19: suggest rewording to …'one of a number of flagship projects…'
[Response] Revised as '…one of the flagship projects…'

Line 22, suggest delete 'critical parameters' and state '…parameters X and Y and Z…' (list them out) Are these ALL glacial lakes? Or just ice-marginal ones? Supraglacial? Subglacial? Please specify. Add this specification into your methods.
[Response] Considering the dataset with 17 attributes, it is not suitable to list all of them here. So we decided to list three parameters for example, and revised this sentence to be: 'An up-to-date, high-quality glacial lake dataset with parameters such as lake type, acquisition date and area, which is fundamental to flood risk assessments and predicting glacier-lake evolutions and cryosphere-hydrological changes…'

This study defines a glacial lake as one that formed as a result of modern or ancient glaciation. All glacial lakes in the study area were mapped according to our definition without any distance limit between lakes and glaciers. So it is not just ice-marginal glacial lakes. The dataset includes supraglacial lakes; however, it does not include subglacial lakes that are not detectable from optical satellite sensors. See section 4.1 and 4.3 for detailed description.

Line 24. I suggest to put the resolution(s) after the dataset type. Split sentence into two. One for lakes, one for glaciers, for clarity (because as written it is not clear if OI was for lakes or glaciers or both).
[Response] Following the suggestion, we have revised this sentence to be: 'This dataset includes (1) a glacial lake inventory for the year 2020 at 10 m resolution produced from Sentinel spectral images, and (2) multi-temporal inventories for 1990, 2000, and 2020 produced from 30 m resolution Landsat images.…'

Line 30…is this 5 pixel threshold for both Landsat and Sentinel? Please clarify the thresholds for BOTH datasets.
[Response] Yes, the 5-pixel threshold is for both Landsat and Sentinel images. See 'Landsat derived 2234 glacial lakes in 2020, covering a total area of 86.31 ±14.98 km$^2$ with a minimum mapping **unit of 5 pixels (4500 m$^2$)**, whereas Sentinel derived 7560 glacial lakes in 2020 with a total area of 103.70±8.45 km$^2$ with a minimum mapping unit of **5 pixels (500 m$^2$)**.'

Line 31…I'm not sure this is 'discrepancy', rather simply a result that can be interpreted to be due to many small lakes.
[Response] We agree that the discrepancy cannot be simply interpreted by the differences in total number and area of both Landsat-derived and Sentinel-derived lake dataset. Hence, we performed a comparison between Sentinel-2 and Landsat derived products in section 6.1 and interpreted the discrepancy. It reads:

'Glacial lakes from Landsat and Sentinel images have a high consistency in number and area with overlap rates from 85.7% to 94.26% for all lakes greater than about 0.0045 km$^2$ (Table 5),

implying a good potential for coordinated utility with Landsat archived observation (Figure 11). Lake extents extracted from Landsat and Sentinel images match well for various lake types and sizes (Table 4). The best consistency rate reaches 94% for the glacial lakes between 0.1 km$^2$ and 0.2 km$^2$. The difference in area of glacial lakes extracted from Landsat and Sentinel images generally lies within the uncertainty ranges.'

Also, 'Spatial resolution of satellite images plays a primary role in the discrepancies in count and area of glacial lakes extracted from Landsat (30 m) and Sentinel (10 m) observations. Due to a finer spatial resolution, Sentinel images can extract more glacial lakes and more accurate extents than those from Landsat images. We set the same 5 pixels as the minimum mapping unit for both Landsat and Sentinel images, which corresponds to a minimum area of 0.0045 km$^2$ and 0.0005 km$^2$, respectively. The minimum mapping area results in generating nearly 5000 more lakes from Sentinel images than from Landsat images, causing the greatest discrepancy in number of the two glacial lake products (Table 5), such as Figure 12a.'

Line 32. Are (very) small lakes important? For hazards/GLOFs? Why? I think you need to discuss/show this….in the main text of the manuscript as well as here in the abstract…else the whole premise of your work is not represented/defended/argued (?!).
[Response] We agree that small lakes have little or no hazardous impact due to their limited water release. The focus of this study is to generate a new glacial lake dataset based on Landsat and Sentinel images in the CPEC, using 5 pixels as mapping threshold for both Landsat and Sentinel images. We had to map all the glacial lakes, including small ones. If users want to conduct GLOF hazard risk studies, they can set a minimum threshold, such as greater than 0.05 km$^2$, to select a subset of lakes from our dataset to eliminate the impact of small lakes. Small lakes, such as supraglacial lakes play an important role in glacier and lake evolutions, and affect cryosphere-hydrological changes. Thus our dataset has potential to be widely applied in studies on glacial lake-related hazards, glacier-lake interactions and cryospheric hydrology. We have made some changes in the Abstract and main text. It now reads:

'An up-to-date high-quality glacial lake dataset with parameters such as lake type, acquisition date and area, which is fundamental to flood risk assessments and predicting glacier-lake evolutions and cryosphere-hydrological changes…'

'This comprehensive glacial lake dataset has potential to be widely applied in studies on glacial lake-related hazards, glacier-lake interactions and cryospheric hydrology…'

'Small lakes such as supraglacial lakes play an important role in understanding meltwater runoff and supraglacial drainage systems (Miles et al., 2018; Liu and Mayer, 2015). Our dataset can be used not only for GLOFs evaluation, but also for glacial lake evolution simulation and glacio-hydrological prediction. '

Line 36…would be more useful to state the types of lakes please. And state the two classifications systems please. Be explicit (!). what is the improved equation?! Name it!
[Response] The two classifications systems contain a total of nine types of glacial lakes, so specifying all types in the abstract will take up too much space. To keep the abstract concise and present improved equation, we changed this sentence to: 'A range of critical attributes have been generated in the dataset, including lake types and mapping uncertainty estimated by an improved Hanshaw's equation'

Line 37. Potentials is not plural. Remove the 's'.
[Response] We have changed this sentence to 'This comprehensive glacial lake dataset has potential to be widely applied in studies on glacial lake-related hazards, glacier-lake interactions and cryospheric hydrology…'

Line 48. You really must have to cite Carrivick and Tweed (2016)
https://www.sciencedirect.com/science/article/pii/S0921818116301023?via%3Dihub please!
here. Furthermore, if you read that paper, the size of lakes producing hazardous GLOFs is
reported. Small lakes (like the ones detected by your sentinel analysis v landsat) are not
hazardous (!).
[Response] We agree and cited the reference from Carrivick and Tweed (2016). It now reads:
'…glaciers and impacting downstream ecosystem services, agriculture, hydropower and other
socioeconomic values (Nie et al., 2021; Carrivick and Tweed, 2016).'

About small lakes, we have responded to the previous inquiries.

Line 97. Please explain 'type' is this glacier terminus environment? Is it lake dam type? Is it
lake position (supraglacial or ice-marginal for example?).
[Response] As suggested, we have revised this to be: 'Dam type classification of glacial lakes
provides a crucial attribute for glacier-lake interactions and risk assessment…'.

Line 173. A glacial lake is one that receives meltwater from a glacier. Of these most are
proglacial (beyond the glacier) and can be attached (ice-marginal or ice-contact) or detached
from the edge of the glacier. PLEASE correct this terminology. Then say what you do (which
means you need to evaluate what sort of lakes you are actually analysing!).
[Response] We agree to divide proglacial lakes into ice-contact and unconnected-glacier-fed
(detached) lakes in this study. We revised the classification system of glacial lakes. See the
section 4.3.

We consider a glacial lake as one that formed as a result of modern or ancient glaciation. In this
study, all glacial lakes were mapped according to this definition and are attributed using the two
classification systems.

Line 186 'without any distance limit'…oh come on there must have been some limit?! The
catchment or study area boundary at least?! Please evaluate what you have done and report it
carefully.
[Response] Thank you for this comment. Our dataset is indeed limited by the study area
boundary. To avoid any misunderstanding, this sentence was changed to 'All glacial lakes in the
study area were mapped according to our definition regardless of the distance to glaciers.'

Line 207 this info. on mapping units needs to be accurately represented in the abstract.
[Response] We agree, and have revised this sentence to be: 'The results show that Landsat
derived 2234 glacial lakes in 2020, covering a total area of 86.31 ±14.98 km$^2$ with a minimum
mapping unit of 5 pixels (4500 m$^2$), whereas Sentinel derived 7560 glacial lakes in 2020 with a
total area of 103.70 ±8.45 km$^2$ with a minimum mapping unit of 5 pixels (500 m$^2$).'

Line 233. This spatial relationship needs to be explicitly named above in the manuscript where I
have already queried it. I dislike this classification. See Carrivick and Tweed (2013)
https://www.sciencedirect.com/science/article/pii/S027737911300293X for definition of
proglacial lakes (my comment for line 173). Supraglacial is a distinct group so that is OK.
Proglacial and unconnected are the same/overlap…you need 'ice-marginal' and 'other
proglacial' I think, then 'other lakes' as your classes/types.
[Response] Thank you for this valuable suggestion. As we responded earlier, we agree to divide
proglacial lakes into ice-contact and unconnected-glacier-fed (detached) lakes. We have revised
this consistently throughout the main text, figures, tables and attribute of our glacial lake dataset.

This sentence was changed to '…glacial lakes were classified into four types based on their
spatial relationship to upstream glaciers: supraglacial, ice-contact, unconnected-glacier-fed lakes,
and non-glacier-fed lakes according to Gardelle et al. (2011) and Carrivick et al. (2013).'

Line 238. The terminology again is wrong here and confusing because mixes position and dam type. See Carrivick and Tweed (2013) https://www.sciencedirect.com/science/article/pii/S027737911300293X . You should have supraglacial, terminus moraine-dammed, lateral moraine dammed, ice-dammed and bedrock-dammed I suggest.

[Response] Considering the formation mechanism and dam properties of glacial lakes, the second glacial lake classification system was established via modifying Yao's classification system (2018). According to your suggestion, we have revised the terminology of the classification system to ice-dammed, end-moraine-dammed, lateral-moraine-dammed and supraglacial lakes. Glacial-erosion lakes contain both bedrock-dominated dam and top-moraine-mixed dam, so we prefer to use glacial-erosion lakes instead of bedrock-dammed lakes. It now reads:

'Alternatively, combining the formation mechanism of glacial lakes and the properties of natural dam features, glacial lakes were classified into five categories (herein named GLCS2) modified from Yao's classification system (2018): supraglacial, end-moraine-dammed, lateral-moraine-dammed, glacial-erosion lakes and ice-dammed lakes.'

Line 318 to 326. I suggest to compare to (and cite) Carrivick and Quincey (2014) who also consider uncertainty v lake area. https://www.sciencedirect.com/science/article/pii/S092181811400054X?via%3Dihub

[Response] We cited the reference from Carrivick and Quincey (2014). It now reads 'Lake perimeter and displacement error are widely used to estimate the uncertainty of glacier and lake mapping from satellite observation (Carrivick and Quincey, 2014; Hanshaw and Bookhagen, 2014; Wang et al., 2020).'

The difference in uncertainty estimation between Carrivick's and Hanshaw's methods is that Carrivick assumes an uncertainty of ±1 pixel, while Hanshaw assumes an uncertainty of ±0.5 pixels and counts the number of edge pixels. In this study, we discovered and solved the problem of repeatedly calculated edge pixels. Considering that the mean lake size in the study area is smaller than that in the Greenland, we prefer to choose the improved Hanshaw's equation to estimate the mapping uncertainty.

Line 453…so do we need Sentinel images for lake mapping?? If Landsat is doing a good job v sentinel (detection as well as accuracy) then why do we need the extra resolution? What importance do the numerous small lakes have? They are not important volumetrically? Are they important for hazards/GLOFs? (I don't think so!). I really think the 'promoted capacity of GLOF risk assessment' (line 543) needs further elaboration.

[Response] We believe Sentinel images do offer their unique benefits in mapping glacier lakes, owing to their finer spatial resolution, increasing capacity of revisit observation and accurately depicting lake boundaries with a lower uncertainty. We further clarified these as: 'Due to a finer spatial resolution, Sentinel images can extract more glacial lakes and more accurate extents than those from Landsat images….Meanwhile, Sentinel images are able to depict boundaries of glacial lake with a lower uncertainty (Figure 12b-d). For example, some small islands and narrow channels (Figure 12b and c) were mapped from Sentinel imagery that were unable to be detected in Landsat imagery.'

Regarding small lakes, we have responded to a similar query earlier:
'Small lakes such as supraglacial lakes play an important role in understanding meltwater runoff and supraglacial drainage systems (Miles et al., 2018; Liu and Mayer, 2015). Our dataset can be used not only for GLOFs evaluation, but also for glacial lake evolution simulation and glacio-hydrological prediction. '

About 'promoted capacity of GLOF risk assessment' (line 543), we revised this sentence to be

‘Herein, we provide an up-to-date glacial lake dataset derived from both Landsat and Sentinel observations, which further increased the availability of glacial lake datasets for GLOFs risk assessment, predicting glacier evolutions and understanding cryosphere-hydrological changes in the context of climate change.’

In contrast, I think a utility of your dataset and indeed your sentinel-based detection of many small lakes is that those small lakes could be the onset of fast-developing proglacial landscapes…and they will likely grow as glaciers diminish further and affect glacier dynamics (see Carrivick et al., 2020 for example https://www.frontiersin.org/articles/10.3389/feart.2020.577068/full )
[Response] Thank you for your affirmation and encouragement. The Sentinel-derived lake dataset has a wider potential than Landsat-derived dataset to be used in studies on proglacial landscape change and glacier dynamic assessment. The recommended reference is important and cited in the main text.

studies on glacial lake-related hazards, glacier-lake interactions and cryospheric hydrology. Considering freshwater transfer and storage in the region is part of the study of cryospheric hydrology.

We put the definition of glacial lake in the section 4 Glacial lake inventory methods. This section is composed of 4.1 Definition of glacial lakes, 4.2 Interactive lake mapping, 4.3 Classification of glacial lakes, 4.4 Attributes of glacial lake data and 4.5 Improved uncertainty estimating method.

As suggested, we add one paragraph in the discussion to analyze active glacial lakes related to GLOFs. It now reads:
'The high consistency of Sentinel-2 and Landsat derived glacial lake products in 2020 assures the value of our lake dataset. Taking the usage in assessing GLOFs as an example, we set 0.05 km$^2$ as the area threshold to select object lakes, including ice-contact lakes and ice-dammed lakes that are the most active lakes and source lakes of GLOFs in the CPEC (Nie et al., 2021). A total of 24 and 29 ice-contact lakes were selected from Landsat and Sentinel-derived products, respectively. Among them, there were 4 ice-dammed lakes from the Landsat-derived product and 5 from the Sentinel-derived product. These selected lakes can be used for GLOFs hazard evaluation. Because of the high consistency between our Landsat and Sentinel-based mappings, users may have the flexibility to customize the lake size criteria to facilitate their specific purposes.
'

Regarding to the comparison to other glacial lake datasets, it is arduous to compare discrepancies between glacial lake datasets in the classification methodologies due to their low overlap. The highest overlapping rate is only 74% in count with Wang's data (2020), followed by Chen' highest overlapping rate of 45% in count. Zhang's and Shugar's dataset do not include lake type attribute. Hence, we can not make a comparison in the classification methodologies.

**3. Broader overview of remote sensing classification methods**
Optical classification methods are solely focused on in the introduction section of the manuscript (L86-103), which falsely represents them as the sole classification method readily used in remote sensing. I would like to see the overview include other remote sensing classification methods, namely SAR backscatter classification, but also other alternative approaches such as from hydrological sink analysis and from land surface temperature.
[Response] We added other classification methods besides optimal remote sensing. It new reads:
'Backscatter images from Synthetic Aperture Radar (SAR) (How et al., 2021; Wangchuk and Bolch, 2020) were used to remove the impact of cloud cover for lake mapping. Besides, other approaches such as hydrological sink detection using DEM (How et al., 2021) and land surface temperature-based detection method (Zhao et al., 2020) were also used for lake inventories. Different classification methods impact the results of lake mapping and monitoring.'

I am not sure if there are any studies in this region where alternative classification methods are used to detect water bodies; but if there are any then I think they would be a great addition to the dataset comparison section to serve as an inter-comparison of methodologies beyond alike optical classification approaches.
[Response] To our best knowledge, glacial lake dataset produced based on SAR backscatter classification or hydrological sink analysis is not available in the study area. If available in the future, we are glad to make such a comparison.

**Specific comments**

L41-66: I think this a detailed and concise overview of the importance of glacial lakes and GLOFs in a regional context. However, I think a global perspective is needed to thoroughly

illustrate the significance of this study - especially if you are referring to global studies of glacial lakes, such as Shugar et al. (2021). Please include a sentence or two near the beginning about glacial lakes and GLOFs globally (i.e. importance, general trends etc.)
[Response] As suggested, we have added a sentence herein 'Global glacial lake number and total area both increased between 1990 and 2018 in response to glacier retreat and climate change (Shugar et al., 2020), which inevitably affected the risk of GLOFs.'

L67-85: You largely focus on remote sensing efforts in HMA regional studies, but there are also references to papers from other regions such as Greenland and the Alps. Either open up this section as an overview of remote sensing studies from all regions, or keep it refined to the HMA region. There have been many regional studies that have been published recently (e.g. Alaska, Rick et al., 2022; Greenland, How et al., 2021), not just in HMA, so I would recommend widening this section to outline the methods in a general context, rather than focusing on HMA.
[Response] Thank you and we have cited the suggested recent publications for other regions. It now reads:
'…the Alaska (Rick et al., 2022), the Greenland (How et al., 2021)…'

L92: What exactly do you mean by object-oriented classification here? This term is generally used in programming rather than in reference to a classification approach. Please change this, or clarify what is meant here; preferably with a more suitable term.
[Response] We deleted 'object-oriented classification'.

L117-119: Are these sub-basins divided by catchments and/or watershed? What determines these sub-basins?
[Response] Yes, these sub-basins are divided by catchment based on major tributary rivers and DEM data.

L132-170: Great outline of data sources.
[Response] Thank you for your positive comment.

L178: Why are landslide-dammed lakes irrelevant to glaciation? Can some glacial lakes also be landslide-dammed lakes?
[Response] In this study, we accept the definition of a glacial lake as one that formed as a result of modern or ancient glaciation. Landslide-dammed lakes formed behind landslides, and have little connection with glaciation. Landslide-dammed lakes vary greatly with time and differ from glacial lakes, hence being exceeded in our dataset.

In a particular situation, glacial lakes are also dammed by landslides, someone may define those lakes as landslide-dammed lakes. Our study focuses on all glacial lakes formed as a result of glaciation.

L199: Change 'the method automatically generated the histogram...' to 'the method calculated the histogram...'
[Response] Revised as 'Specifically, the method calculated the NDWI histogram based on the pixels with each user-defined and manually-drawn region of interest.'

L201: Change 'interactively' to 'manually'. In reference to this comment and the last, I think it needs to be clear in the methodology how this approach is 'semi-automated'.
[Response] We think 'interactively' is more suitable than 'manually' to depict the process of lake mapping. We needed to switch the screening NDWI and original image to determine an optimal threshold, and this is an interactive process. In the process of interactive lake mapping, manual inputs refer to drawing user-defined region of interest (ROI) and tuning the NDWI threshold in each ROI, whereas calculating the histogram of NDWI and converting raster lake extent to vector polygon are automated. To avoid the misunderstanding, we define the method as a humaninteractive and automated lake mapping method and made some revision. It now reads:

'Specifically, the method calculated the NDWI histogram based on the pixels with each user-defined and manually-drawn region of interest. The NDWI threshold that separates lake surface from land was interactively determined by screening the NDWI histogram against the lake region in the imagery (Nie et al., 2020; Wang et al., 2014). This way, the determined NDWI threshold can be well-tuned to adapt various spectral conditions of the studied glacier lakes. The raster lake extents segmented by the thresholds were then automatically converted to vector polygons.'

L224-228: False classifications from cloud and topographic shadows can be eliminated with cloud and terrain masking, which are well-established remote sensing methods in land classification. Why did you choose not to include this in the automated component of your workflow?
[Response] We selected high-quality images to map glacial lakes for each time period. However, false lake extents resulting from cloud or snow cover, lake ice, and topographic shadows are unavoidable but limited. Then, we removed those false lakes and again mapped the lakes using our lake mapping method according to alternative images acquired in adjacent years. This method meets the needs of lake mapping. Incorporating cloud and terrain masking in the automated process is an excellent suggestion, and we will consider this in the future research.

Table 1: The characteristics of a proglacial lake should specify that these lakes share a boundary with the ice margin, according to your definition - 'shared boundary' is a better description than 'connected with glaciers' as this could be interpreted as hydrologically connected instead of physically adjacent.
[Response] Replaced 'connected with glaciers' with 'shared boundary with glaciers '.

Table 2: There must be occurrences where a lake's formation and/or dam material properties are ambiguous (especially in relation to GLCS2), even from Google Earth imagery. I see in the dataset that there are no instances where a lake's classification is determined as uncertain; even though you state later on that occassional misclassifications are inevitable (L561). In such instances of ambiguous lake types, how do you decide the classification?
[Response] Yes, some dam material properties are ambiguous from satellite observations. This is a challenge for GLCS2. Differentiating moraine-dammed and glacial-erosion lakes is challenging due to unclear moraine dam or bedrock superimposed by top moraine. To differentiate those dam types, we considered auxiliary factors that help classify lake dam types, such as location, surface slope, roughness and shape of the glacial lakes. We established the classification system of lake types and collected typical samples for each lake type to train our operators at the beginning of the classification. We then used our expert knowledge to classify all lakes with a combination of glacier data, DEM, geomorphological features. When indeterminate lake types emerged, we used group discussions to attribute the type. All these steps help us improve the quality of lake datasets that are more useful to users.

We proposed the limitation and updating plan in the main text:
'Although very high-resolution Google Earth images were utilized to assist in lake type interpretation, occasional misclassification was inevitable. We implemented two types of classification systems based on a careful utilization of glacier data, DEM, geomorphological features and expert knowledge. However, the lack of in situ survey prohibited a thorough validation of the glacial lake types.'

'The glacial lake dataset will be updated using newly collected Landsat and Sentinel images at a five-year interval or modified according to user feedbacks.'

L272-273: Please provide references to studies that use lake perimeter and displacement error to estimate uncertainty.

[Response] We added the citations, and it now reads:
'Lake perimeter and displacement error are widely used to estimate the uncertainty of glacier and lake mapping from satellite observation (Carrivick and Quincey, 2014; Hanshaw and Bookhagen, 2014; Wang et al., 2020).'

L270-295: Repetition with the corresponding uncertainty estimation section in the supplementary materials. I would suggest refining this section in the main manuscript, and keeping the full description for the supplementary materials.
[Response] We removed the duplicate section on uncertainty estimation in the supplementary material, and moved Tutorial for Improved Uncertainty Estimating Method at the end of the main text as an appendix. We want editors to approve these changes.

L294 and L305: Change figure names from Figure S3a/S3b to Figure 3a/3b, unless you would rather move them to the supplementary materials.
[Response] Revised as "Figure 3a" and "Figure 3b".

Figure 4: This is a great figure. Please add labelling to the figure to indicate that one set of graphs is from Landsat and the other from Sentinel - you only understand this once you have read to the end of the caption, and it needs to be signposted earlier.
[Response] We have added Landsat and Sentinel at the upper left corner of the figure 4 to differentiate.

L339: 'proglacier' >> 'proglacial'
[Response] According to suggestions from Reviewer #1, we replaced 'proglacial' with 'ice-contact' and corrected the typo.

Figure 5: The four maps are somewhat repetitive and it is difficult to see differences between the Sentinel and Landsat lake sizes/abundance from this. I would suggest changing this figure to have an overview map on the left showing all detected lakes from both methods, and a series of inset maps to the right displaying a closer look at certain regions of interest; divided into Sentinel and Landsat lakes. Also, maybe change the outline colour of the lake points to a darker shade, as it is hard to identify the lake points in the current figure.
[Response] The figure 5 aims to describe the distribution of glacial lakes in 2020 extracted from Landsat and Sentinel, and all lakes are classified by GLCS1 and GLCS2. From this point of view, it is not repetitive. For clarity, we added 'Landsat' for 'panels a and b', and 'Sentinel' for 'panels c and d'. Meanwhile, differences in glacial lakes from Landsat and Sentinel can be clearly seen compared to "Panels a and c" using GLCS1 and "Panels b and d" using GLCS2. A closer look at certain regions of interest is showed in figure 11 and 12. That is why we designed this Figure.

As suggested, we set the outline of lake points to black with a thicker size in order to better differentiate the lake points.

L352: This is a hanging line, and I am not sure which panels and sub-graphs are being referred to here. Does this belong somewhere else or is this a fault with the journal formatting?
[Response] We corrected this typo and ensure all captions are complete in the main text.

Figure 7 and 8: These graphs are very effective at showing changes in lakes - a refreshing take on presenting this type of dataset.
[Response] Thank you very much.

L382: '...while the area grew by a less extent (1.21 km$^2$ or 1.42%).' >> '...while the area grew by 1.21 km$^2$ (or 1.42%).'

[Response] Revised as 'while the area grew by 1.21 km$^2$ (or 1.42%).'

L408: '...including being stable for Shingo...' >> '...including a stable trend for Shingo...'
[Response] Revised as '...including a stable trend for Shingo...'

L411-412: 'The total numbers of Kashgar and Hunza basins decreased...' >> 'The total number of lakes in Kashgar and Hunza basins decreased...'
[Response] Revised as 'The total number of lakes in Kashgar and Hunza basins decreased...'

L426-27 and Table 5: Can you include some statistics on the link between lake size and % overlap between the Sentinel-2 and Landsat counts? - this would help gauge how much spatial resolution (differentiated from image acquisition) affects lake classification in this study.
[Response] Thanks for this suggestion. We tried but did not find any significant statistics between lake sizes and overlapping rates. Impact of spatial resolutions on classification accuracy of lake types is very interesting. We hope to conduct such studies in the future, however this is beyond the goal of this study.

L441-443: Are these lakes persistently large or just at a particular time step? Do you have evidence as to why they are disproportionally large?
[Response] These lakes are persistently large, and we deleted 'disproportionally' to avoid any misunderstandings.

L481-485: Can you state here the number of instances where overlapping acquisitions were acquired?
[Response] Revised as '…is relatively low (only 7 scenes of Sentinel images or 112 glacial lakes in 2020)…'

L492: 'approximate' >> 'approximately'
[Response] Changed to 'approximately'.

L503-504: Are there any studies that present glacial lake datasets derived from Sentinel-2? If so, please reference them here. If not, then change this to state that there are no comparable datasets, rather than a scarce number of datasets.
[Response] Agreed, it new reads:
'Regional glacial lake datasets using Sentinel images are scarce. Lack of Sentinel-derived glacial lake datasets in the study area makes it impossible to compare.'

L515-521: I think, similar to your suggestion regarding landslide-dammed lakes, a likely answer is that Wang et al. focus more on glacier-connected lakes, given that they adopt a 10 km buffer to filter out unconnected lakes. And therefore they identify an increasing trend, possibly reflective of a subset of your lake types. Could you subset your lake dataset to match the lakes identified by Wang et al., and examine whether you also see this increasing trend evident in your subsetted dataset? (And perhaps also include landslide-dammed lake for the purpose of this comparison?)
[Response] Both Wang' study and ours are regional-scale glacial lake inventories and do not only focus on active lakes. Regarding discrepancy, all glacial lakes in the study area were mapped according to our definition regardless of the proximity to glaciers whilst Wang's data was filtered by a 10 km buffer zone from glaciers. That is the reason that we mapped more lakes than Wang's for each time period. Our lakes dataset maps all glacial lakes in the study area for the first time, making up for missing lakes in other datasets. The missing lakes are mostly non-glacier-fed lakes that remain relatively stable in the past decades (Figure 7), having little impact on changing trend in glacial lakes.

Our study demonstrates that end-moraine-dammed lakes increased by 2.48 km$^2$ and contributed

most of the glacier lake area expansion, whereas supraglacial, ice-dammed and lateral-moraine-dammed lakes decreased slightly in both number and area. This trend is consistent with the slightly negative mass balance of glaciers in the study area (covering Pamir, Karakoram etc.). Based on the analysis of Wang's data, we find that newly-emerged and expanded landslide-dammed lakes contributed most to the increase in lake area, and manually delineating glacial lakes twice by different operators exacerbated the errors of mapping. These reasons result in an increasing trend in their study. We have proposed the reason in the main text.

Landslide-dammed lakes fluctuate greatly with time and expanded recently in the study area, differing from typical glacial lakes. According to our definition on glacial lakes, Landslide-dammed lakes are excluded. Our dataset shows a long-term regional changing trend of all glacial lakes. A comparison in the condition of including landslide-dammed lakes is beyond the goal of our present study and seems unnecessary. Thanks for your support and understanding.

L530: "Zhang's dataset..." >> "The dataset from Zhang et al. (2015) ..."
[Response] Revised as 'The dataset from Zhang et al. (2015)...'

L531-536: I am unsure how this study and Zhang et al. could have discrepancies in image availability when both studies are classifying lakes from the same satellite image collection (Landsat). Some clarification is needed here to demonstrate how your dataset could classify these lakes when Zhang et al. could not.
[Response] We revised this sentence. It now reads:
'…we attributed this anomalous discrepancy to a range of glacial lakes that were missing due to lack of thorough cross-check quality assurance and the limit of a 10-km buffer zone from glaciers during their manual delineation.'

L538-544: Discrepancies in glacial lake datasets can be because of minimum lake size, classification method (i.e. not just optical), image acquisition and post-filtering. However, if the purpose of this dataset is to 'further promote the capacity of GLOF risk assessment and predicting glacier evolutions' then I am unsure why there is no spatial filter (relative to ice margin position) adopted to remove lakes that are unconnected to the glacial system. I think the focus of this study needs to be shifted (as stated earlier in major comments), and further analysis needs to be presented that demonstrates changes in GLOF and glacier-fed lakes specifically (i.e. filtered by lake type and size) - see major comments for full details.
[Response] The focus of this study is to provide a complete glacial lake dataset based on Landsat and Sentinel images, which has potential to be widely applied in studies on glacial lake-related hazards, glacier-lake interactions and cryospheric hydrology. We made some modifications to be more explicit about the goals of this study and the importance of our dataset. The sentence was changed to:
'…we provide an up-to-date glacial lake dataset derived from both Landsat and Sentinel observations, which further increased the availability of glacial lake datasets for GLOFs risk assessment, predicting glacier evolutions (Carrivick et al., 2020) cryosphere-hydrological changes…'

L550: 'Even though an capacity of repetitive observations...' >> 'Even though the capacity of repeat observations...'
[Response] Revised as 'Even though the capacity of repeat observations...'

L557: '...inter- and intra-annual changes (Liu et al., 2020) in glacial lake dataset of each time period...' >> '...inter- and intra-annual changes (Lie et al., 2020) for each time period...'
[Response] Revised as '...inter- and intra-annual changes (Liu et al., 2020) for each time period...'

L545-564: This is a valuable section to include in the study. The temporal range of these

datasets and limited image availability (especially in formative years) will not adequately capture the dynamic nature of draining glacial lakes; and therefore such datasets serve as a gauge of long-term, regional trends rather than individual lake change.
[Response] We are thankful for your support and understanding.

L565-575: It is great to hear that this work will be continued, and new time steps will be included in the dataset in the future.
[Response] Thanks for your encouragement.

L584: '...spatial-temporal changes at longer time scale...' >> '...spatial-temporal changes at a longer time scale...'
[Response] Revised as ' ...spatial-temporal changes at a longer time scale... '.

L584: 'observation' >> 'observations'
[Response] Revised as ' observations '.

L585: 'started' >> 'starting'
[Response] Revised as ' starting '.

L595-596: See previous comment from L538-544 regarding this statement.
[Response] As responded to earlier questions. This sentence was changed to: '…maximize their potential utility for GLOFs risk evaluation, cryosphere-hydrological and glacier-lake evolution projection.'

L602: 'values for cryospheric-hydrology research, assessment of...' >> 'value to cryospheric-hydrology research, the assessment of...'
[Response] Revised as 'value to cryospheric-hydrology research, the assessment of... '

---

## Referee Report (RR1)

**Review of ESSD-2021-468: "Landsat and Sentinel-derived glacial lake dataset in the China-Pakistan Economic Corridor from 1990 to 2020"**

**Summary**

This is my first review of the manuscript by Lesi et al., which presents a new glacial lake dataset, derived through remote sensing and semi-automated classification, over the CPEC area. The dataset is novel, well described and presented, and the manuscript is generally clear and well written. However, I have two concerns; the first relating to the content vs. scope of the article type, and the second to the discussion of errors:

1. My understanding of the scope of ESSD is that a novel dataset is presented with minimal analysis (at least none that goes beyond assessing the accuracy of the dataset) and no interpretation. While I notice that recently published data papers in ESSD vary in the amount of analysis/interpretation they contain, my view is that here, there is analysis and interpretation of the dataset that goes beyond the scope of a data description paper (e.g., a large part of the Results, and part of the Discussion – detailed below). To me, the manuscript is presented in the form of a research article rather than a data description paper, and so, if the authors wish to keep the interpretation, I am not sure that ESSD is the most suitable journal. If the authors choose to stay with ESSD, a reconsideration of the scope is necessary, in my opinion. A slight reframing of the aims could perhaps help the manuscript stay on track with presenting (rather than interpreting) a dataset, for example: 1) present an up-to-date glacial lake dataset in the CPEC in 2020 using…; 2) present two historical glacial lake datasets for the CPEC to show extent in 1990 and 2000, using… The third aim could read well as a justification for the first two aims, rather than being an aim itself.

2. The errors are presented really nicely in Figure 4, and then do not seem to be mentioned again in the Results/Discussion/Conclusions. Regardless of the type of article this ends up as, I would expect more explicit mention of the absolute errors throughout the Results and Discussion (particularly where reasons for errors are discussed) – and for these to be summarised in the Abstract and Conclusions.

**Minor comments**

L24: The method is threshold-based, not object-oriented classification.

L28: I think this would be clearer if you rearranged to "2234 lakes were derived from the Landsat images…" and the same for the Sentinel clause in L30.

L32: Are there no existing inventories that use imagery with a lower resolution than Landsat? If so, it would be clearer if you specified this first, and then made a comparison – at the moment, the results presented here only show that Sentinel can detect smaller lakes than Landsat, which would be expected from the differing spatial resolutions.

L47: I think a reference to the passing of peak water e.g., (Huss and Hock, 2018) would be a better phrasing than "Unsustainable glacier melt … reducing the hydrological role of glaciers"

L54: I would avoid claims such as "inevitably affected" without an appropriate reference, so recommend removing this clause.

L54: Recommend changing "The increasing…" to "An increasing…"

L58-69: This paragraph feels like it belongs in the 'Study site' section due to the amount of detail and reference to Figure 1. I would move L61-69 there (or delete if repetitive), and simply summarise the first

sentence at the end of the previous paragraph – something like: "…and highways, such as One Belt One Road Initiative (BRI) infrastructure construction projects, which aim to strength connections between countries."

L87 and throughout: I recommend specifying Sentinel-2 for clarity – for instance, there are now many studies of lakes using Sentinel-1.

L109-110: Repetition of "mainly" – recommend changing first to "generally used". I'm also not entirely sure what is meant by "respective" – do you mean individual to each study using a classification system?

Figure 1: Recommend labelling the two panels to refer to in the caption, improving legibility of coordinates (by moving outside map or using a white background as for the legend), referencing source of layers such as glacier area and population count in the caption, and perhaps consider also labelling countries in the inset.

L133-134: Are there few GLOFs because there are very few glaciers in this region? If so, perhaps specify this – if not, a reference to support this statement could be useful.

L135 and throughout: A minor point, but there is inconsistency in the use of "altitude" and "elevation". To me, "altitude" is the height of something above the land surface and "elevation" is the height of the land surface (e.g., a plane would be at 300 m altitude whether above the elevation of the sea or a high mountain). I know others interpret these two terms differently, so perhaps just use one consistently for clarity.

L155-156: "We were unable to map lakes in 2010 due to Landsat 7's scan-line corrector…" would be better than "we had to give up".

L162: Can you state how many scenes were used for each baseline year in the main text? While Figure 2 is a helpful portrayal of the temporal range of images, the spatial coverage would also be useful – how did you decide when to stop choosing scenes in each baseline year, when each lake was imaged unobscured once/twice/…? Perhaps this can also be briefly summarised – having read on, I assume that one clear image is used to delineate each lake?

L207: If the method is human-interactive, surely it cannot be automated? Indeed, having read on, I do not believe this method is fully automated and recommend changing all mentions throughout to semi-automated.

L240: Remove the "and" for clarity here.

Tables 1 and 2: Please include the source of the glacier outlines (and reference if appropriate) and a description of the yellow markers in the captions.

L299: Was the coefficient revised in the current study? If so, can you provide brief details relating to why, how, and what the original coefficient was?

Figure 4: This is a very clear and effective way of showing the errors – nice!

L358-367 (and rest of Results): For those readers without an in-depth knowledge of the mountain ranges, this paragraph would be clearer if you referred to the river basins that are labelled in Figure 9. However, this point in the text and Figure 9 onwards strikes me as data interpretation beyond the scope of this journal article type and, unfortunately, I would recommend removing most of it – unless the target journal and article type were changed. In the latter case, I would just summarise this information more succinctly, focusing on highlighting the main points in the text, and include more consideration of the mapping errors.

Figures 7 and 8: Can you label in the caption that these are for GLCS 1 and 2, respectively?

L518-527: I am a little confused by this paragraph. Even if one Sentinel scene required manual georeferencing, once that was carried out there should not have been any subsequent errors in the lake areas calculated from that scene compared to any other – if so, I would remove the reference here and briefly mention in the Methods that one scene required manual georeferencing. The only way I can see an error

propagating through to the lake areas is if the manual georeferencing was not entirely accurate – is that the case? If so, please explain in more detail here.

L531-578: I'm afraid my understanding of the journal brief is that this section is beyond the scope of this article type and should be removed.

L619: There is Landsat imagery available long before 1990 – can you specify why you put this particular year?

L623: I would be more careful in stating how accurate your dataset is – Figure 4 shows that the uncertainty in some lake areas is > ± 80%, with many > ± 50%. I expected more discussion of the absolute errors, perhaps in the Discussion, and they should certainly be acknowledged here in the Conclusions.

**References**

Huss, M., and Hock, R. (2018). Global-scale hydrological response to future glacier mass loss. *Nat. Clim. Chang.* 8, 135–140. doi:10.1038/s41558-017-0049-x.

---

## Author Response (AR2)

Response to the editor

As per your request, and after conferring with co-editors, I have decided to give you an opportunity to further address and respond to the reviewer comments.
[Response] Thank you very much for providing the opportunity to further address all comments from reviewers and the editor. Here we respond all your comments point by point.

Their comment and frustration, evident in the statement "merely wanted to avoid using a spatial filter as it was "arduous"" is using the word 'arduous' from your manuscript. If there is/are a different reason(s) for those methods, you can clarify the text in the next revision.
[Response] To avoid any misunderstanding, we deleted the sentence that contains 'arduous'. We also revised this paragraph on Definition of glacial lakes and added one new Figure 3 for clarity.
Now it reads 'A 10-km buffering distance of RGI 6.0 glacier boundaries that has been widely used in previous studies (Zhang et al., 2015; Wang et al., 2020), was created to help mapping glacial lakes. A few glacial lakes (a total of 84 lakes for Sentinel-2 dataset and 55 lakes for Landsat dataset in 2020) beyond the buffering zone, located near buffering boundaries, were intentionally included due to clear evidence of glaciation (Figure 1).'

[Figure]

**Figure 1.** The 10-km buffer zone of RGI 6.0 glacier boundaries (a) and Sentinel-derived glacial lakes located near buffering boundary within the study area (b).

I agree that an arbitrary X-km buffer may not be the best approach, but also with the reviewers that distant lakes, and small lakes, may not matter for GLOFs - the focus of the manuscript is not quite the focus of the data. It is probably easier to re-focus the text than the data at this point.
[Response] The 10-km buffering distance of glacier boundaries has been used to help lake mapping, as stated at an earlier response. We agree that the value of our new dataset should

highlight the importance for water resource foremost throughout the manuscript, and we rewrote the Introduction and revised the focus throughout the manuscript. Please see the response to reviewer 2.

In addition to making sure the claims of the paper are completely supported by the data, I note R4 had concerns about the treatment of errors and uncertainty. This topic is of interest to me, and I'm generally uncertain about how certain most people are about how small their uncertainties are, but I do not claim to know the best way to handle errors myself. I appreciate your appendix demonstrating the error treatment. Are your errors random or likely to be biases? I need to know this if I'm going to do aggregate statistics on lake area, for example.

[Response] We are pleased to hear your interest to R4 or equation 4 on improved uncertainty estimating method, which is supposed to attract a broad range of users.

Uncertainty estimating method considering lake perimeter and displacement error are widely used in glacier and lake mapping from satellite observation, and one of the famous equations is the Hanshaw's equation. We firstly find that the number of edge pixels varies by the shape of lake and cannot be simply indicated by $\frac{P}{G}$. So we improved the equation and wrote a tutorial for users. My understanding is that the estimated uncertainty or relative error is likely to be biases. This is a way to help understanding accuracy of dataset. We also find the relative error is greater than absolute error calculated by randomly-selected lake boundaries derived from Google Earth high resolution images. More detail is presented in the main text and response to reviewer 3.

Some general comments from my reading of the latest version:
Similar to my interpretation of reviewer comments, I am concerned there is a lack of detail and precision throughout the manuscript. Many reviewer comments request more careful wording. Somewhat related, I note that Figure 4c is cut-off at ~35 % relative error, but Figure 4a appears to have brown dots up to ~60 %. The figure caption should more clearly describe what is being shown, that sub-panels are repeating data zoomed in (not something new/different (unless I'm mis-interpreting it)), and if any data is missing/cut-off/discarded, and if so, why.

[Response] We have to thank you again for pointing out the errors. We added a new Section '6.1 Error and uncertainty of lake mapping' to present relative and absolute errors in lake mapping, as also suggested by the reviewer 3. More detail can be read at Section 6.1.

We have fixed the mistake in previous Figure 4 (now Figure 8). Revised as below:

[Figure]

**Figure 2.** Estimated relative error for glacial lakes of all or specific size ranges in study area. Error estimation is based on the modified equation and lake data extracted from Landsat (a-d) and Sentinel-2 images (e-h).

L513 claims Figure 12e shows lakes "evolve dramatically in a short period", but the image does not support this claim. The changes appear to be Landsat vs. Sentinel.

[Response] We revised the current Figure 11 by removing unnecessary subgraphs and labeling the date in yellow. Now we can clearly see the supraglacial lakes evolving dramatically in a short period (between on September 19 in Figure 11d1 from Landsat and on August 25, 2020 in Figure 11d2 from Sentinel-2). Yes, the changes appear to be Landsat vs. Sentinel. And revised as '…evolve dramatically in a short period observed between Landsat and Sentinel-2 images…'

Section 6.2. It would be good to compare like with like - when comparing against 3rd party products as a form of validation, take common subsets both spatially (use the common area of the two products) and method-wise (limit your and their product to common areas, for example, lakes > X m or pixels). Then perform a quantitative or qualitative comparison. Of course, explaining how your data product is different (resolution, temporal, spatial) is also important to help users understand the difference and benefits of your products, once they know how it compares (for comparable items). But I'm not sure what the benefit of just "Overlap % (%)" is, nor do I know how to properly interpret that column with two % signs from only the limited

Table 6 caption.

[Response] Yes, what we wrote is as you described. We compared lake count and area greater than minimum mapping unit in the same common areas, and performed a quantitative or qualitative comparison, referring to mapping method and temporal differing etc. About the "overlap % (%)", we can change to 'Other's / ours % (%)' and add a note to claim that The % (%) represent the rates in count and area calculated by dividing individual glacial lake dataset by our Landsat-derived data in the nearest baseline year respectively.

However, the third reviewer suggests deleting this Section 'Comparison with previous similar dataset' to avoid any overshooting, and we deleted the Section. If the editor agrees to retain this Section, we can recover it using our backup.

Section 6.3 doesn't seem to address "updating plan" in the text, only the title. Nor should it. The "limitations" section is very important and totally separate from "Updates" (which is mentioned in Section 7 "Data availability"). Is this short paragraph all of your limitations? No other notes or warnings that you might want users to be aware of before they start performing analysis with this data product?

[Response] We moved the update plan from Section 7 "Data availability". As suggested by the second reviewer, we added two sentences to present "what we plan to do next", referring to more robust methods to reduce misclassifications and uncertainty. Those changes make the Section in a good shape. Thanks for your comments.

You now have four reviews, and some additional comments from me. Two reviews have not yet been addressed. Please re-read them, (and possibly the first two reviews too), and carefully address as many issues as you are able and willing to, and respond to the points you choose not to address. I look forward to the next revision.

[Response] We have carefully addressed all the comments point by point and revised the main text, including Figures. To address all the comments, we invited Jianrong Fan to join us in revising the manuscript and add her as a co-author. One thing we would like to clarify is that Reviewer 3 suggested to delete the previous Section 6.2 'Comparison with previous similar dataset' in order to avoid any overshooting, whereas Reviewer 2 suggested doing an inter-study comparison that takes into account at the same minimum lake size. After careful considerations, we decided to adopt the suggestion from Reviewer 3 to delete the Section, as we explained at an earlier response.

**Response to reviewer's comments:**
Note: in the text that follows, reviewer comments appear in black, whilst author responses appear in blue.

Now Referee #1 (the previous Reviewer 2)

The dataset paper "Landsat and Sentinel-derived glacial lake dataset in the China-Pakistan Economic Corridor from 1990 to 2020" presents a 1990-2020 inventory of glacial lakes for the CPEC region, including two classifications based on ice proximity and dam type. Overall, the dataset is a valuable addition with detailed metadata. However, changes to the manuscript do not adequately reflect the major revisions recommended from the previous round of reviews from both Reviewer #1 and Reviewer #2. I therefore recommend major revisions that are similar to the previous reviews, as two key shortcomings in the paper remain unaddressed:
[Response] We thank the reviewer again for her/his positive assessment of the work and for her/his constructive comments, which have helped us to improve the quality of this article. We try our best to address all the comments and respond to those point by point as follows.

1. The spatial definition of a glacial lake
A glacial lake is defined here as an active or ancient lake formed due to glaciation, deviating from the more readily used definition based on its spatial proximity to the margin of a glacier (as stated in L188-205). This alternative definition has not been used elsewhere in glacial lake and remote sensing papers (as far as I am aware), and gives the impression that the authors merely wanted to avoid using a spatial filter as it was "arduous" (as they state in L199).
[Response] We revised the paragraph 'Definition of glacial lakes' and added one new Figure 3 for clarity.
Now it reads 'A 10-km buffering distance of RGI 6.0 glacier boundaries that has been widely used in previous studies (Zhang et al., 2015; Wang et al., 2020), was created to help mapping glacial lakes. A few glacial lakes in the study area (a total of 84 lakes for Sentinel-2 dataset and 55 lakes for Landsat dataset in 2020) beyond the buffering zone, located near buffering boundaries, were intentionally included due to clear evidence of glaciation (Figure 1).'

To avoid any misunderstanding, we deleted the sentence that contains 'arduous'.

[Figure]

**Figure 1.** The 10-km buffer zone of RGI 6.0 glacier boundaries (a) and Sentinel-derived glacial lakes located near buffering boundary within the study area (b).

With their alternative definition, the mapping of these glacial lakes is useful as an overview of general water resources in the area for 1990 to near-present, such as municipal water supplies, agricultural irrigation and hydropower (as stated in L42-45). There is no doubt that this dataset is also of value for GLOF hazard monitoring. However, as the dataset also includes non-GLOF lakes, there is an imbalanced focus on GLOFs in this paper - for example, the large passage in the introduction (L49-89), and its inclusion in Aim 3 (L125-126) with no mention of water resources. To put this in perspective, the term "GLOF" is referred to 17 times in this paper, whereas references to water resources (i.e. the terms "freshwater", "water" and "resource") are only referred to 4 times.

Because of this, the dataset is mis-represented in the paper. In order to resolve this, the paper needs to be framed more strongly around its value to glacio-hydrology modelling and water resources, rather than GLOF hazards. This shift in focus should include:

A. Including more details about the importance of water resources in the CPEC region, such as the details about how much municipal water supply comes from glacial catchments, and the extent of hydropower infrastructure and its uses in the region. This should be added throughout the manuscript, including the introduction and conclusion

[Response] We thank the reviewer for her/his detailed explanation and comments. We rewrote the first two paragraphs in the Introduction to present the values of our novel dataset. We highlighted the importance for water resource foremost throughout the manuscript and added a few new citations.

It reads 'Global glacial lake number and total area both increased between 1990 and 2018 in response to glacier retreat and climate change (Shugar et al., 2020), affecting the allocation of freshwater resource. The Indus is globally the most important and vulnerable water tower unit where glaciers, lakes and reservoir storage contribute about two-thirds of the water supply (Immerzeel et al., 2020). Ice-marginal lakes store ~1% of total ice discharge in Greenland and accelerate lake-terminating ice velocity by ~25% (Mankoff et al., 2020; Carrivick et al.,

2022)…'

As a result, the term 'water resource' is referred to 9 times whereas 5 times for 'GLOF' in this paper.

B. Adjustment of the aims to better reflect the inventory's value to water resources, rather than GLOF hazards
[Response] The aims have been revised as:
'This study aims to (1) present an up-to-date glacial lake dataset in the CPEC in 2020 using both Landsat 8 and Sentinel-2 images to accurately document its detailed lake distribution; (2) present two historical glacial lake datasets for the CPEC to show extent in 1990 and 2000 using consistent 30-m Landsat images to reveal glacial lake changes at three time periods (1990, 2000 and 2020); and (3) generate a range of critical attributes for glacial lake inventories to benefit studies on water resource evaluation, risk assessment of GLOFs, glacier –lake evolution modeling in the HMA.'

C. Shortening of big passages about GLOF hazards in the introduction
[Response] As suggested, we shortened the Introduction particularly about GLOF hazards.

D. Stating the importance of the inventory to water resources first and foremost over GLOF hazards (e.g. L532, L577, L630)
[Response] We stated the importance of our dataset to water resource first in the entire main text. As a result, the term 'water resource' is referred to 9 times and 'GLOF' 5 times in this paper.
We deleted the previous Section 6.2 as suggested by the third reviewer, referring to L532 and L577. In previous L630, it now reads 'Our glacial lake dataset contains a range of critical parameters that maximize their potential utility for water resource…'

2. The size definition of a glacial lake
The dataset includes all lakes with a minimum mapping unit of 5 pixels, equating to a minimum lake size of 4500 sq m from 30-m resolution Landsat imagery, and 500 sq m from Sentinel-2 imagery. This is a smaller minimum size compared to similar studies (Rick et al., 2021; Shugar et al., 2020; Wang et al., 2020).
This choice is fine in itself, but becomes problematic when comparing between Sentinel- and Landsat-derived lakes, and comparing to other studies. The discrepancies in Sentinel- and Landsat-derived lakes is unsurprising, not only due to the differing spatial resolution (which is well documented), but also because of the difference in minimum lake size. Little can be concluded from the inter-study comparison other than the minimum lake size accounts for most of the visible discrepancies in lake count. In a dataset paper, you would expect to see much more robust methodology certainty analysis from an inter-study comparison. For example, it would be interesting to explore whether the use of the NDWI creates discrepancies in lake outline/delineation compared to others used solely or in combination (e.g. MNDWI, adopted by Shugar et al., 2021).
In order to address this, a more rigorous and insightful comparison needs to be presented in the

Discussion section. This should include:

A.   A more thorough analysis of discrepancies in lake form, as well as lake count, in Section 6.1

[Response] We reframed the structure of Discussion, starting from statement of 6.1 Error and uncertainty of lake mapping, then 6.2 Comparison of Sentinel-2 and Landsat derived products. In current Section 6.2, we removed the previous second paragraph on application in GLOFs and previous fifth paragraph on displacement-induced uncertainty to avoid any overshooting as suggested by the third reviewer and the editor. Consequently, we redrew the current Figure 11.

We show that Sentinel-2 images can be used to extract more glacial lakes and more accurate extents than those from Landsat images by Figures 10-11 and Table 4. The discrepancy between Landsat- and Sentinel-derived dataset is well presented. We would better not to interpret the dataset more in this study in order to meet the scope of ESSD papers. Thank you for this suggestion.

B.   An inter-study comparison that takes into account minimum lake size, with an exclusive areal comparison of lakes that appear in both inventories (in Section 6.2)

[Response] The previous Section 6.2 was deleted as suggested by the third reviewer to meet the scope of the ESSD articles. In that case, we cannot do inter-study comparison, meanwhile avoiding an overshooting to interpret the dataset. In addition, we compared lake dataset in the common area (the study area) using minimum mapping unit (MMU) of individual dataset. However, you suggested an exclusive areal comparison of lakes between inventories. We are willing to make such comparison by redoing statistics except Zhang's dataset with a smaller MMU (3 pixels) than ours in the case of the Section being approved by the editor to retain.

Thanks for your comprehension.

C. An exploration of the robustness of the NDWI method, with comparison to other spectral index methods. In addition, it would be interesting to see whether implementing a cloud and shadow masking step would reduce the uncertainty and need for manual intervention (as stated in L238-243). This could be framed as a tentative "what we plan to do next" section, using a smaller subset test region of CPEC. This should be added as its own section after Section 6.2. As a side note, the NDWI methodology presented here is a simple semi-automated workflow. Cloud and shadow masking is a fairly straightforward procedure that is well-established in remote sensing, yet is not included here. No doubt, this creates a greater needed for manual intervention where cloud and shadow misclassifications arise. In addition, the large need for manual intervention will require dedicated time to upkeep in future inventories (as promised by the authors). It is therefore in the authors' interest to begin exploring more robust methods and filtering processes to reduce misclassifications and uncertainty, and steer away from the need for manual intervention.

[Response] In this study, we minimized the effect of cloud cover or shadow by selecting high-quality images and redid the semi-automated lake mapping based on alternative images if glacial lake extents were contaminated by cloud or shadow in previous image. We did not

implement a cloud and shadow masking step in our processing of this study and could not analyze the impact of masking step on reducing the uncertainty.

In the Data sources, we added one sentence for clarity 'Spatially, high-quality images in the given baseline years were preferentially chosen, or we selected one or more alternative images acquired in adjacent years to delineate glacial lakes by removing the effect of cloud and snow covers.'

In previous L240, revised as '…were modified using previous semi-automated mapping method based on alternative images acquired in adjacent years.'

The robustness of the NDWI method has been presented in earlier publications of our team (Nie et al., 2013; Wang et al., 2014; Nie et al., 2017, 2020). Mapping approach combing NDWI and NDSI was adopted by Shugar et al., (2020) with a reference of our earlier publication (**Nie Yong**, Liu Qiao, Liu Shiyin. Glacial Lake Expansion in the Central Himalayas by Landsat Images, 1990–2010. PLoS ONE, 2013, 8: e83973.). Given the literature supporting the robustness of NDWI and to avoiding any overshooting, we opted not to do more comparison to other spectral index methods in this study.

As a response for your comment, we presented "what we plan to do next" in the end of first paragraph of the Section '6.3 Limitation and updating plan', but not an independent section.

It reads 'Third, the rigorous quality assurance and cross check after semi-automated lake mapping assure the quality of our lake dataset but are still time and cost prohibitive. State-of-the-art mapping methods, such as deep learning method (Wu et al., 2020), Google Earth Engine cloud-computing (Chen et al., 2021) and synergy of SAR and optical images (Wangchuk and Bolch, 2020; How et al., 2021), would be used in the future to balance product accuracy and time cost.'

Thanks for your comprehension and invaluable suggestions. We are exploring more robust methods to reduce misclassifications and uncertainty as you suggested. Hopefully, it will be successful soon.

**Response to reviewer's comments:**
Note: in the text that follows, reviewer comments appear in black, whilst author responses appear in blue.

The new Referee #2 (now Reviewer 3)

Review of ESSD-2021-468: "Landsat and Sentinel-derived glacial lake dataset in the China Pakistan Economic Corridor from 1990 to 2020"

Summary
This is my first review of the manuscript by Lesi et al., which presents a new glacial lake dataset, derived through remote sensing and semi-automated classification, over the CPEC area. The dataset is novel, well described and presented, and the manuscript is generally clear and well written.
[Response] First of all, thank you very much for your detailed and constructive comments. Your advice plays a very important role in improving the quality of our paper. We have responded to the questions or suggestions one by one.

However, I have two concerns; the first relating to the content vs. scope of the article type, and the second to the discussion of errors:
1. My understanding of the scope of ESSD is that a novel dataset is presented with minimal analysis (at least none that goes beyond assessing the accuracy of the dataset) and no interpretation. While I notice that recently published data papers in ESSD vary in the amount of analysis/interpretation they contain, my view is that here, there is analysis and interpretation of the dataset that goes beyond the scope of a data description paper (e.g., a large part of the Results, and part of the Discussion – detailed below). To me, the manuscript is presented in the form of a research article rather than a data description paper, and so, if the authors wish to keep the interpretation, I am not sure that ESSD is the most suitable journal. If the authors choose to stay with ESSD, a reconsideration of the scope is necessary, in my opinion. A slight reframing of the aims could perhaps help the manuscript stay on track with presenting (rather than interpreting) a dataset, for example: 1) present an up-to-date glacial lake dataset in the CPEC in 2020 using…; 2) present two historical glacial lake datasets for the CPEC to show extent in 1990 and 2000, using… The third aim could read well as a justification for the first two aims, rather than being an aim itself.
[Response] Thank you very much for this comment. As suggested, we have reframed the aims, and it now reads:
'This study aims to (1) present an up-to-date glacial lake dataset in the CPEC in 2020 using both Landsat 8 and Sentinel-2 images to accurately document its detailed lake distribution; (2) present two historical glacial lake datasets for the CPEC to show extent in 1990 and 2000 using consistent 30-m Landsat images to reveal glacial lake changes at three time periods (1990, 2000 and 2020); and (3) generate a range of critical attributes for glacial lake inventories to benefit studies on water resource evaluation, risk assessment of GLOFs, glacier –lake evolution modeling in the HMA.'

To stay on track with presenting (rather than interpreting), we deleted unnecessary sentences and paragraphs in the results and discussion. For example, we deleted previous Figure 6, Figures 9-10 and Table 4, as well as the corresponding main text. We also responded the following detailed comments for your approval.

2. The errors are presented really nicely in Figure 4, and then do not seem to be mentioned again in the Results/Discussion/Conclusions. Regardless of the type of article this ends up as, I would expect more explicit mention of the absolute errors throughout the Results and Discussion (particularly where reasons for errors are discussed) – and for these to be summarised in the Abstract and Conclusions.
[Response] This is a key point that we nearly ignored in previous version. Thank you. We moved the Figure 4 and corresponding description to the Discussion entitled '6.1 Error and uncertainty of lake mapping' and added a paragraph to introduce Validation of glacial lake mapping in the Section '4.5 Error and uncertainty assessment'.

In the Section 6.1 'Error and uncertainty of lake mapping', we presented the relative error estimated by our improved uncertainty estimating method and absolute error validated by manually digitized lake boundaries based on Google Earth high resolution images.
It reads 'Total area error of glacial lakes in study area is approximate $\pm14.98$ km$^2$ and $\pm8.45$ km$^2$ in 2020 for Landsat and Sentinel-2 dataset, respectively, and the average relative error is $\pm17.36\%$ and $\pm8.15\%$...Our Landsat- and Sentinel-derived glacial lake dataset match well lake boundaries in Google Earth high resolution images (Figure 9). A dense cluster of validation samples along the 1:1 line indicates a high accuracy in lake mapping (Figure 9c and d)... Our glacial lake dataset shows a satisfactory mapping accuracy, and of which Sentinel-derived lake data performs more accurate than those from Landsat images.'

We put the Section 'Error and uncertainty of lake mapping' at the beginning of the Discussion, then discussed the lake dataset difference and associated causes.

We also summarized the mapping error in the Abstract and Conclusions.
In the Abstract, it reads 'Glacial lake data in 2020 was validated by Google Earth-derived lake boundaries with a median ($\pm$standard deviation) difference of 7.66$\pm$4.96 % for Landsat-derived product and 4.46$\pm$4.62 % for Sentinel-derived product.'
In the Conclusions, it reads 'The average relative error is $\pm17.36\%$ for Landsat-derived product and $\pm8.15\%$ for product from Sentinel-2.'

Minor comments
L24: The method is threshold-based, not object-oriented classification.
[Response] Revised as '…based on threshold-based mapping method…'

L28: I think this would be clearer if you rearranged to "2234 lakes were derived from the Landsat images…" and the same for the Sentinel clause in L30.

[Response] Revised as 'The results show that, in 2020, 2234 lakes were derived from the Landsat images, covering a total area of 86.31±14.98 km$^2$ with a minimum mapping unit of 5 pixels (4500 m$^2$), whereas, 7560 glacial lakes were derived from the Sentinel-2 images with a total area of 103.70±8.45 km$^2$ with a minimum mapping unit of 5 pixels (500 m$^2$).'

L32: Are there no existing inventories that use imagery with a lower resolution than Landsat? If so, it would be clearer if you specified this first, and then made a comparison – at the moment, the results presented here only show that Sentinel can detect smaller lakes than Landsat, which would be expected from the differing spatial resolutions.
Response: As we know none of glacial lake inventories using a lower resolution than Landsat exists in the entire study area, and we firstly produce glacial lake dataset from both Sentinel-2 and Landsat in the CPEC.
To be clearer, this sentence revised as 'The discrepancy shows that Sentinel-2 is able to detect a significant quantity of smaller lakes than Landsat due to its finer spatial resolution.'

L47: I think a reference to the passing of peak water e.g., (Huss and Hock, 2018) would be a better phrasing than "Unsustainable glacier melt … reducing the hydrological role of glaciers"
[Response] Revised as '…unsustainable glacier melt and the passing of peak water are reducing the hydrological role of glaciers (Huss and Hock, 2018) and…'

L54: I would avoid claims such as "inevitably affected" without an appropriate reference, so recommend removing this clause.
[Response] It has been removed.

L54: Recommend changing "The increasing…" to "An increasing…"
[Response] Revised as 'An increasing…'.

L58-69: This paragraph feels like it belongs in the 'Study site' section due to the amount of detail and reference to Figure 1. I would move L61-69 there (or delete if repetitive), and simply summarise the first sentence at the end of the previous paragraph – something like: "…and highways, such as One Belt One Road Initiative (BRI) infrastructure construction projects, which aim to strength connections between countries."
[Response] We rewrote this paragraph by putting some sentences to the end of the previous paragraph and moving some sentences to the Study area.

In the first paragraph, it now reads 'in the mountain ranges, such as the China-Pakistan Economic Corridor (CPEC), as a flagship component of One Belt One Road Initiative (Battamo et al., 2021; Li et al., 2021). The northern section of the CPEC passes through Pamir, Karakoram, Hindu Kush and Himalaya mountains where droughts and glacier-related hazards are frequent and severe (Hewitt, 2014; Bhambri et al., 2019; Pritchard,

2019), threatening local people, the existing, under-construction and planned infrastructures, such as highways, hydropower plants and railways.'

In the Study area, it reads 'The northern part of the CPEC is selected as the study area (Figure 1). The CPCE, originating from Kashgar of the Xinjiang Uygur Autonomous region, China and extending to Gwadar Port, Pakistan (Ullah et al., 2019; Yao et al., 2020), is connecting China and Pakistan via the only Karakoram Highway.'

L87 and throughout: I recommend specifying Sentinel-2 for clarity – for instance, there are now many studies of lakes using Sentinel-1.
[Response] We changed 'Sentinel' to 'Sentinel-2' in the main text and Figures.

L109-110: Repetition of "mainly" – recommend changing first to "generally used". I'm also not entirely sure what is meant by "respective" – do you mean individual to each study using a classification system?
[Response] Revised as 'Existing classification systems are generally used for their individual research purposes…'

Figure 1: Recommend labelling the two panels to refer to in the caption, improving legibility of coordinates (by moving outside map or using a white background as for the legend), referencing source of layers such as glacier area and population count in the caption, and perhaps consider also labelling countries in the inset.
[Response] Thanks for your valuable advice, the Figure 1 was modified accordingly and the references for glacier and population count were added.

[Figure]

**Figure 1.** Location of the study area associated with distribution of glaciers (RGI Consortium, 2017), mountains, basins and population (Rose et al., 2021) (a), and its

location within the CPCE (b).

L133-134: Are there few GLOFs because there are very few glaciers in this region? If so, perhaps specify this – if not, a reference to support this statement could be useful.
[Response] To avoid any misunderstanding, this sentence is revised as 'The upper Indus basins beyond the Pakistani-administrated border are excluded in this study due to spatial coverage of the CPCE.'

L135 and throughout: A minor point, but there is inconsistency in the use of "altitude" and "elevation". To me, "altitude" is the height of something above the land surface and "elevation" is the height of the sea surface (e.g., a plane would be at 300 m altitude whether above the elevation of the sea or a high mountain). I know others interpret these two terms differently, so perhaps just use one consistently for clarity.
[Response] Now it has been revised as '…covering most of the Karakoram with the highest elevation up to 8611 m…', and we also corrected this in the other part of main text.

L155-156: "We were unable to map lakes in 2010 due to Landsat 7's scan-line corrector…" would be better than "we had to give up".
[Response] Revised as '…so we were unable to map lakes in 2010 due to Landsat 7's scan-line corrector errors and significant cloud covers…'

L162: Can you state how many scenes were used for each baseline year in the main text? While Figure 2 is a helpful portrayal of the temporal range of images, the spatial coverage would also be useful – how did you decide when to stop choosing scenes in each baseline year, when each lake was imaged unobscured once/twice/…? Perhaps this can also be briefly summarised – having read on, I assume that one clear image is used to delineate each lake?
[Response] Yes, we first chose the best quality image in the given baseline year in each scene. If the image was not available, we searched for alternative images until meeting the demand of glacial lake mapping. The main text has been revised as:
'Only 4 images in 1990 (the largest covering the study area), 16 images in 2000 and 23 images in 2020 were used for matching baseline year. Spatially, high-quality images in given baseline years were preferentially chosen, or we selected one or more alternative images acquired in adjacent years to delineate glacial lakes by removing the effect of cloud and snow covers.'

L207: If the method is human-interactive, surely it cannot be automated? Indeed, having read on, I do not believe this method is fully automated and recommend changing all mentions throughout to semiautomated.
[Response] Thanks for your suggestion, 'semi-automated' was more rigorous here. We have changed the 'automated' into 'semi-automated' throughout the manuscript.

L240: Remove the "and" for clarity here.

[Response] Removed 'and'.

Tables 1 and 2: Please include the source of the glacier outlines (and reference if appropriate) and a description of the yellow markers in the captions.
[Response] Now, the source of the glacier outlines and the description of the yellow marker were added to the captions as follows:
'Glacier outlines are from RGI 6.0 (RGI Consortium, 2017), and the yellow marker represents target lake.'

L299: Was the coefficient revised in the current study? If so, can you provide brief details relating to why, how, and what the original coefficient was?
[Response] We deleted 'revised' to avoid any misunderstanding. In this equation for uncertainty estimation, we revised the equation by removing repeatedly calculated edge pixels, not revising the coefficient. As stated 'The Hanshaw's error estimation method for pixel-based lake mapping was improved by removing repeatedly calculated edge pixels that vary with lake shape.'

Figure 4: This is a very clear and effective way of showing the errors – nice!
[Response] Thank you very much. As suggested, we moved this Figure to the Discussion.

L358-367 (and rest of Results): For those readers without an in-depth knowledge of the mountain ranges, this paragraph would be clearer if you referred to the river basins that are labelled in Figure 9. However, this point in the text and Figure 9 onwards strikes me as data interpretation beyond the scope of this journal article type and, unfortunately, I would recommend removing most of it – unless the target journal and article type were changed. In the latter case, I would just summarise this information more succinctly, focusing on highlighting the main points in the text, and include more consideration of the mapping errors.
[Response] The description of the glacial lake distribution among the mountain ranges has been removed. Furthermore, we shortened the Result by deleting previous Figure 6, Figure 9-10 and Table 4, as well as corresponding text.

We tried our best to focus on data description, and presented the dataset via location and two classification systems.

We highlighted the mapping errors all in the Discussion as it refers to some references that generally cannot appear in the Results. Thanks for your understanding.

Figures 7 and 8: Can you label in the caption that these are for GLCS 1 and 2, respectively?
[Response] Yes, sure. Now it has been added to the captions for clarity.

L518-527: I am a little confused by this paragraph. Even if one Sentinel scene required

manual georeferencing, once that was carried out there should not have been any subsequent errors in the lake areas calculated from that scene compared to any other – if so, *I would remove the reference here and briefly mention in the Methods that one scene required manual georeferencing.* The only way I can see an error propagating through to the lake areas is if the manual georeferencing was not entirely accurate – is that the case? If so, please explain in more detail here.

[Response] We agree. We deleted this paragraph here, modified the previous Figure 12, and briefly introduced this in the Data sources.

In the last two sentences of the first paragraph of Data sources, it reads 'All images used in this study have been orthorectified before download, but we still find that one Sentinel-2 image was not well matched with Landsat images, leading to the discrepancy between the two glacial lake datasets. We manually georeferenced the shifted image to minimize the difference between Sentinel and Landsat derived glacial lakes.'

L531-578: I'm afraid my understanding of the journal brief is that this Section is beyond the scope of this article type and should be removed.

[Response] We deleted the previous L531-578.

L619: There is Landsat imagery available long before 1990 – can you specify why you put this particular year?

[Response] We agree and revised as '…at a consistent spatial resolution of 30 m starting from the late 1980s.'

L623: I would be more careful in stating how accurate your dataset is – Figure 4 shows that the uncertainty in some lake areas is $> \pm 80\%$, with many $> \pm 50\%$. I expected more discussion of the absolute errors, perhaps in the Discussion, and they should certainly be acknowledged here in the Conclusions.

Response: We adopted this comment. We added one paragraph in the Section 4.5.2 to validate the absolute error of lake mapping based on manually digitized lake boundaries based on Google Earth high resolution images and also discussed this in the Discussion 6.1.

About the relative error or uncertainty, we add one sentence to explain 'Because the relative error was estimated as a function of satellite image spatial resolution and lake perimeter, the calculated error for large lake is proportionally smaller than that of small lake (Salerno et al., 2012) and the error for Landsat-derived lake is naturally greater than that of Sentinel-derived lake at the same size group.'

We also summarized the mapping error in the Abstract and Conclusions as stated in the earlier response.

References
Huss, M., and Hock, R. (2018). Global-scale hydrological response to future glacier mass loss. Nat. Clim. Chang. 8, 135–140. doi:10.1038/s41558-017-0049-x.
Response: Thanks for recommending this reference that has been cited.

**Response to reviewer's comments:**
Note: in the text that follows, reviewer comments appear in black, whilst author responses appear in blue.

The previous Referee #1

Overall comment:
This manuscript is in good shape. But there are two issues that MUST have attention given to them. (i) Terminology..the classes of lakes are not named at all well. I have suggested what to do. (ii) Argument of why we need to detect small lakes.; I am totally unconvinced by the GLOF angle...rather I suggest thinking about the effects of lakes on glaciers and the fact that many of these newly-formed small lakes will become larger with ongoing glacier mass loss. I have offered a coitation to start these thoughts but really a whole paragraph needs adding. Else the abstract needs a complete overhaul too.
[Response] Thank you for your encouragement and constructive comments, which have helped us to improve the quality of this article. We respond to the comments point by point as follows.

Regarding the terminology of lake classification systems, we have revised it as suggested. Now, in the first glacial lake classification system, glacial lakes were classified into four types based on their spatial relationship to upstream glaciers: supraglacial, ice-contact, unconnected-glacier-fed lakes, and non-glacier-fed lakes. In the second glacial lake classification system, glacial lakes were classified into five categories (herein named GLCS2) modified based on Yao's classification system (2018): supraglacial, end-moraine-dammed, lateral-moraine-dammed, glacial-erosion lakes and ice-dammed lakes.

Regarding small lakes, we agree that small lakes have little or no hazardous impact due to their limited water release. The focus of this study is to generate a new glacial lake dataset for the CPEC, using 5 pixels as the mapping threshold for both Landsat and Sentinel-2 images. We had to map all the glacial lakes, including small ones. As suggested by the second reviewer, we rewrote the first two paragraphs in the Introduction to present the values of our novel dataset. We highlighted the importance for water resource foremost throughout the manuscript and added a few new citations.

We have also cited the recommended references and revised the abstract.

Specific comments:

Line 19: suggest rewording to …'one of a number of flagship projects…'
[Response] Revised as '…one of the flagship projects…'

Line 22, suggest delete 'critical parameters' and state '…parameters X and Y and Z…' (list them out) Are these ALL glacial lakes? Or just ice-marginal ones? Supraglacial? Subglacial? Please specify. Add this specification into your methods.
[Response] Considering the dataset with 18 attributes, it is not suitable to list all of them here. So we decided to list three parameters for example, and revised this sentence to be: 'An up-to-date high-quality glacial lake dataset with parameters such as lake area, volume and type, which is fundamental to water resource and flood risk assessments …'

This study defines a glacial lake as one that formed as a result of modern or ancient glaciation. All glacial lakes in the study area were mapped according to our definition. So it is not just ice-marginal glacial lakes. The dataset includes supraglacial lakes; however, it does not include subglacial lakes that are not detectable from optical satellite sensors. See Section 4.1 and 4.3 for detailed description.

Line 24. I suggest to put the resolution(s) after the dataset type. Split sentence into two. One for lakes, one for glaciers, for clarity (because as written it is not clear if OI was for lakes or glaciers or both).

[Response] Following the suggestion, we have revised this sentence to be: 'This dataset includes (1) multi-temporal inventories for 1990, 2000, and 2020 produced from 30 m resolution Landsat images, and (2) a glacial lake inventory for the year 2020 at 10 m resolution produced from Sentinel-2 images.'

Line 30…is this 5 pixel threshold for both Landsat and Sentinel? Please clarify the thresholds for BOTH datasets.

[Response] Yes, the 5-pixel threshold is for both Landsat and Sentinel images. It reads 'The results show that, in 2020, 2234 lakes were derived from the Landsat images, covering a total area of $86.31\pm14.98$ km$^2$ with a minimum mapping unit of 5 pixels (4500 m$^2$), whereas, 7560 glacial lakes were derived from the Sentinel-2 images with a total area of $103.70\pm8.45$ km$^2$ with a minimum mapping unit of 5 pixels (500 m$^2$).'

Line 31…I'm not sure this is 'discrepancy', rather simply a result that can be interpreted to be due to many small lakes.

[Response] We revised as 'The discrepancy shows that Sentinel-2 is able to detect a significant quantity of smaller lakes than Landsat due to its finer spatial resolution.'

Line 32. Are (very) small lakes important? For hazards/GLOFs? Why? I think you need to discuss/show this….in the main text of the manuscript as well as here in the abstract…else the whole premise of your work is not represented/defended/argued (?!).

[Response] As suggested by the second reviewer, we rewrote the Abstract, Introduction and main text to present the values of our novel dataset. We highlighted the importance for water resource foremost throughout the manuscript, followed by glacial lake-related hazards, glacier-lake interactions.
Consequently, the aims have been revised as: 'This study aims to (1) present an up-to-date glacial lake dataset in the CPEC in 2020 using both Landsat 8 and Sentinel-2 images to accurately document its detailed lake distribution; (2) present two historical glacial lake datasets for the CPEC to show extent in 1990 and 2000 using consistent 30-m Landsat images to reveal glacial lake changes at three time periods (1990, 2000 and 2020); and (3) generate a range of critical attributes for glacial lake inventories to benefit studies on water resource evaluation, risk assessment of GLOFs, glacier –lake evolution modeling in the HMA.'

Line 36…would be more useful to state the types of lakes please. And state the two classifications systems please. Be explicit (!). what is the improved equation?! Name it!

[Response] The two classifications systems contain a total of nine types of glacial lakes, so specifying all types in the abstract will take up too much space. To keep the abstract concise and present improved equation, we changed this sentence to: 'A range of critical attributes have been generated in the dataset, including lake types and mapping uncertainty estimated by an improved Hanshaw's equation.'

Line 37. Potentials is not plural. Remove the 's'.

[Response] We have changed this sentence to '…glacial lake dataset has potential to be widely applied in studies on water resource assessment, glacial lake-related hazards, glacier-lake interactions …'

Line 48. You really must have to cite Carrivick and Tweed (2016)

https://www.sciencedirect.com/science/article/pii/S0921818116301023?via%3Dihub please! here. Furthermore, if you read that paper, the size of lakes producing hazardous GLOFs is reported. Small lakes (like the ones detected by your sentinel analysis v landsat) are not hazardous (!).

[Response] We agree and have cited the reference from Carrivick and Tweed (2016). It now reads:

'…impacting downstream ecosystem services, agriculture, hydropower and other socioeconomic values (Carrivick and Tweed, 2016; Nie et al., 2021).'

About small lakes, we have responded at an earlier comment.

Line 97. Please explain 'type' is this glacier terminus environment? Is it lake dam type? Is it lake position (supraglacial or ice-marginal for example?).

[Response] As suggested, we have revised this to be: 'Dam type classification of glacial lakes provides a crucial attribute for glacier-lake interactions …'

Line 173. A glacial lake is one that receives meltwater from a glacier. Of these most are proglacial (beyond the glacier) and can be attached (ice-marginal or ice-contact) or detached from the edge of the glacier. PLEASE correct this terminology. Then say what you do (which means you need to evaluate what sort of lakes you are actually analysing!).

[Response] We agree to divide proglacial lakes into ice-contact and unconnected-glacier-fed (detached) lakes in this study. We revised the classification system of glacial lakes. See the Section 4.3.

We consider a glacial lake as one that formed as a result of modern or ancient glaciation. In this study, all glacial lakes were mapped according to this definition and are attributed using the two classification systems.

Line 186 'without any distance limit'…oh come on there must have been some limit?! The catchment or study area boundary at least?! Please evaluate what you have done and report it carefully.

[Response] Thank you for this comment. We deleted this sentence, revised this paragraph 'Definition of glacial lakes' and added one new Figure 3 for clarity.

Now it reads 'A 10-km buffering distance of RGI 6.0 glacier boundaries that has been widely used in previous studies (Zhang et al., 2015; Wang et al., 2020), was created to help mapping glacial lakes. A few glacial lakes in the study area (a total of 84 lakes for Sentinel-2 dataset and 55 lakes for Landsat dataset in 2020) beyond the buffering zone, located near buffering boundaries, were intentionally included due to clear evidence of glaciation (Figure 1).'

[Figure]

**Figure 1.** The 10-km buffer zone of RGI 6.0 glacier boundaries (a) and Sentinel-derived glacial lakes located near buffering boundary within the study area (b).

Line 207 this info. on mapping units needs to be accurately represented in the abstract.
[Response] We agree, and revised as: 'in 2020, 2234 lakes were derived from the Landsat images, covering a total area of 86.31±14.98 km$^2$ with a minimum mapping unit of 5 pixels (4500 m$^2$), whereas, 7560 glacial lakes were derived from the Sentinel-2 images with a total area of 103.70±8.45 km$^2$ with a minimum mapping unit of 5 pixels (500 m$^2$).'

Line 233. This spatial relationship needs to be explicitly named above in the manuscript where I have already queried it. I dislike this classification. See Carrivick and Tweed (2013) https://www.sciencedirect.com/science/article/pii/S027737911300293X for definition of proglacial lakes (my comment for line 173). Supraglacial is a distinct group so that is OK. Proglacial and unconnected are the same/overlap…you need 'ice-marginal' and 'other proglacial' I think, then 'other lakes' as your classes/types.
[Response] Thank you for this valuable suggestion. As we responded earlier, we agree to divide proglacial lakes into ice-contact and unconnected-glacier-fed (detached) lakes. We have revised this consistently throughout the main text, figures, tables and attribute of our glacial lake dataset.

This sentence was changed to '…glacial lakes were classified into four types based on their spatial relationship to upstream glaciers: supraglacial, ice-contact, unconnected-glacier-fed lakes, and non-glacier-fed lakes according to Gardelle et al. (2011) and Carrivick et al. (2013).'

Line 238. The terminology again is wrong here and confusing because mixes position and dam type. See Carrivick and Tweed (2013) https://www.sciencedirect.com/science/article/pii/S027737911300293X . You should have supraglacial, terminus moraine-dammed, lateral moraine dammed, ice-dammed and bedrock-dammed I suggest.
[Response] Considering the formation mechanism and dam properties of glacial lakes, the second glacial lake classification system was established via modifying Yao's classification system (2018). According to your suggestion, we have revised the terminology of the classification system to ice-dammed, end-moraine-dammed, lateral-moraine-dammed and supraglacial lakes. Glacial-erosion lakes contain both bedrock-dominated dam and top-moraine-mixed dam, so we prefer to use glacial-erosion lakes instead of bedrock-dammed lakes. It now reads:

'Alternatively, combining the formation mechanism of glacial lakes and the properties of natural dam features, glacial lakes were classified into five categories (herein named GLCS2) modified

from Yao's classification system (2018): supraglacial, end-moraine-dammed, lateral-moraine-dammed, glacial-erosion lakes and ice-dammed lakes.'

Line 318 to 326. I suggest to compare to (and cite) Carrivick and Quincey (2014) who also consider uncertainty v lake area.
https://www.sciencedirect.com/science/article/pii/S092181811400054X?via%3Dihub
[Response] We cited the reference from Carrivick and Quincey (2014). It now reads 'Lake perimeter and displacement error are widely used to estimate the uncertainty of glacier and lake mapping from satellite observation (Carrivick and Quincey, 2014; Hanshaw and Bookhagen, 2014; Wang et al., 2020).'

The difference in uncertainty estimation between Carrivick's and Hanshaw's methods is that Carrivick assumes an uncertainty of ±1 pixel, while Hanshaw assumes an uncertainty of ±0.5 pixels and counts the number of edge pixels. In this study, we discovered and solved the problem of repeatedly calculated edge pixels. Considering that the mean lake size in the study area is smaller than that in the Greenland, we prefer to choose the improved Hanshaw's equation to estimate the mapping uncertainty.

Line 453…so do we need Sentinel images for lake mapping?? If Landsat is doing a good job v sentinel (detection as well as accuracy) then why do we need the extra resolution? What importance do the numerous small lakes have? They are not important volumetrically? Are they important for hazards/GLOFs? (I don't think so!). I really think the 'promoted capacity of GLOF risk assessment' (line 543) needs further elaboration.
[Response] We believe Sentinel images do offer their unique benefits in mapping glacier lakes, owing to their finer spatial resolution, increasing capacity of revisit observation and accurately depicting lake boundaries with a lower uncertainty. We further clarified these as: 'Due to a finer spatial resolution, Sentinel images can extract more glacial lakes and more accurate extents than those from Landsat images…. Sentinel-2 images are able to depict boundaries of glacial lake with a lower uncertainty, as for some small islands and narrow channels (Figure 11b and c) were mapped from Sentinel-2 imagery that were unable to be detected in Landsat imagery.'

Regarding small lakes, we have responded to a similar query earlier:

The previous Section 6.2 was deleted as suggested by the third reviewer to meet the scope of the ESSD articles. The sentence 'promoted capacity of GLOF risk assessment' (line 543) was also deleted.

In contrast, I think a utility of your dataset and indeed your sentinel-based detection of many small lakes is that those small lakes could be the onset of fast-developing proglacial landscapes…and they will likely grow as glaciers diminish further and affect glacier dynamics (see Carrivick et al., 2020 for example
https://www.frontiersin.org/articles/10.3389/feart.2020.577068/full )
[Response] Thank you for your affirmation and encouragement. The Sentinel-derived lake dataset has a wider potential than Landsat-derived dataset to be used in studies on proglacial landscape change and glacier dynamic assessment. The recommended reference is important and cited in the main text.

**Response to reviewer's comments:**
Note: in the text that follows, reviewer comments appear in black, whilst author responses appear in blue.

The previous Reviewer 2 (round 1)

The glacial lake dataset presented in 'Landsat and Sentinel-derived glacial lake dataset in the China-Pakistan Economic Corridor from 1990 to 2020' identifies and classifies lakes in the CPEC region of High-mountain Asia from three time steps - 1990, 2000 and 2020.
Lakes are identified from Landsat and Sentinel-2 optical satellite imagery using a semi-automated approach that utilises the well-established NDWI (Normalised Difference Water Index) method. Statistics and analysis of lake abundance and size distribution are presented, along with changes in lakes over the course of the time-series and inter- comparison of the Landsat- and Sentinel-derived lake outlines. The dataset is then compared to alike glacial lake datasets from the same region, in order to examine and evaluate discrepancies.

This is a valuable dataset that I foresee will be readily used by the cryosphere and hydrology research communities. In particular, the use of two highly-detailed lake classification systems (based on Gardelle et al., 2011, and a modified version of Yao, 2018) is a unique aspect of the dataset that is insightful alongside the general size and abundance information. This type of classification is seldom seen in glacial lake datasets, and reflects the thoroughness of the dataset.

The manuscript is structured in a clear and concise manner, guiding the reader through the dataset methods and description, results and statistics from the datastes, followed by an evaluation of the dataset scope and certainty.

Several key points need to be addressed, which are detailed below, largely regarding the dataset itself and the definition of a glacial lake. The comparison to alike datasets is flawed given that many of the discrepancies are due to the differing definitions of a glacial lake, rather than the classification method itself. Once these major revisions have been addressed, then the manuscript and associated dataset will be a great addition to ESSD.
[Response] We thank the reviewer for her/his positive assessment of the work and for her/his constructive comments, which have helped us to improve the quality of this article. We respond to the comments point by point as follows.

**Major comments**

**1. Dataset transparency**
A large part of the presented dataset is manually derived - metadata generation, georeferencing, and outline modifications. This can make reproducibility challenging. I would like to see a version of this dataset provided in the supplementary material that presents the dataset before manual intervention/inputs. Therefore, readers can see the dataset before and after manual modifications, and tangibly distinguish the automated and user-defined components of the methodology presented.
[Response] In this study, we used a human-interactive, semi-automated lake mapping method (Wang et al., 2014; Nie et al., 2017; Nie et al., 2020) to accurately extract glacial lake extents. The used method is flexible and of high reproducibility to map lake boundary by tuning NDWI threshold while screening the NDWI histogram, and automatically generating vector polygons. More detailed information can be seen in our previous publications, such as Wang et al., 2014. In the process of interactive lake mapping, manual inputs refer to the drawing of user-defined region

of interest (ROI) and tuning the NDWI threshold in each ROI, whereas calculating the histogram of NDWI and converting raster lake extent to vector polygon were automated. Our lake dataset contains pixelated polygons, rather than manually digitized polygons. We do not provide the dataset before and after manual modifications because of the absence of manually-modified lake polygons. To avoid misunderstanding, we define the method as a human-interactive and semi-automated lake mapping method and made some revision.

It now reads 'Specifically, the method calculated the NDWI histogram based on the pixels with each user-defined and manually-drawn region of interest. The NDWI threshold that separates lake surface from land was interactively determined by screening the NDWI histogram against the lake region in the imagery (Nie et al., 2020; Wang et al., 2014). This way, the determined NDWI threshold can be well-tuned to adapt various spectral conditions of the studied glacier lakes. The raster lake extents segmented by the thresholds were then automatically converted to vector polygons.'

**2. Definition of a glacial lake**

A large focus in the manuscript is glacier-related hazards, specifically GLOFs and draining lakes that are either on or share a boundary with a glacier. However, the dataset includes lakes that are not influenced or effect glacier dynamics, such as lakes that are hydrologically unconnected from a glacier. The abundance of glaciologically-unconnected lakes markedly influences the identified trends in the dataset, such as the visible influence on lake abundance in GLCS1. In addition, the dataset lacks a spatial filter relative to the ice margin. If indeed the aim of the dataset is to inform on glacier-related hazards, this dataset should focus exclusively on active glacial lakes, rather than active and ancient lakes.

[Response] Thank you for constructive comments. We agree and present the values of our novel dataset by highlighting the importance for water resource foremost throughout the manuscript to keep small lakes and non-GLOF lakes in the lake dataset. Correspondingly, we revised the aims to be '…(1) present an up-to-date glacial lake dataset in the CPEC in 2020 using both Landsat 8 and Sentinel-2 images to accurately document its detailed lake distribution; (2) present two historical glacial lake datasets for the CPEC to show extent in 1990 and 2000 using consistent 30-m Landsat images to reveal glacial lake changes at three time periods (1990, 2000 and 2020); and (3) generate a range of critical attributes for glacial lake inventories to benefit studies on water resource evaluation, risk assessment of GLOFs, glacier –lake evolution modeling in the HMA.'

We revised the Paragraph 'Definition of glacial lakes' and added one new Figure 3 to present a spatial filter for clarity.

Now it reads 'A 10-km buffering distance of RGI 6.0 glacier boundaries that has been widely used in previous studies (Zhang et al., 2015; Wang et al., 2020), was created to help mapping glacial lakes. A few glacial lakes in the study area (a total of 84 lakes for Sentinel-2 dataset and 55 lakes for Landsat dataset in 2020) beyond the buffering zone, located near buffering boundaries, were intentionally included due to clear evidence of glaciation.'

Another aspect is the small threshold size used for the glacial lake dataset. Again, if the focus of this paper is glacier-related hazards then small lakes (<0.05 sq km; Shugar et al., 2021) are largely irrelevant to this study as they have limited GLOF impact. These small lakes make up over 80% of the dataset and heavily influence the identified temporal trends.

[Response] We present the values of our lake dataset by highlighting the importance for water resource foremost throughout the manuscript to keep small lakes and non-GLOF lakes in the lake dataset, as stated in an earlier response. Thank you for pointing out this.

In order to overcome this, I would suggest shifting the focus of the manuscript away from glacier-related hazards and framing the manuscript under the importance of freshwater transfer and storage in the region. Whilst Section 4.1 adequately outlines the definition of glacial lake, I

think a brief definition should be defined early in the manuscript to assist framing the focus of the manuscript. Additionally, I would like to see a passage in the results/discussion that analyses active glacial lakes, under which their relation to glacier- hazards and GLOFs can be addressed. The comparison to other glacial lake datasets should be revisited to provide an adequate examination that focuses on discrepancies in the classification methodologies rather than the definition of a glacial lake.

[Response] Thanks for this suggestion. We agree and shifted the focus of our dataset on water resource foremost. Please see the Abstract, Introduction and the other parts.

We put the Definition of glacial lakes in the Section 'Glacial lake inventory methods'. This section is composed of 4.1 Definition of glacial lakes, 4.2 Interactive lake mapping, 4.3 Classification of glacial lakes, 4.4 Attributes of glacial lake data and 4.5 Error and uncertainty assessment. Putting the Definition of glacial lakes here is acceptable by considering the structure of the manuscript.

As suggested, we shifted the focus of our dataset on water resource foremost, so we need not add one paragraph in the discussion to analyze active glacial lakes related to GLOFs.

The previous Section 6.2 'Comparison with other dataset' was deleted as suggested by the third reviewer to meet the scope of the ESSD articles. In that case, we cannot focus on discrepancies in the classification methodologies in order to avoid an overshooting to interpret the dataset.

**3. Broader overview of remote sensing classification methods**

Optical classification methods are solely focused on in the introduction section of the manuscript (L86-103), which falsely represents them as the sole classification method readily used in remote sensing. I would like to see the overview include other remote sensing classification methods, namely SAR backscatter classification, but also other alternative approaches such as from hydrological sink analysis and from land surface temperature.

[Response] We added other classification methods besides optimal remote sensing. It now reads: 'Backscatter images from Synthetic Aperture Radar (SAR) (How et al., 2021; Wangchuk and Bolch, 2020) were used to remove the impact of cloud cover for lake mapping. Besides, other approaches such as hydrological sink detection using DEM (How et al., 2021) and land surface temperature-based detection method (Zhao et al., 2020) were also used for lake inventories. Different classification methods impact the results of lake mapping and monitoring.'

I am not sure if there are any studies in this region where alternative classification methods are used to detect water bodies; but if there are any then I think they would be a great addition to the dataset comparison section to serve as an inter-comparison of methodologies beyond alike optical classification approaches.

[Response] To our best knowledge, glacial lake dataset produced based on SAR backscatter classification or hydrological sink analysis is not available in the study area. The previous Section 6.2 'Comparison with other dataset' was deleted as suggested by the third reviewer to meet the scope of the ESSD articles. As suggested, we added two sentence to present "what we plan to do next", referring to more robust methods to reduce misclassifications and uncertainty.

**Specific comments**

L41-66: I think this a detailed and concise overview of the importance of glacial lakes and GLOFs in a regional context. However, I think a global perspective is needed to thoroughly illustrate the significance of this study - especially if you are referring to global studies of glacial lakes, such as Shugar et al. (2021). Please include a sentence or two near the beginning

about glacial lakes and GLOFs globally (i.e. importance, general trends etc.)

[Response] As suggested, we have added a sentence herein 'Global glacial lake number and total area both increased between 1990 and 2018 in response to glacier retreat and climate change (Shugar et al., 2020), affecting the allocation of freshwater resource.'

L67-85: You largely focus on remote sensing efforts in HMA regional studies, but there are also references to papers from other regions such as Greenland and the Alps. Either open up this section as an overview of remote sensing studies from all regions, or keep it refined to the HMA region. There have been many regional studies that have been published recently (e.g. Alaska, Rick et al., 2022; Greenland, How et al., 2021), not just in HMA, so I would recommend widening this section to outline the methods in a general context, rather than focusing on HMA.

[Response] Thank you and we have cited the suggested recent publications on other regions. It now reads:

'…the Alaska (Rick et al., 2022), the Greenland (How et al., 2021)…'

L92: What exactly do you mean by object-oriented classification here? This term is generally used in programming rather than in reference to a classification approach. Please change this, or clarify what is meant here; preferably with a more suitable term.

[Response] We deleted 'object-oriented classification'.

L117-119: Are these sub-basins divided by catchments and/or watershed? What determines these sub-basins?

[Response] Yes, these sub-basins are divided by catchment based on major tributary rivers and DEM data.

L132-170: Great outline of data sources.

[Response] Thank you for your positive comment.

L178: Why are landslide-dammed lakes irrelevant to glaciation? Can some glacial lakes also be landslide-dammed lakes?

[Response] In this study, we accept the definition of a glacial lake as one that formed as a result of modern or ancient glaciation. Landslide-dammed lakes formed behind landslides, and have little connection with glaciation. Landslide-dammed lakes vary greatly with time and differ from glacial lakes, hence being exceeded in our dataset.

In a particular situation, glacial lakes are also dammed by landslides, someone may define those lakes as landslide-dammed lakes. Our study focuses on all glacial lakes formed as a result of glaciation.

L199: Change 'the method automatically generated the histogram...' to 'the method calculated the histogram...'

[Response] Revised as 'Specifically, the method calculated the NDWI histogram based on the pixels with each user-defined and manually-drawn region of interest.'

L201: Change 'interactively' to 'manually'. In reference to this comment and the last, I think it needs to be clear in the methodology how this approach is 'semi-automated'.

[Response] We think 'interactively' is more suitable than 'manually' to depict the process of lake mapping. We needed to switch the screening NDWI and original image to determine an optimal threshold, and this is an interactive process. In the process of interactive lake mapping, manual inputs refer to drawing user-defined region of interest (ROI) and tuning the NDWI threshold in each ROI, whereas calculating the histogram of NDWI and converting raster lake extent to vector polygon are automated. To avoid the misunderstanding, we define the method as a human-interactive and semi-automated lake mapping method and made some revision. It now reads:

'Specifically, the method calculated the NDWI histogram based on the pixels with each user-defined and manually-drawn region of interest. The NDWI threshold that separates lake surface from land was interactively determined by screening the NDWI histogram against the lake region in the imagery (Nie et al., 2020; Wang et al., 2014). This way, the determined NDWI threshold can be well-tuned to adapt various spectral conditions of the studied glacier lakes. The raster lake extents segmented by the thresholds were then automatically converted to vector polygons.'

L224-228: False classifications from cloud and topographic shadows can be eliminated with cloud and terrain masking, which are well-established remote sensing methods in land classification. Why did you choose not to include this in the automated component of your workflow?
[Response] In this study, we minimized the effect of cloud cover or shadow by selecting high-quality images and redid the semi-automated lake mapping based on alternative image if glacial lake extents were contaminated by cloud or shadow in previous image. Our method meets the needs of lake mapping. We did not implement a cloud and shadow masking step in our processing of this study. Incorporating cloud and terrain masking in the automated process is an excellent suggestion, and we are considering this in the future research.

Table 1: The characteristics of a proglacial lake should specify that these lakes share a boundary with the ice margin, according to your definition - 'shared boundary' is a better description than 'connected with glaciers' as this could be interpreted as hydrologically connected instead of physically adjacent.
[Response] Replaced 'connected with glaciers' with 'shared boundary with glaciers '.

Table 2: There must be occurrences where a lake's formation and/or dam material properties are ambiguous (especially in relation to GLCS2), even from Google Earth imagery. I see in the dataset that there are no instances where a lake's classification is determined as uncertain; even though you state later on that occassional misclassifications are inevitable (L561). In such instances of ambiguous lake types, how do you decide the classification?
[Response] Yes, some dam material properties are ambiguous from satellite observations. This is a challenge for GLCS2. Differentiating moraine-dammed and glacial-erosion lakes is challenging due to unclear moraine dam or bedrock superimposed by top moraine. To differentiate those dam types, we considered auxiliary factors that help classify lake dam types, such as location, surface slope, roughness and shape of the glacial lakes. We established the classification system of lake types and collected typical samples for each lake type to train our operators at the beginning of the classification. We then used our expert knowledge to classify all lakes with a combination of glacier data, DEM, geomorphological features. When indeterminate lake types emerged, we used group discussions to attribute the type. All these steps help us improve the quality of lake datasets that are more useful to users.

We proposed the Section 'Limitation and updating plan' in the main text:
'Although very high-resolution Google Earth images were utilized to assist in lake type interpretation, occasional misclassification was unavoidable. We implemented two types of classification systems based on a careful utilization of glacier data, DEM, geomorphological features and expert knowledge. However, the lack of in situ survey prohibited a thorough validation of the glacial lake types.'

L272-273: Please provide references to studies that use lake perimeter and displacement error to estimate uncertainty.
[Response] We added the citations, and it now reads:
'Lake perimeter and displacement error are widely used to estimate the uncertainty of glacier and lake mapping from satellite observation (Carrivick and Quincey, 2014; Hanshaw and Bookhagen, 2014; Wang et al., 2020).'

L270-295: Repetition with the corresponding uncertainty estimation section in the supplementary materials. I would suggest refining this section in the main manuscript, and keeping the full description for the supplementary materials.

[Response] We removed the repetition on uncertainty estimation in the supplementary material, and moved Tutorial for Improved Uncertainty Estimating Method at the end of the main text as an appendix.

L294 and L305: Change figure names from Figure S3a/S3b to Figure 3a/3b, unless you would rather move them to the supplementary materials.

[Response] Revised accordingly.

Figure 4: This is a great figure. Please add labelling to the figure to indicate that one set of graphs is from Landsat and the other from Sentinel - you only understand this once you have read to the end of the caption, and it needs to be signposted earlier.

[Response] We have added Landsat and Sentinel-2 at the upper left corner of the previous figure 4 to differentiate.

L339: 'proglacier' >> 'proglacial'

[Response] According to suggestions from Reviewer #1, we replaced 'proglacial' with 'ice-contact' and corrected the typo.

Figure 5: The four maps are somewhat repetitive and it is difficult to see differences between the Sentinel and Landsat lake sizes/abundance from this. I would suggest changing this figure to have an overview map on the left showing all detected lakes from both methods, and a series of inset maps to the right displaying a closer look at certain regions of interest; divided into Sentinel and Landsat lakes. Also, maybe change the outline colour of the lake points to a darker shade, as it is hard to identify the lake points in the current figure.

[Response] The figure 5 aims to describe the distribution of glacial lakes in 2020 extracted from Landsat and Sentinel-2, and all lakes are classified by GLCS1 and GLCS2. From this point of view, it is not repetitive. For clarity, we added 'Landsat' for 'panels a and b', and 'Sentinel-2' for 'panels c and d'. Meanwhile, differences in glacial lakes from Landsat and Sentinel-2 can be clearly seen compared to "Panels a and c" using GLCS1 and "Panels b and d" using GLCS2. A closer look at certain regions of interest is showed in figure 10 and 11. That is why we designed this Figure.

As suggested, we set the outline of lake points to black with a thicker size in order to better differentiate the lake points.

L352: This is a hanging line, and I am not sure which panels and sub-graphs are being referred to here. Does this belong somewhere else or is this a fault with the journal formatting?

[Response] We corrected this typo and ensure all captions are complete in the main text.

Figure 7 and 8: These graphs are very effective at showing changes in lakes - a refreshing take on presenting this type of dataset.

[Response] Thank you very much.

L382: '...while the area grew by a less extent (1.21 km$^2$ or 1.42%).' >> '...while the area grew by 1.21 km$^2$ (or 1.42%).'

[Response] Revised as 'while the area grew by 1.21 km$^2$ (or 1.42%).'

L408: '...including being stable for Shingo...' >> '...including a stable trend for Shingo...'

[Response] The sentence was deleted to avoid an overshooting.

L411-412: 'The total numbers of Kashgar and Hunza basins decreased...' >> 'The total number of lakes in Kashgar and Hunza basins decreased...'
[Response] The sentence was deleted to avoid an overshooting.

L426-27 and Table 5: Can you include some statistics on the link between lake size and % overlap between the Sentinel-2 and Landsat counts? - this would help gauge how much spatial resolution (differentiated from image acquisition) affects lake classification in this study.
[Response] Thanks for this suggestion. We tried but did not find any significant statistics between lake sizes and overlapping rates. Impact of spatial resolutions on classification accuracy is very interesting. However, we have to present our lake dataset rather interpret in order to meet the scope of ESSD papers. In that case, we did not analyze the impact in this study.

L441-443: Are these lakes persistently large or just at a particular time step? Do you have evidence as to why they are disproportionally large?
[Response] The paragraph was deleted to avoid an overshooting.

L481-485: Can you state here the number of instances where overlapping acquisitions were acquired?
[Response] Revised as '…is relatively low (only 7 scenes of Sentinel-2 images or 112 glacial lakes in 2020)…'

L492: 'approximate' >> 'approximately'
[Response] The sentence was deleted to avoid an overshooting.

L503-504: Are there any studies that present glacial lake datasets derived from Sentinel-2? If so, please reference them here. If not, then change this to state that there are no comparable datasets, rather than a scarce number of datasets.
[Response] The previous Section 6.2 was deleted as suggested by the third reviewer to meet the scope of the ESSD articles.

L515-521: I think, similar to your suggestion regarding landslide-dammed lakes, a likely answer is that Wang et al. focus more on glacier-connected lakes, given that they adopt a 10 km buffer to filter out unconnected lakes. And therefore they identify an increasing trend, possibly reflective of a subset of your lake types. Could you subset your lake dataset to match the lakes identified by Wang et al., and examine whether you also see this increasing trend evident in your subsetted dataset? (And perhaps also include landslide-dammed lake for the purpose of this comparison?)
[Response] The previous Section 6.2 was deleted as suggested by the third reviewer to meet the scope of the ESSD articles.

L530: "Zhang's dataset..." >> "The dataset from Zhang et al. (2015) ..."
[Response] The previous Section 6.2 was deleted as suggested by the third reviewer to meet the scope of the ESSD articles.

L531-536: I am unsure how this study and Zhang et al. could have discrepancies in image availability when both studies are classifying lakes from the same satellite image collection (Landsat). Some clarification is needed here to demonstrate how your dataset could classify these lakes when Zhang et al. could not.
[Response] The previous Section 6.2 was deleted as suggested by the third reviewer to meet the scope of the ESSD articles.

L538-544: Discrepancies in glacial lake datasets can be becasuse of minimum lake size, classification method (i.e. not just optical), image acquisition and post-filtering. However, if the purpose of this dataset is to 'further promote the capacity of GLOF risk assessment and

predicting glacier evolutions' then I am unsure why there is no spatial filter (relative to ice margin position) adopted to remove lakes that are unconnected to the glacial system. I think the focus of this study needs to be shifted (as stated earlier in major comments), and further analysis needs to be presented that demonstrates changes in GLOF and glacier-fed lakes specifically (i.e. filtered by lake type and size) - see major comments for full details.

[Response] The previous Section 6.2 was deleted as suggested by the third reviewer to meet the scope of the ESSD articles.

L550: 'Even though an capacity of repetitive observations...' >> 'Even though the capacity of repeat observations...'

[Response] Revised as 'Even though the capacity of repeat observations...'

L557: '...inter- and intra-annual changes (Liu et al., 2020) in glacial lake dataset of each time period...' >> '...inter- and intra-annual changes (Lie et al., 2020) for each time period...'

[Response] Revised as '...inter- and intra-annual changes (Liu et al., 2020) for each time period...'

L545-564: This is a valuable section to include in the study. The temporal range of these datasets and limited image availability (especially in formative years) will not adequately capture the dynamic nature of draining glacial lakes; and therefore such datasets serve as a gauge of long-term, regional trends rather than individual lake change.

[Response] We agree and are thankful for your comprehension.

L565-575: It is great to hear that this work will be continued, and new time steps will be included in the dataset in the future.

[Response] Thanks for your encouragement.

L584: '...spatial-temporal changes at longer time scale...' >> '...spatial-temporal changes at a longer time scale...'

[Response] Revised as ' ...spatial-temporal changes at a longer time scale... '.

L584: 'observation' >> 'observations'

[Response] Revised as ' observations '.

L585: 'started' >> 'starting'

[Response] Revised as ' starting '.

L595-596: See previous comment from L538-544 regarding this statement.

[Response] This sentence was changed to:

'Our glacial lake dataset contains a range of critical parameters that maximize their potential utility for water resource and GLOFs risk evaluation...'

L602: 'values for cryospheric-hydrology research, assessment of...' >> 'value to cryospheric-hydrology research, the assessment of...'

[Response] Revised as 'value to cryospheric-hydrology research, the assessment of... '

---

## Author Response (AR3)

Response to the editor

16 Jul 2022
Topical Editor decision: Reconsider after major revisions
by Kenneth Mankoff
Comments to the author:

Dear Authors,
The manuscript is much improved based on the two rounds of reviews, but I still have some concerns about the depth of the validation, and the presentation of some of the data. I would like to see additional changes before I again consider publication.
[Response] Thank you very much your constructive comments. Here we respond all your comments point by point.

L62: Citing "Mankoff /et al./, 2020" here is not relevant and therefore incorrect. Unfortunately this citation now raises concern with the rest of your citations, which I am not as familiar with.
[Response] Thank you. We have removed this incorrect citation and went through the main text to make sure all citations are reasonable and relevant.

Eq1: Missing "d" from "Ban_NIR"
[Response] This has been corrected to 'Band$_{NIR}$'.

Table 1 should be labeled GLCS1 and Table 2 should be labeled GLCS2 in the caption.
[Response] We have corrected this to: 'Table 1. Classification system of glacial lake types (GLCS1) …' and 'Table 2. Classification system of glacial lake types (GLCS2) …'

L297: Shouldn't lake elevation be constant? If it isn't this is a metric of DEM quality?
[Response] Thank you for this question. Yes theoretically, the lake elevation should be fairly constant. However, we found that lake elevation varies slightly within the lake extent due to DEM quality, acquisition date of the used DEM and lake evolution. So we agree that this may be a metric of DEM quality. As a result, and decided to employ the widely used SRTM DEM (acquired in 2000) to calculate the centroid elevations to represent individual lake elevations. We did not analyze the accuracy of DEM quality which is beyond the scope of this study.

L300: I tried to download your data to look at Lake Volume but could not download it. Do you include lake volume uncertainty? I notice in the version I have downloaded there is "Perimeter" "Area" and "Uncertain". I assume this is area uncertainty and not perimeter uncertainty? I could not be sure about this. Is Orbital number and image source the actual scene ID? Or just path/row? It is important to easily be able to find the source/original image. I suggest include SceneID, not just path/row and time/date, which can be used to figure out scene ID, but is complicated.
[Response] We have updated our lake dataset at the Mountain Science Data Center. The dataset is now open to reviewers and the editor, please download from
https://cloud.imde.ac.cn:5001/sharing/nyzEXaYha with a password of imde123456. The dataset

will be open access after the publication of our article. Before that, the dataset is available upon reasonable requests to the corresponding author.

This uncertainty attribute is the uncertainty of the mapped lake water area. For improved clarity, we have renamed the attribute name to be 'AreaUncer'.

We have also replaced "IMGSOURCE" by "SceneID" which contains identifying information, consisted of the orbital ID, sensor ID and acquisition date (YYYYMMDD) for Landsat image, or the orbital ID and acquisition date (YYYYMMDD) for Sentinel-2 image. Everyone can identify individual original Landsat or Sentinel-2 image using a combination of the orbital ID and acquisition date, which is unique. In the pre-processing, we unzipped the original image files, did band composition and renamed the new stacked bands using our defined naming rule, and did not keep all original images for saving storage volume. Herein, we can not fill the SceneID using original image file names. Thank you for your understanding.

Table 3: Volume units should be cubic not square.
[Response] Revised to 'cubic meter'.

Figure 8: Should this be ylog?
[Response] As your suggestion, Figure 8 has been revised to xlog for a better presentation, considering that y-axis represents relative error (%). Then we also deleted the previous panels b-d and f-h.
The Figure 8 has been revised to be:

[Figure]

Figure 8. Estimated relative error for glacial lakes of all or specific size ranges in study area. Error estimation is based on the modified equation and lake data extracted from Landsat (a) and Sentinel-2 images (b).

Figure 9: I'm concerned that you're presenting log-distributed data over a wide range so that it appears to fit well, but the small values may have large disagreement. This can be remedied by using a log scale (if appropriate - I'm not sure if it is), and possibly using a Tukey mean-difference plot (a.k.a Bland-Altman plot). I used this, in combination with a heatmap density plot, in my 2020 "Freshwater" paper https://doi.org/10.5194/essd-12-2811-2020 (don't cite it! Not appropriate!) in case you want to see what I think might be a better presentation than Fig 9.

[Response] Thank you. We adopted Bland-Altman plot to present the results validated by Google Earth high-resolution images derived lake data in Figure 9c-d.

The Figure 9 has been revised to be:

[Figure]

Figure 9. Distribution of validation samples (a), comparison of glacial lakes derived from Landsat and Sentinel-2 overlaying Google Earth image (© Google Earth 2019) in a zoomed site (b), and glacial lake product validated by Google Earth derived lake boundaries (c and d).

L501: I still disagree that you present evidence of dramatic changes between 2020-08-25 and 2020-09-19 and that this is not just changes due to satellites in Fig. 11. Furthermore, the 'dramatic' change seen in panels c1/c2 is same date and therefore only due to satellite. Writing of c1/c2, why are the two dark blobs in the lower right quadrant not considered lakes?
[Response] We agreed that the detected lake discrepancy resulted from both satellite image sources and the temporal changes of glacial lakes. To absorb this point, we have modified this sentence to be: '… The example is given to supraglacial lakes which showed considerable changes during a short period of time between the applied Landsat and Sentinel-2 images …'
Panels c1/c2 were designed to represent that some narrow portions of the lakes (Figure 11 c1/c2) were mapped from Sentinel-2 imagery but were unable to be detected from Landsat imagery. The two dark blobs in the lower right quadrant were mapped as glacial lakes in Sentinel-2 derived dataset (not shown in previous panel c2, now we added the lake boundaries in panel c2.) but not included in Landsat-derived dataset due to lake areas less than the minimum mapping unit of 4500 m$^2$.

Finally, you have one validation section against Google Earth. Unfortunately Google Earth is

non-reproducible. Because of the nature of Google Earth I do not know what images you validated against. It is even possible that Google presented some of the same images to you, so you were performing a self-validation with high autocorrelation and the differences are only due to operator or interface differences. I would still like to see a more thorough validation against existing data products, and I note other reviewers have requested this too.

[Response] Google Earth images can be qualified as "valid" and "good" validation data, because (1) they are of higher resolution than our mapping source images and (2) we have digitized the lake boundaries carefully from the Google Earth. During the digitization, we zoomed the Google Earth images to be very high resolution scale in order to survey the lake boundary in situ nearly and avoid using any of the same images (Landsat or Sentinel-2 images). Truly, we do not know what images we validated against, but we make sure digitizing a lake boundary based on image with a spatial resolution of ∼2 m. To shorten the difference caused by acquisition date, the Google Earth images we used for validation were mainly acquired in the circa 2020 (2016-2021) from the similar date of our mapping source images around 2020.

In Section 4.5.2., we have revised this to 'A total of 89 glacial lakes were selected by stratified random sampling and manually digitized based on the Google Earth images in circa 2020 with a spatial resolution of ∼2 m acquired from WorldView, GeoEye, Pleiades etc. satellites to further validate the absolute error of our glacial lake products in 2020 …'

We did not use any Landsat or Sentinel-2 images in the Google Earth as a reference to digitize lake boundaries, so we removed a self-validation with high autocorrelation.

As your suggestion, we added a validation against existing Landsat-extracted data produced by Wang et al. (2020). It now reads 'We also validated the sampling Landsat-derived 89 lakes by the existing Landsat-extracted lake data produced by Wang et al. (2020). A total of 83 lakes are available in Wang's data with a mean difference of 0.005 $km^2$ in lake area (Figure A8). This also shows an improvement of our lake product in contrast to the existing dataset.'

In your latest reply-to-reviewers you have removed a section "Comparison with previous similar dataset". I think a quantitative version of this comparison, carefully comparing like with like, is a necessary requirement for publication of this data set. Please add it back in. The last version had a good discussion of different methods, but did not include a like:like comparison of a subset of the data.

[Response] As requested we have added back the section "Comparison with previous similar dataset". We also did a specific areal comparison for lakes that appeared in both inventories (with the same minimum lake size). Accordingly, we have also revised the related main text. Thank you for this suggestion.

---

## Author Response (AR4)

Comments to the author:

We sincerely appreciate the patient and constructive comments from this reviewer, which are crucial to helping us further improve the quality of our manuscript. We have addressed each of the reviewer's comments as thoroughly and meticulously as possible, and believe have provided sufficient validation and justification for our datasets. We invite the reviewer and the editor to see our responses below.

+ Figure 9 X axis: Mean of lake areas from only one product, not two. I assume Landsat and Sentinel 2, *not Google Earth*. Google Earth is only included in the Y axis?

[Response] We are sorry for the confusion of the axis label. The Y-axis in Figure 9c/d shows the differences between our mapping results (from Landsat/Sentiel-2 images) and the validation reference (digitized from Google Earth imagery), and the horizontal axis reflects the means of glacier lake areas from both our mapping (Landsat/Setinel-2) and the reference (Google Earth). As for the X-axis, we prefer using the means (as a compromise) of our mapping and the reference, rather than our mapping alone, to better reflect the scales of each of the lake areas. We hope the reviewer finds this reasonable and acceptable.

[Figure]

Figure 9. Distribution of the validation sample (a), visual comparison of glacial lakes derived from Landsat and Sentinel-2 images overlaying Google Earth imagery (© Google Earth 2019) in a zoomed site (b), and differences between our glacial lake product (mapped from Landsat and Sentinel-2 images) and the validation reference (digitized from Google Earth

images) (c and d).

+ L506 to 509. You cannot say that the lakes changed due to time, when you are comparing across sensors. Clearly c1/c2 are only due to sensor (same date). The same is possible of panel (d). We don't know.
[Response] Thank you very much for pointing this out. We agree that the lake area changes are attributed to not only temporal difference but also sensor difference. For improved accuracy, we have clarified the initial sentences of this paragraph as below:

"In addition to the difference in image resolution, different acquisition dates between Sentinel-2 and Landsat images can also contribute to the discrepancy of those two glacial lake datasets. Acquiring same-day images from the two sensors were not always possible due to the impacts of cloud contaminations, topographic shadows, snow cover and revisit periods (Williamson et al., 2018; Paul et al., 2020). As exemplified in Figure 11d, the mapped glacial lake areas exhibit a substantial discrepancy, which is likely a joint consequence of both sensor difference and actual glacier lake dynamics that occurred during this short period of time."

+ Table 4 - you're finally getting towards a robust validation. I've repeatedly asked you to compare using properties that exist in both datasets. Why not remove the first row which is only S2 so that you can also compute useful statistics with the bottom Total row? If you want, you can of course mention in the text or figure caption that S2 column excludes 4969 lakes in the 0.0005 to 0.0045 km^2 range.
[Response] Thanks for your constructive suggestion, which we have adopted now. Please see the revised Table 4 below.

Table 1. Count and area of glacial lakes mapped from Sentinel-2 and Landsat images in 2020 in various size classes.

| Lake size km$^2$ | Glacial lakes from Sentinel-2 count (km$^2$) | Glacial lakes from Landsat count (km$^2$) | Overlap % (%) |
|---|---|---|---|
| 0.0045-0.05 | 2182 (35.52±3.72) | 1870 (31.47±9.57) | 85.70 (88.60) |
| 0.05-0.1 | 237 (16.37±0.89) | 204 (14.07±2.18) | 86.08 (85.95) |
| 0.1-0.2 | 122 (16.88±0.68) | 115 (15.91±1.83) | 94.26 (94.25) |
| ≥0.2 | 50 (27.20±0.54) | 45 (24.86±1.40) | 90.00 (91.40) |
| Total | 2591 (95.97±5.83) | 2234 (86.31±14.98) | 86.22 (89.93) |

Note: Second column excludes 4969 (7.73±0.54) lakes in the 0.0005 to 0.0045 km$^2$ range. Overlap % (%) represent the ratios between our Landsat-derived dataset and Sentinel-derived product in count and area, respectively.

+ I would also like to see the Table 4 methods (comparing using 1:1 properties, so the comparisons are useful) done with 3rd party data - that is external validation. Table 5 (and associated analysis) should be reformatted. It may take a lot of work to for example filter Zhang et al., 2015 with an MMU of 2700/3 to match your 4500/5. Perhaps that is not needed. But it should be fairly easy, given your metadata and familiarity with your own product, to filter yours to match all the other products with MMU pixels > 5.
In my last comments to you I thought this request was clear. I apologize if it was not. I requested

"a like:like comparison" in the section "Comparison with previous similar dataset" which I took to mean external data sets, not comparison within your own dataset.

[Response] We really appreciate this constructive suggestion, and completely agree that a "apples-to-apples (like:like) comparison" is more useful and valid. After a careful deliberation that considered both scientific value and feasibility, we decided to perform a thorough comparison between our mapping and each of the third-party products (as listed in Table 5) based on the possible minimum mapping unit (MMU) for both datasets. For example, the MMU in the dataset of Zhang et al. (2015) is 3 pixels, finer than 5 pixels in our product, so a MMU threshold of 5 pixels was used for this comparison. The other comparisons in Table 5 all follow this MMU logic. For improved clarity, we have reformatted Table 5 to reflect this change. We kindly invite the reviewers to see our revised Table 5 below.

Table 2. Comparison between our Landsat-based mapping and other third-party Landsat-based glacial lake datasets in the study area.

| Baseline year (period) | Method | MMU m$^2$ (pixels) | Count (km$^2$) | Other data/our product % (%) | Reference |
|---|---|---|---|---|---|
| 1990 (1988-1993) | Manual | 5400 (6) | 1720 (89.68±13.69) | 83.13 (105.87) | Wang et al., 2020 |
| *1990 (1989-1994)* | *Semi-automated* | 5400 (6) | *2069 (84.71±14.41)* | | *This study* |
| 1990 (1990-1999) | Automated | 50000 (55) | 145 (20.28) | 38.77 (36.98) | Shugar et al., 2020 |
| *1990 (1989-1994)* | *Semi-automated* | 50000 (55) | *374 (54.84±5.49)* | | *This study* |
| 1990 (1989-1992) | Manual | 4500 (5)* | 622 (51.93±10.15) | 28.88 (61.02) | Zhang et al., 2015 |
| *1990 (1989-1994)* | *Semi-automated* | 4500 (5)* | *2154 (85.10±14.66)* | | *This study* |
| 2000 (1999-2001) | Manual | 4500 (5)* | 724 (61.41±11.91) | 33.15 (71.32) | Zhang et al., 2015 |
| *2000 (1996-2004)* | *Semi-automated* | 4500 (5)* | *2184 (86.10±14.83)* | | *This study* |
| 2000 (2000-2004) | Automated | 50000 (55) | 155 (22.35) | 42.94 (40.70) | Shugar et al., 2020 |
| *2000 (1996-2004)* | *Semi-automated* | 50000 (55) | *361 (54.91±5.40)* | | *This study* |
| 2008 | Automated & Manual | 8100 (9) | 1067 (65.45) | 59.28 (78.08) | Chen et al., 2021 |
| *2000 (1996-2004)* | *Semi-automated* | 8100 (9) | *1800 (83.82±13.59)* | | *This study* |
| 2015 (2015-2018) | Automated | 50000 (55) | 148 (21.45) | 40.66 (39.11) | Shugar et al., 2020 |
| *2020 (2016-2020)* | *Semi-automated* | 50000 (55) | *364 (54.84±5.41)* | | *This study* |
| 2017 | Automated & Manual | 8100 (9) | 1063 (63.23) | 58.63 (75.45) | Chen et al., 2021 |
| *2020 (2016-2020)* | *Semi-automated* | 8100 (9) | *1813 (83.80±13.63)* | | *This study* |
| 2018 (2017-2018) | Manual | 5400 (6) | 1956 (102.46±15.48) | 91.02 (119.24) | Wang et al., 2020 |
| *2020 (2016-2020)* | *Semi-automated* | 5400 (6) | *2149 (85.93±14.74)* | | *This study* |

Note: MMU represents the minimum mapping unit that is possible to enable a valid comparison between our product and each of the third-party datasets. * The MMU in the dataset of Zhang et al. (2015) is 3 pixels, finer than 5 pixels in our product, so a MMU threshold of 5 pixels was used for this comparison. "% (%)" represents the ratios between the third-party dataset and our product in count and area, respectively.

For Table 4, our purpose here is to understand the difference in our own mappings caused by Landsat and Sentinel-2 sensors. To explore the sensor impact, we compared our own mapping under different lake size categories in Table 4. However, our validation with the third-party datasets (as in Table 5) are all consistently based on Landsat images. If we replicate Table 4 for

each of the third-party datasets, the validation will be very lengthy, and we are afraid there would be a lot of numbers (generated for each of the lake size categories and for each of the third-party datasets) that eventually become too overwhelming and disorienting for readers to grasp. Therefore, we prefer keeping our validation more succinct (as in Table 5), i.e., against each of the third-party datasets based on the minimum mapping unit (rather than each lake size category). We hope the reviewer finds it reasonable.

**Once again, we very much appreciate the constructive and patient suggestions from the reviewer and apologize for not responding sufficiently well in the last revision. We sincerely hope that the reviewer finds our extended revision this time sufficient and to-the-point. We will be very happy to make further changes should the reviewer find more improvement necessary. Thank you.**

---

## Author Response (AR5)

**Dear Chief Editor (ice) Mankoff,**

We sincerely appreciate your patience and constructive comments, which are crucial to helping us improve the quality of our manuscript. We tried our best to address each of your comments as meticulously as possible and revised typos and grammar throughout the main text. We invite you to see our responses below.

**Comments to the author:**

Table 4: Good. Now can you discuss why one product is 10 - 15 % off of the other product? [Response] Yes, we agree with you that a discussion about the difference between the two products is important. Table 4 is put in the Results, so we discuss the discrepancy in Section 6.2 (Comparison of Sentinel- and Landsat-derived products), please read the analysis there. For clarity, we also revised the sentences as below:

'In each size class, the overlap ratios are greater than 85% in count and area, and there are also a higher number and larger area of glacial lakes from Sentinel than that from Landsat images. Sentinel-2 images (10 m) with a finer spatial resolution produce more glacial lakes than those from Landsat images (30 m). The discrepancy is mainly attributed to the inconsistency of spatial resolutions and image acquisition dates, as discussed in section 6.2.'

Figure 9: You replied, "As for the X-axis, we prefer using the means (as a compromise) of our mapping and the reference, rather than our mapping alone, to better reflect the scales of each of the lake areas. We hope the reviewer finds this reasonable and acceptable.".

I do not find it reasonable. The point of this graphic and section is \*validation\* - that is, independently comparing your product to some other product, so that you and your users can understand quality issues. By averaging your product with the validation product on the X axis, you are hiding the differences between the two that would appear on the Y axis.

[Response] Thank you for these constructive comments. We originally used the means of both methods on the x-axis, following the convention of the Bland-Atlman plot (please see the last figure in: https://en.wikipedia.org/wiki/Bland%E2%80%93Altman\_plot). But we do fully agree that for the validation purpose, it makes better sense to not mix up the two on the same axis. Accordingly, we have adjusted the horizontal axis of Figure 9c-d, and the lake areas derived from high-resolution Google Earth images (the reference values alone) are shown on the x-axis.

---

## Author Response (AR6)

Comments to the author:
Dear Authors,

I'm pleased by the significant effort and improvements you've made on this manuscript, and am happy to accept it for publication. Thank you.

Ken Mankoff
[Response] Thank you for accepting our article.

Non-public comments to the Author:
Dear Yong,

Thank you for this last round of revisions. I think the manuscript is greatly improved from the original submission and I am happy to accept it for publication.

You are also correct and I am incorrect that the Bland-Altman plots do have the average of the two on the X-axis. I leave it to you to decide if you want to present the current version, which I think may be better as a validator because it does not combine the two products on both axes, only on the Y-axis. But if you prefer to use a standard method (Bland-Altman/Tukey) and not a modified method, then you are welcome to revert to the previous version.
[Response] We completely agree with you, and finally let us keep the current version, please.

More importantly, I do request one change, if possible, before publication. Your validation imagery is not "Google Earth images" as you state in the paper. Google Earth only displays them, it does not acquire them. At the bottom of the Google Earth app, it usually says "Copyright" and then Landsat, or Airbus (Plaides satellite?) or Copernicus (Sentinel satellite?).
[Response] We agree and revised the text referring to the Google Earth app, as you suggested. And responded one by one as below.

So...
L35: Correct use of Google if you derived them in Google Earth. If you used Google basemap in QGIS or Arc, then that should be stated.
[Response] We directly derived the glacial lake boundary in Google Earth app, not using Google basemap in QGIS or Arc. So 'validated by Google Earth-derived lake boundaries' is ok.

L257/283: Incorrect use.
[Response] L257, the sentence has been revised as '…according to Landsat and Sentinel-2 images, and Google Earth at a finer scale overlaying preliminarily lake boundary extraction at the given period.' L283, the sentence has been revised as 'characteristics interpreted from Landsat and Sentinel images, and Google Earth.'

Table 1 and Table 2 caption: Correct.
L365: OK.

L461, 462, 464 - if this could be specific, it would be better. But Landsat is sometimes in Google Earth. This could confuse people. What about "...Our Landsat and the imagery provided by Google Earth..." (at a minimum) or better yet, if the "copyright statement" at the bottom of the scene tells you it is usually Airbus and MAXAR, then state that. If possible.

[Response] A better way is to show earlier in Section 4.5.2, L365-367, the sentence has been revised as '… based on the Google Earth images in circa 2020 with a spatial resolution of ~ 2 m acquired from WorldView, GeoEye, Pleiades, etc. satellites (© 2022 Maxar technologies and © 2022 CNES/Airbus)…' We show that the reference lake data for validating our products were delineated from Google Earth higher spatial resolution images provided by Maxar technologies and CNES/Airbus. We did not use any Landsat images in Google Earth. Hence, we can keep current writing for L461, 462, and 464.

L465/466/467 "Google Earth-derived data" is ok.
Other than that, I have no more issues.
Thank you,
Ken Mankoff

[Response] Thank you again, Chief Editor Mankoff, your comments help us greatly improve our manuscripts.

I would be more than happy to make further adjustments during copyediting and proofing.

Best wishes,

Yong